# Proteome-wide copy-number estimation from transcriptomics

Andrew J Sweatt [iD][1], Cameron D Griffiths [iD][1], Sarah M Groves[1], B Bishal Paudel[1], Lixin Wang[1], David F Kashatus [iD][2] & Kevin A Janes [iD][1,3✉]

## Abstract

**Protein copy numbers constrain systems-level properties of regulatory networks, but proportional proteomic data remain scarce compared to RNA-seq. We related mRNA to protein statistically using best-available data from quantitative proteomics and transcriptomics for 4366 genes in 369 cell lines. The approach starts with a protein's median copy number and hierarchically appends mRNA–protein and mRNA–mRNA dependencies to define an optimal gene-specific model linking mRNAs to protein. For dozens of cell lines and primary samples, these protein inferences from mRNA outmatch stringent null models, a count-based protein-abundance repository, empirical mRNA-to-protein ratios, and a proteogenomic DREAM challenge winner. The optimal mRNA-to-protein relationships capture biological processes along with hundreds of known protein-protein complexes, suggesting mechanistic relationships. We use the method to identify a viral-receptor abundance threshold for coxsackievirus B3 susceptibility from 1489 systems-biology infection models parameterized by protein inference. When applied to 796 RNA-seq profiles of breast cancer, inferred copy-number estimates collectively re-classify 26–29% of luminal tumors. By adopting a gene-centered perspective of mRNA–protein covariation across different biological contexts, we achieve accuracies comparable to the technical reproducibility of contemporary proteomics.**

**Keywords** CCLE; CVB3; Pinferna; SWATH; TMT
**Subject Categories** Computational Biology; Proteomics

## Introduction

Proportional numbers of molecules add important bounds to biological systems, but they are hard to come by (Phillips and Milo, 2009). One exception is deep RNA sequencing (RNA-seq) of bulk samples, which provides transcript-per-million (TPM) estimates as a proportion of all expressed genes (Wang et al, 2009). The commoditization of sequencing has made RNA-seq the prevailing omics approach: as of mid-2024, the leading repository (Barrett et al, 2013) contains >63,000 studies with human samples. RNA-seq profiles are useful for reading out the state of the genome (Duren et al, 2017; Fehrmann et al, 2015), but relating transcript copies to the abundance of proteins is complex. In tumor classification, for example, the number and identity of cancer subtypes change when using quantitative measurements of mRNA versus protein (Mertins et al, 2016; Zhang et al, 2014a). The challenge is especially acute for mathematical models in systems biology, which need protein quantities to constrain topology, initial conditions, or transition rates (Lewis et al, 2021; Montagud et al, 2022; Pereira et al, 2020). Filling the overall gap requires new strategies for proportional quantification of proteomes suited to different needs.

Progressive experimental innovations in untargeted mass spectrometry have made quantitative proteomics a reality (Pappireddi et al, 2019). Isobaric labeling approaches such as tandem mass tagging (TMT) now quantify up to 18 multiplex samples and are the method of choice for proteogenomics (Ellis et al, 2013; Li et al, 2021; Thompson et al, 2003). Comparisons of individual proteins across samples indicate that linear mRNA–protein relationships vary greatly in quality (Pearson $R = -0.4$ to 0.8) depending on the gene and gene category (Mertins et al, 2016; Zhang et al, 2014a). Multiplex labeling yields peptide-specific relative quantities, but other analytical methods are needed to give information about proportional differences among proteins within a sample (Ahrne et al, 2013; Pappireddi et al, 2019). Intensity-based quantification of MS1 precursor ions is possible across proteins within a sample, but stochasticity in the ions selected for peptide identification creates sparsity challenges when data from many runs must be combined. One robust alternative is to use data-independent acquisition methods like sequential window acquisition of all theoretical mass spectra (SWATH), which analyzes all precursor ions in a series of mass-to-charge ratio windows (Gillet et al, 2012). After data acquisition, the best-ionizing peptide(s) of a protein are summed by intensity, and the resulting data are centered at a reasonable per-cell average ($10^4$ copies) to estimate proportional copy numbers for the detectable proteome. SWATH is more reproducible but also more computationally intensive, harder to set up, and lower throughput (Collins et al, 2017; Ludwig et al, 2018). The need for proportional protein estimates may forever outpace the ability to generate them directly.

[1]Department of Biomedical Engineering, University of Virginia, Charlottesville, VA 22908, USA. [2]Department of Microbiology, Immunology & Cancer Biology, University of Virginia, Charlottesville, VA 22908, USA. [3]Department of Biochemistry & Molecular Genetics, University of Virginia, Charlottesville, VA 22908, USA. ✉E-mail: kjanes@virginia.edu

As a means for estimating proportional protein copy numbers, an appealing starting point is RNA-seq. mRNA is the template for protein translation, and in terms of scale, depositions of human RNA-seq exceed those of quantitative proteomics (all species, all methods) by ~tenfold (Barrett et al, 2013; Perez-Riverol et al, 2019). However, despite useful transcriptomic inference of protein activities from the gene networks surrounding them (Alvarez et al, 2016), directly estimating proportional protein copy numbers from mRNA is challenging. Protein abundance differences may track somewhat linearly with mRNA when considering all genes within a sample, but it is often difficult for mRNA to predict quantitative protein differences among samples for any given gene (Buccitelli and Selbach, 2020; Fortelny et al, 2017; Tasaki et al, 2022; Wilhelm et al, 2014). The latter is important for systems biology when using transcriptome profiles to instantiate personalized models of function (Lewis et al, 2021; Montagud et al, 2022; Pereira et al, 2020). The current thinking is that the steady-state abundance of mRNA and its translation rate create a general "set point" for protein copies, which are buffered or tuned according to the protein's characteristics and its (de)stabilizing interactions with other proteins in a cell context (Buccitelli and Selbach, 2020; Taggart et al, 2020). Unfortunately, our working knowledge of these characteristics and interactions in mammalian cells remains incomplete (Giurgiu et al, 2019; Richards et al, 2021), which has thus far prevented a bottom-up reconstruction of mRNA-to-protein relationships that are accurate and conditional.

Here, we surmounted this challenge by adopting a top-down perspective that statistically identifies the best working mRNA-to-protein relationship for each gene based on paired data in several hundred cancer cell lines. SWATH and TMT datasets from different sources are encouragingly self-consistent, enabling the meta-assembly used for model training and selection of three relationship classes. Although relationship classes are entirely data driven, we find biological meaning and gene-specific mechanisms in each. The approach consistently improves the accuracy of proteome-wide inferences from RNA-seq transcriptomes of cells, tumors, and tissues when compared to other tools and a stringent null hypothesis specific to each gene's protein set point. We use the method to build 1489 personalized systems-biology models of virus infection (Griffiths et al, 2021; Lopacinski et al, 2021) and re-classify 796 cases of breast cancer (Ciriello et al, 2015) according to inferred proportional protein abundance from public RNA-seq datasets. This study provides an open and accessible route to gleaning protein copy numbers from RNA-seq when it is impractical or impossible to quantify the proteome directly (http://janeslab.shinyapps.io/Pinferna).

## Results

### Deriving three gene-specific biological classes of mRNA–protein relationships

To estimate mRNA–protein relationships, we obtained quantitative proteomics measured by TMT mass spectrometry in 375 cancer cell lines (Nusinow et al, 2020) and placed these data on a proportional scale by using independent SWATH proteomics from two lines—one breast carcinoma (Liu et al, 2017) and one osteosarcoma (Rosenberger et al, 2014)—within the TMT dataset (Figs. 1A, Step 1

and EV1A). Training with a large, diverse panel of cancer lines avoids confounding gene or protein covariations that may arise in primary tissues and tumors because of cell mixtures (Srivastava et al, 2022). When TMT profiles scaled to one reference line were compared to SWATH data measured directly in the other reference line, correlations were above 0.7 in both cases (Figs. 1B and EV1B), placing bounds on the internal consistency of the two data sources. Overall, the meta-assembly yielded protein copy number measurements for 4384 proteins across 375 cell lines.

We discerned mathematical relationships that best captured abundance relationships between mRNA and protein by merging SWATH-scaled proteomics with paired Cancer Cell Line Encyclopedia (CCLE) transcriptomics (Ghandi et al, 2019) and building gene-specific regressions of different classes (Fig. 1A, Step 2; Datasets EV1–EV3). In the simplest case, protein abundance varies nominally around the median (M) regardless of transcript abundance (Fig. 1C). To incorporate abundance information about the transcript, we also evaluated a hyperbolic-to-linear (HL) relationship where low-abundance changes in mRNA cause larger nonlinear changes in protein that linearize as mRNA abundance increases (Fig. 1D; further explanation below). Finally, we considered that the abundance of some proteins may be partly informed by the abundance of other transcripts and applied the least absolute shrinkage and selection operator (LASSO) to residuals of the HL fit (Fig. 1E). The best model among the three for each gene was distinguished by the Bayesian Information Criterion (BIC; Figs. 1A, Step 3 and EV1C–E) to arrive at preferred M, HL, or HL + LASSO relationships for 4366 genes (Dataset EV4). The best model for each gene was strongly preferred over the others in 98% of cases (Fig. EV1F). These relationships create a template for protein inference from RNA (Pinferna) given new samples with transcriptomic profiles (Fig. 1A, Step 4).

We examined characteristics of the genes in each model class. Consistent with previous findings (Mun et al, 2019; Zhang et al, 2014a), M genes with no clear transcript dependence showed gene ontology (GO) enrichments for translation and mitochondrial electron transport (Fig. 2A; Dataset EV5). Although M genes were not longer lived than others (Fig. EV2A,B), we found that they had high transcript abundances (Fig. EV2C) and were enriched in multi-protein complexes in the CORUM database ($P = 3.3\mathrm{e}{-}60$) (Ruepp et al, 2010). Proteins residing in stable complexes may saturate for all measured abundances of mRNA because their copy numbers are stoichiometrically limited by other subunits (Goncalves et al, 2017; Taggart et al, 2020), causing the loss of an observable relationship between mRNA and protein.

Most genes exhibited some dependence on their mRNA (Fig. 1D,E). To incorporate transcript information, we assessed various low-complexity models involving one (linear), two (hyperbolic) or three (3-parameter logistic or HL) free parameters. The four models were compared by BIC, and HL was overwhelmingly the best or near-best model for most genes (Fig. 2B). Results were similar when using BIC weights (Wagenmakers and Farrell, 2004) to assess the relative likelihood of HL against the others (Fig. EV2D). HL accommodated rare log-concave relationships that occurred when protein abundance saturated at high transcript abundances (Fig. 2C). Loss of transcript dependence arises biologically when a protein subunit surpasses the abundance of the complex in which it resides (Taggart et al, 2020; Wuhr et al, 2014). Accordingly, among HL genes, those that were log concave

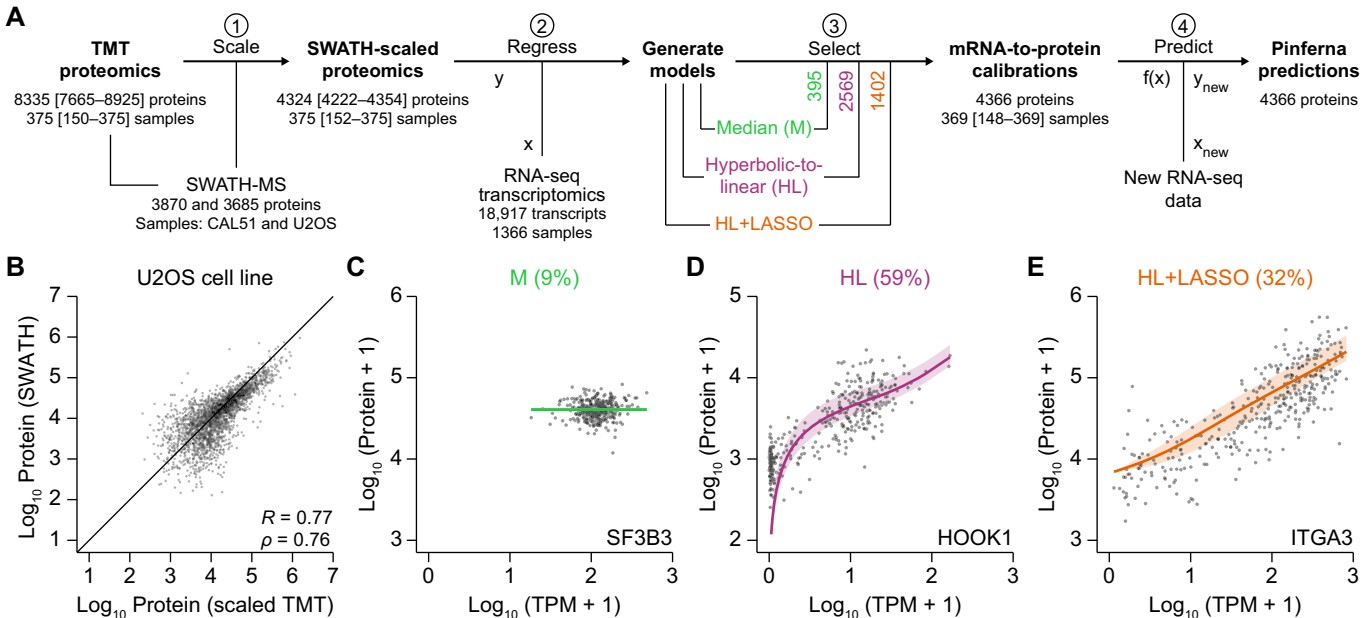

**Figure 1. Meta-assembly and inference of conditional mRNA-to-protein relationships for 4366 human genes.**

(A) Data fusion and model discrimination. (1) Tandem mass tag (TMT) proteomics of 375 cancer cell lines (Nusinow et al, 2020) were calibrated to a proportional scale based on sequential window acquisition of all theoretical mass spectra (SWATH) proteomics of CAL51 and U2OS cells (PXD003278; PXD000954). (2) SWATH-scaled proteins were regressed using three models that incorporate transcript abundance from RNA sequencing (RNA-seq) to different extents: median (M), no contribution of mRNA; hyperbolic-to-linear (HL) relationship incorporating mRNA of the gene, $a \bullet \left( \frac{b \bullet mRNA}{c + mRNA} + mRNA \right)$; HL + least absolute shrinkage and selection operator (LASSO) regressors with mRNAs other than the gene of interest. (3) Model selection for each gene was based on the Bayesian Information Criterion. The number of genes selected in each class is indicated. (4) New samples profiled by RNA-seq were used with the calibrated models to make protein inference from RNA (Pinferna) predictions. The number of proteins measured per sample or number of samples with data per protein is shown at each step as the median with the range in brackets. (B) Reliable cross-calibration of the TMT and SWATH meta-assembly. Step 1 of (A) was performed with CAL51 data alone and the SWATH-scaled TMT proteomics of U2OS cells compared with data obtained directly by SWATH. The reciprocal cross-calibration is shown in Fig. EV1B. (C–E) Representative M, HL, and HL + LASSO genes. Proportional protein copies per cell were regressed against the mRNA abundance normalized as transcripts per million (TPM). Evidence for model selection is shown in Fig. EV1C–E. Data information: For (B), Pearson's $R$ and Spearman's $\rho$ are shown. For (C–E), best-fit calibrations ± 95% confidence intervals are overlaid on the proteomic–transcriptomic data from $n = 369$ cancer cell lines.

were mildly enriched in protein complexes in the CORUM database ($P = 4.0e{-}2$) (Ruepp et al, 2010). More common were log-convex HL relationships in which protein abundance increased only at higher mRNA abundances (Fig. 2D). Some of this behavior was attributable to the ratio compression that occurs when low-abundance proteins are quantified by TMT (Savitski et al, 2013), which we confirmed by quantitative immunoblotting (Fig. 2E,F). However, using a simple computational model for synthesis and turnover of mRNA and protein with dimensionless rate-parameter estimates, we also found that log-convex relationships arose naturally when steady-state abundances of mRNA and protein were halved and randomly sampled along the trajectory back to steady state (Fig. EV2E,F). Such "halving-and-random-sampling" occurs when cells asynchronously undergo cytokinesis, halving the protein copies per cell and sporadically re-entering into G1. Other HL genes showed mixed concavities or were linear to different extents (Fig. EV2G–I). Taken together, HL regressions provided the flexibility needed to capture various biological mechanisms that relate mRNA to protein (Fig. 1D).

One-third of HL regressions were statistically improved by adding mRNA features selected and weighted using LASSO (Fig. 1E). LASSO features were enriched for cytoplasmic translation (GO:0002181; $q = 3.0e{-}25$) and the proteasomal pathway

(GO:0043161; $q = 7.5e{-}18$), suggesting dependencies that may promote protein synthesis or turnover. Specific subunits of the ribosome and proteasome were also enriched among LASSO-selected features ($P_{ribosome} = 3.4e{-}110$, $P_{proteasome} = 7.4e{-}19$). Ribosome feature weights were disproportionately positive, whereas proteasome subunits were negative (Fig. 2G), consistent with their expected influence on protein abundance. Overall, we asked whether LASSO-selected genes were more likely to interact physically with the protein of interest. Using the STRING database (Szklarczyk et al, 2023), we found a remarkable enrichment for interactions among LASSO-selected genes (estimated $P \ll 10^{-4}$; Fig. 2H), indicating that features contain more than spurious statistical associations. We conclude that Pinferna's three-tiered modeling approach captures various biological phenomena and mechanisms in its framework.

## Pinferna predictions in cell lines and tissues relative to competing alternatives

To be useful for new samples, gene-specific model predictions should be more accurate than an arbitrary copy-number estimate of the protein in another setting. Therefore, we assessed Pinferna predictions against a null model of "randomized measurements"

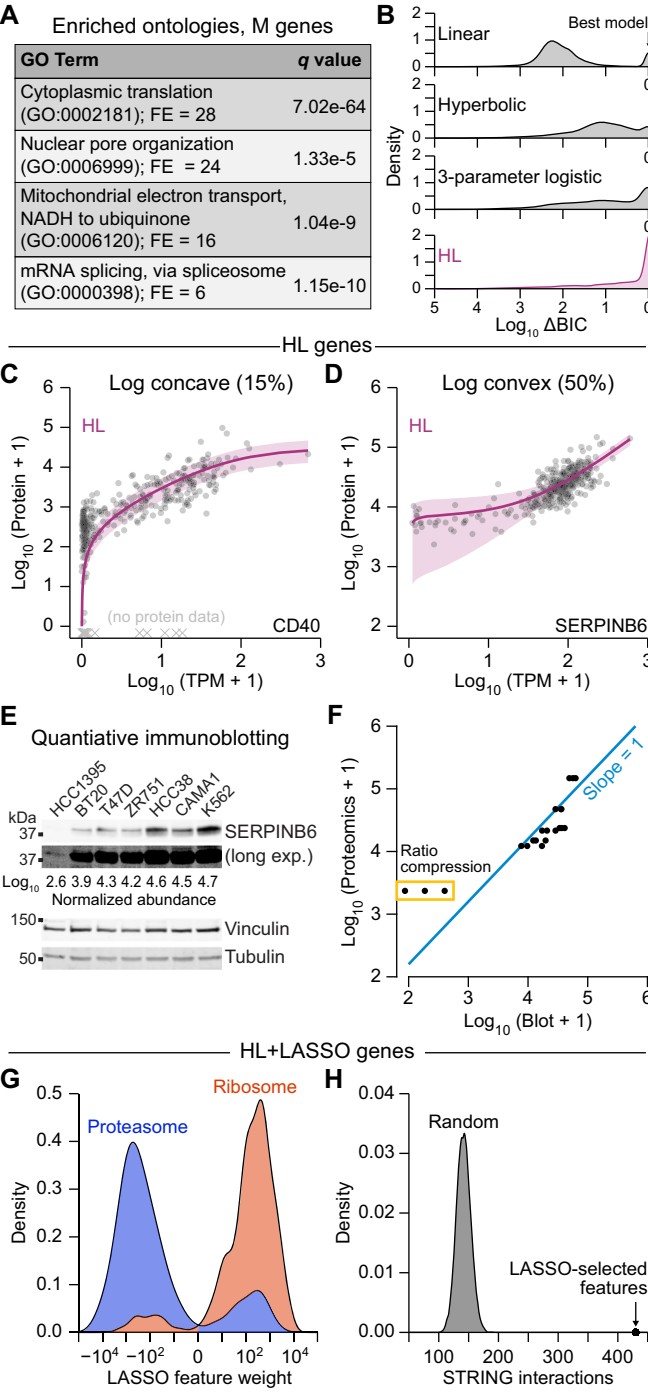

**A** Enriched ontologies, M genes

| GO Term | q value |
|---|---|
| Cytoplasmic translation (GO:0002181); FE = 28 | 7.02e-64 |
| Nuclear pore organization (GO:0006999); FE = 24 | 1.33e-5 |
| Mitochondrial electron transport, NADH to ubiquinone (GO:0006120); FE = 16 | 1.04e-9 |
| mRNA splicing, via spliceosome (GO:0000398); FE = 6 | 1.15e-10 |

**B**

HL genes

**C** Log concave (15%)

**D** Log convex (50%)

**E** Quantiative immunoblotting

**F**

HL+LASSO genes

**G**

**H**

**Figure 2. Pinferna model selection is consistent with known biological mechanisms and mRNA-to-protein relationships.**

(A) Gene ontology (GO) enrichments for M genes. The largest non-redundant GO term is shown with the fold enrichment (FE) and false discovery rate-corrected *p* value (*q*). The complete list of GO enrichments for each relationship class is available in Dataset EV5. (B) HL outperforms competing mRNA-to-protein relationships. Models encoding linear, hyperbolic, three-parameter logistic, and HL relationships were built for all genes and compared by Bayesian Information Criterion (BIC). Results are shown as the smoothed density of BIC differences (ΔBIC) relative to the best model for that gene (ΔBIC = 0). Distributions of BIC weights (Wagenmakers and Farrell, 2004) are shown in Fig. EV2D. (C, D) HL captures different empirical classes of mRNA-to-protein relationships. Log-concave genes (C) saturate at high mRNA abundance, whereas log-convex genes (D) plateau at low mRNA abundance. The remaining genes exhibited characteristics of both fits or linear relationships to varying degrees (Fig. EV2G–I). (E, F) Ratio compression of a log-convex gene. The indicated cell lines were immunoblotted for SERPINB6 along with vinculin and tubulin as loading controls (E), and loading-normalized SERPINB6 abundance was quantified relative to the median copy number for these cell lines in the meta-assembly. Observations were compared to the meta-assembly (F) assuming direct proportionality (blue). (G) Feature weights of HL + LASSO genes are biologically sensible. Smoothed densities of LASSO feature weights (indicating strength and direction of modulation for an HL fit) among mRNAs encoding subunits of the proteasome (blue) and the ribosome (red) are shown. (H) HL + LASSO features are highly enriched for STRING interactions. For each HL + LASSO gene, LASSO-selected features were replaced with random genes to build a null distribution for finding binary interactions in STRING (Szklarczyk et al, 2023). The actual number of STRING interactions among HL + LASSO genes of Pinferna is indicated. Data information: For (B), *n* = 4366 genes. For (F), *n* = 3 biological replicates. For (G), *n* = 127 feature weights (blue) or 397 feature weights (red). For (H), *n* = 10,000 randomizations.

cumulative distribution is sensitive to the correlation between predicted and measured values as well as their direct one-to-one concordance (Fig. EV3A,B).

The first accuracy test was performed with HeLa cells, a line excluded from one of the original meta-assembled resources. We leveraged an independent study that carefully examined HeLa-to-HeLa differences with paired transcriptomics and SWATH proteomics (Liu et al, 2019). Pinferna consistently outperformed randomized measurements for all 12 HeLa derivatives investigated (*P* ≤ 3.4e−7; Figs. 3A and EV3C). Accuracy estimates were comparable when using the reported post-processed protein abundances of (Liu et al, 2019) instead of their raw data quantified with the updated analysis pipeline of the meta-assembly ("Methods"; Fig. EV3C,D). Pinferna predictions were similarly resilient to reductions in transcriptomic sequencing depth—accuracies were comparable down to about 500,000 reads and remained superior to randomized measurements until about 50,000 reads (Fig. EV3E). The results bolster recent claims that typical single-cell RNA-seq data sequenced at ~50,000 reads per cell incompletely reflect the proteome (Brunner et al, 2022; Reimegard et al, 2021) and separately indicate that Pinferna's bulk predictions of protein from mRNA are robust to algorithmic details.

To test Pinferna more broadly against other methods for protein estimation, we integrated the difference between the cumulative distribution functions of a model prediction and the median null to yield a single measure of accuracy improvement (ΔCDF; Fig. 3B). We used ΔCDF to compare Pinferna with two alternative approaches for proportional protein estimation: (1) PaxDb, a meta-repository of protein abundances largely determined by uncalibrated peptide- and spectral-counting methods (Wang et al,

built by randomly selecting an abundance for each protein from measurements of that protein in the meta-assembled dataset originally used for training (Fig. 1A). We quantified accuracy by subtracting the measured value from the predicted value for each protein, taking the absolute value, and dividing by the standard deviation of the individual protein abundance across the 369 cell lines in the training data. This variance-scaled residual inversely weighs error by the breadth of abundances observed in other biological contexts. Finally, we compared the distribution of variance-scaled residuals between Pinferna and 100 null models to arrive at a proteome-wide estimate of model performance. This

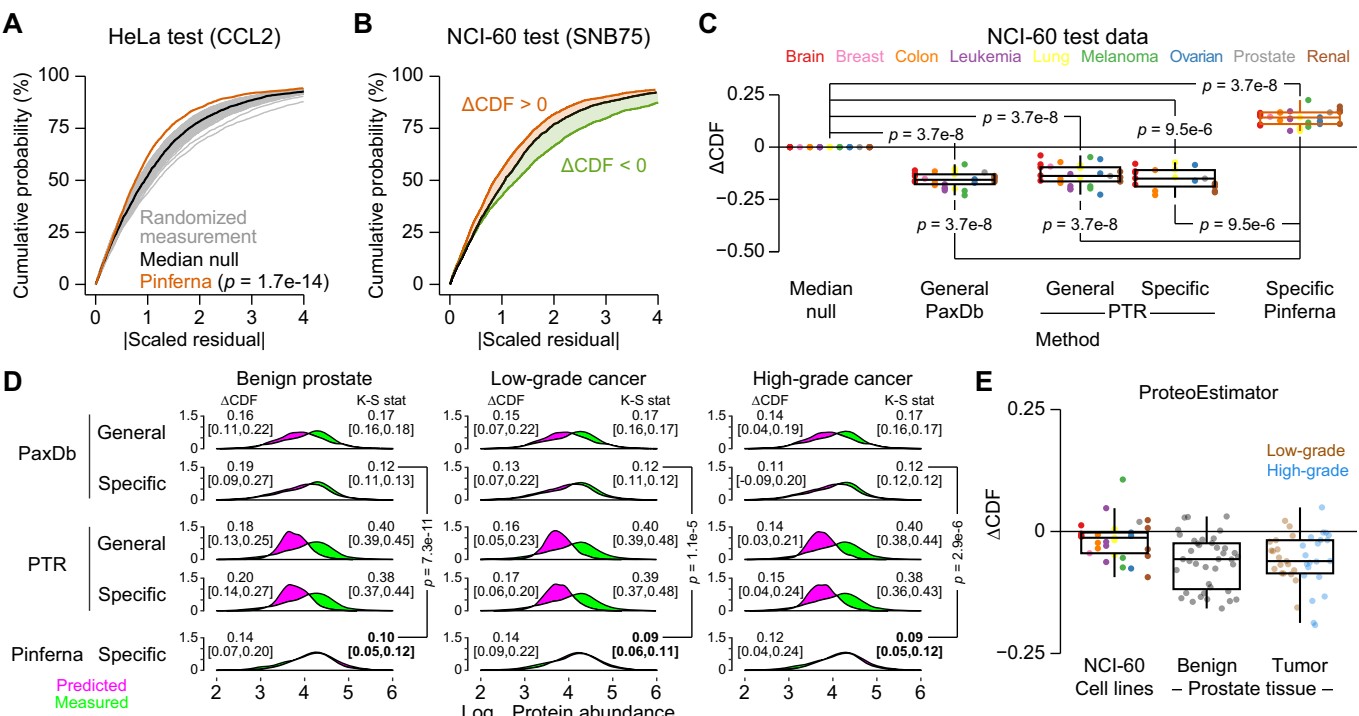

**Figure 3. Pinferna outperforms randomized measurements and competing methods for proportional protein abundance estimation.**

(A) Pinferna compared to random protein-specific measurements. Model predictions were nondimensionalized as a scaled residual by subtracting the measured abundance, dividing by the standard deviation of the SWATH-scaled protein measured across the meta-assembly, and taking the absolute value (|Scaled residual|). The |Scaled residual| cumulative density was compared to randomized measurements drawn from the SWATH-scaled proteomic data for each gene. Randomized measurements were iterated 100 times (gray) to identify a median null (black) that served as a null distribution for model assessment. Left-shifted distributions indicate improved proteome-wide accuracy (relative to each protein's variability) compared to protein-specific randomized measurements. (B) Aggregate performance assessment of protein abundance predictions. The difference in cumulative density functions between test predictions and the median null distribution (ΔCDF) was integrated to identify approaches that performed better (ΔCDF > 0, orange) or worse (ΔCDF < 0, green) than protein-specific randomized measurements. Data are from a prediction of Pinferna (orange) and tissue-specific protein-to-mRNA ratio (PTR; green). (C) Pinferna is superior to empirical guessing in cultured cell lines. ΔCDF values were calculated for NCI-60 cell lines (PRJNA433861; (Guo et al, 2019)) excluded from model training (Fig. 1A) and organized by cancer type. PaxDb (Wang et al, 2015) and PTR (Eraslan et al, 2019) were used generically or in a tissue-specific way as alternative approaches for proportional quantification ("Methods"). A cell line-specific PaxDb estimate was only available for U251 cells. (D) Pinferna reconstitutes proteome-wide distributions of proportional copy numbers in normal prostate tissue (left; case L1N), low-grade prostate cancer (middle; case M7U), and high-grade cancer (right; case H9T) (PRJNA579899; PXD004589; Dataset EV6). (E) ProteoEstimator, the best-performing model of the NCI–CPTAC DREAM proteogenomic challenge (Yang et al, 2020), does not accurately predict protein copy numbers when mapped to a proportional scale ("Methods"). Data information: For (A), Pinferna predictions of HeLa cells (orange; PRJNA437150; PXD009273) were compared to the null distribution by K-S test. For (C, E), n = 5 brain, 1 breast, 3 colon, 4 leukemia, 4 lung, 3 melanoma, 3 ovarian, 1 prostate, 5 renal cell lines. For (D, E), n = 39 normal prostate samples, 19 low-grade, 21 high-grade prostate cancers. For (D), the median ΔCDF and K-S statistic (K-S stat) are reported with range in brackets. For (C, D), differences between groups were assessed by paired sign-rank tests with Šidák correction. Box-and-whisker plots show the median (horizontal line), interquartile range (IQR, box), and an additional 1.5 IQR extension (whiskers) of the data.

2015); and (2) protein-transcript ratios (PTR), which empirically relate mRNA to protein abundances linearly from data collected in 29 tissues (Eraslan et al, 2019). PaxDb and PTR each accommodate generic predictions using all available data and specific predictions restricted to data from a cell line or tissue of interest; both implementations were tested when possible. We applied PaxDb and PTR to the meta-assembly and accounted for method-specific biases by shift-scaling the outputs globally to maximize the ΔCDF for the training data of Pinferna ("Methods"; Appendix Fig. S1). Then, for the comparative evaluation, we assembled transcriptomic and SWATH data from 29 cell lines of nine cancers from the NCI-60 panel that were not included in the meta-assembly (Guo et al, 2019; Reinhold et al, 2019). Overall, Pinferna was significantly more accurate than randomized measurements (*P* = 3.7e-8), whereas PaxDb and PTR were less accurate (Fig. 3C). Although modestly

positive, the ΔCDF value for Pinferna was >50% of the practical maximum for this metric (Appendix Text S1). Results were unchanged when transcriptomics were pre-processed with a different alignment pipeline (Fig. EV3F), reinforcing that Pinferna is tolerant of how mRNA TPMs are calculated. We observed no bias in estimates among cancer types (Fig. 3C) and thus concluded that Pinferna was the preferred method for directly predicting protein copy numbers from mRNA in cultured cell lines.

The performance in cell lines prompted us to ask whether Pinferna estimates would hold in more complex samples such as tissue. We organized transcriptomic and SWATH data for 39 normal human prostate samples and 40 primary prostate cancers (Charmpi et al, 2020) (Dataset EV6). In this setting, all methods performed similarly well with median ΔCDF values ranging from 0.11 to 0.20. However, PaxDb and PTR predictions overall remained systematically left shifted

(Fig. 3D), despite prior shift-scaling to maximize predictive performance ("Methods"). Proteome-wide distributions of Pinferna predictions in prostate were closest to measured distributions, illustrating a dual ability to be quantitatively accurate and representative for SWATH-measured proteins. Moreover, the predictions in prostate demonstrated that Pinferna's proteome-wide copy number estimates from transcriptomics generalize to primary tumors and nonmalignant tissues.

More approaches are available for using transcriptomic profiles to estimate relative differences in proteins among samples (Srivastava et al, 2022; Yang et al, 2020). We tested ProteoEstimator, the best-performing model of 30 in the NCI–CPTAC DREAM proteogenomic challenge (Yang et al, 2020). Relative $\log_2$ fold-change predictions were calibrated to a proportional scale by centering the distribution of each gene about a randomized measurement ("Methods"). The scaled ProteoEstimator predictions remained close to $\Delta$CDF = 0 for all test datasets (Fig. 3E). Separately, we took the NCI–CPTAC transcriptomics and found that $\log_2$ fold-change estimates derived from Pinferna were significantly closer to the measured TMT datasets than the original RNA-seq (Appendix Fig. S2). The results indicated that Pinferna was competitive for relative predictions of protein abundance and distinctive as a tool for proportional predictions of protein copy number.

## Application to in silico modeling

RNA-seq often substitutes for protein quantification when parameterizing systems-biology models of signaling, metabolism, and cell fate (Lewis et al, 2021; Montagud et al, 2022; Pereira et al, 2020). For example, in constructing a mass-action model of cardiomyocyte infection by coxsackievirus B3 (CVB3), RNA-seq was used to estimate abundances of the serial CVB3 receptors, CD55 and CXADR (Fig. 4A) (Lopacinski et al, 2021). Both estimates were HL extrapolations from a very-limited set of SWATH–RNA-seq pairings, which motivated a direct assessment of protein abundance by quantitative immunoblotting with recombinant standards (Janes, 2015; Lopacinski et al, 2021). Direct protein estimation was feasible for cultured cell lines but would be impossible for human hearts, where the severity of CVB3 infections is highly variable (Kim et al, 2001). We reasoned that Pinferna could address this challenge along with similar ones in cancer (Lewis et al, 2021; Montagud et al, 2022; Pereira et al, 2020) and neurologic disease (Tasaki et al, 2022).

RNA-seq data was collected for 1489 healthy and failing human heart samples from the U.S., Australia, and Europe, available through GTEx (Consortium, 2020), MAGNet (Liu et al, 2021), and EGA (Heinig et al, 2017). RNA-seq reads from the three studies were realigned and assembled for comparison (Griffiths et al, 2024). The realignment confirmed no biases in expression based on data source for *CD55* (Fig. EV4A; Dataset EV7). *CXADR* expression increases in cases of cardiomyopathy (Fechner et al, 2003; Kaur et al, 2012), and we reproduced this result by stratifying cases from the three sources ($P$ = 6.2e−245; Fig. EV4B). For both genes, the range of expression in heart samples fell within the variation observed across cancer cell lines in the meta-assembly (Fig. EV4C,D). CD55 is an HL + LASSO gene whose predictions are conditionally dependent on nine other genes. These features reduce CD55 inferences below the smoothed average of the fit at

very low TPM (Fig. EV4C). By contrast, CXADR is an HL gene that is steeply nonlinear at low TPMs where small changes greatly influence the protein copy number estimate (Fig. EV4D). Because of the nonlinear inferences, CD55–CXADR proteins were more strongly coupled than *CD55–CXADR* mRNAs ($P$ = 4.1e−13; Fig. EV4E,F). We re-parameterized an existing model of CVB3 infection (Lopacinski et al, 2021) with the Pinferna estimates of CD55 and CXADR to instantiate model variants that simulate a human population. The individualized models were initiated with a high titer of CVB3 that guaranteed infection of permissive cells, and the concentration of virions was tracked during 24 h of simulated infection ("Methods"). The goal was to investigate whether inferred CD55–CXADR protein variations yielded a wide enough range of infection outcomes in the model that one or both receptors could be nominated as a susceptibility factor.

Examining the predicted distribution of viral loads over time, we noted a strong asymmetry in the onset of infection (Fig. 4B). At 12 h, 64% of individuals were detectably infected, producing mature virions above one plaque-forming unit (1 pfu = 0.48 ± 0.12 nM in these simulations). By 24 h, the models yielded a left-skewed distribution, which straddled the mean lytic yield of viruses in the same genus as CVB3 (~100 pfu = 48 ± 12 nM for a 3700 μm³ cell) (Dunnebacke and Reaume, 1958; Griffiths et al, 2021; Lopacinski et al, 2021). Even at this uncharacteristically late time, 15% of individuals remained uninfected, suggesting they were intrinsically resistant. The remaining cases were best fit as a three-component Gaussian mixture of low, medium, and high susceptibilities (Fig. 4C). Based on mean lytic yield, we interpreted these groups as prone to subinfection, infection, and severe infection, with failing hearts residing almost entirely in the infection and severe-infection groups (Fig. EV4G). For comparison, we abandoned Pinferna and attempted a randomized-measurement approach by linearly scaling the RNA-seq data about a quantity of CD55 and CXADR arbitrarily selected from the training data ("Methods"). As expected, model outputs were so dependent on the randomized measurement that they were uninterpretable when viewed in aggregate (Fig. EV4H). Randomized measurements tended to predict ~100% resistance or ~100% lytic infections and underestimate the low-susceptibility group, although some fortuitously matched the true inferences. We concluded that the Pinferna-derived model outputs were compelling enough to interpret further.

Among heart samples, the distributions of *CD55* and *CXADR* RNA transcripts were quite different (Fig. 4D, E). The range of *CD55* expression was ~tenfold that of *CXADR*, hinting that it might be the dominant receptor for in silico susceptibility. However, these population-wide trends changed when viewed as protein inferences (Fig. 4F,G). Both CD55 and CXADR were more symmetrically distributed, with CXADR exhibiting greater overall variance. Importantly, when individuals were classified based on their inferred susceptibility, we found that CXADR abundance alone was sufficient to stratify the population. This application of Pinferna illustrates how direct substitution of transcriptomics can misconstrue the outputs of systems-biology models built for protein networks.

If modeling predictions derived from Pinferna estimates were correct, then the threshold for CVB3 susceptibility should be ~5000 copies per cell (Fig. 4G). We tested this prediction by engineering non-permissive AC16 cardiomyocyte cells with a doxycycline-inducible and V5-tagged CXADR (iCAR). The resulting AC16-

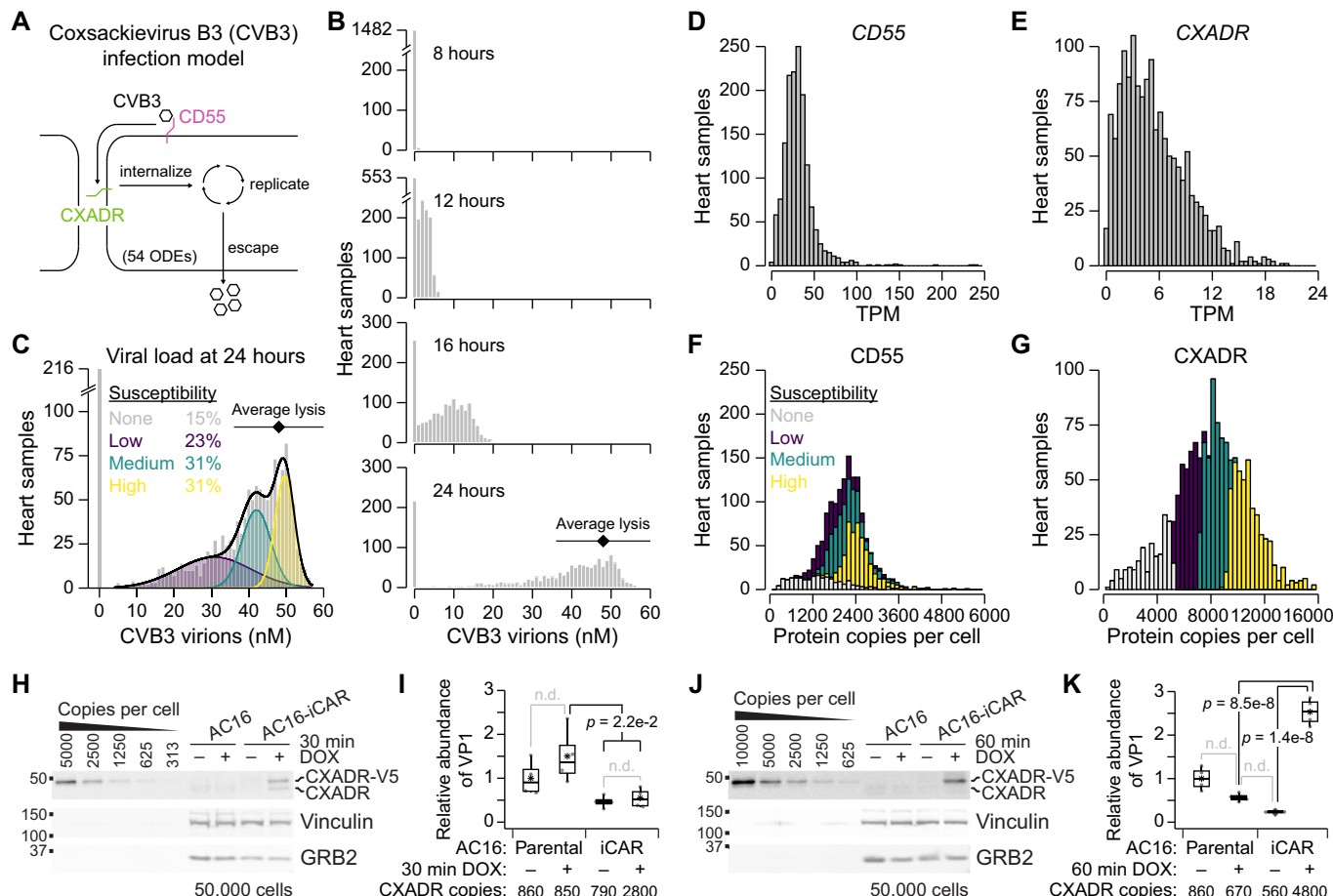

**Figure 4. Simulating degrees of human cardio-susceptibility to coxsackievirus B3 (CVB3) infection based on inferred abundance differences in CVB3 receptors.**

(A) An in silico model of CVB3 initiated by its receptors CD55 and CXADR. After binding, the virus undergoes internalization, replication, and escape. The viral life cycle is mathematically modeled with 54 ordinary differential equations (ODEs; MODEL2110250001). (B) Distribution of viral load over time from 1489 human heart samples. Inferred abundances of CD55 and CXADR from each sample were used to simulate CVB3 infection. Each model run consisted of 100 simulated infections up to 24 h with a coefficient of variation in model parameters of 5%. Viral loads (gray) at the indicated time points are shown along with the estimated point of lysis. (C) Four modes of infection susceptibility to terminal CVB3 infection. Viral load at 24 h was replotted from (B) fit to a Gaussian mixture model (black) of three components (purple, green, yellow). Relative population densities in each of the susceptibility groups is shown along with the estimated point of lysis. (D, E) Distribution of mRNA abundances for *CD55* (D) and *CXADR* (E) normalized as TPM. (F, G) Distribution of inferred protein copy numbers per cell for CD55 (F) and CXADR (G) with each sample colored by its susceptibility. (H–K) AC16 cardiomyocytes were stably transduced with doxycycline (DOX)-inducible CXADR-V5 (iCAR), induced for 30 min (H, I) or 60 min (J, K) and immunoblotted for CXADR with vinculin and GRB2 as loading controls (H, J) or infected with CVB3 (multiplicity of infection = 5) for 6 h and quantified for VP1 (I, K). Immunoblots of (H, J) are quantified in Fig. EV4I,K, and representative immunoblots of (I, K) are shown in Fig. EV4J,L. The average CXADR copy number per cell is indicated for each group (I, K). Data information: For (B, C), the black diamond indicates the mean estimated lytic yield ± s.d. (Dunnebacke and Reaume, 1958; Griffiths et al, 2021; Lopacinski et al, 2021). For (B–G), n = 1489 heart samples. For (I, K), box-and-whisker plots show the median VP1 (horizontal line), mean VP1 (asterisk), interquartile range (IQR, box), and an additional 1.5 IQR extension (whiskers) from n = 4 biological replicates. Differences were assessed with a one-way ANOVA followed by Tukey's Honest Significant Difference post hoc test for DOX- and iCAR-associated increases. n.d. no difference.

iCAR line was then used at different times after doxycycline induction to infect with CVB3 for 6 h. At 30 min of induction, AC16-iCAR cells expressed 2850 ± 430 copies of CXADR per cell and showed no more synthesis of CVB3 viral protein 1 (VP1) than negative controls (Figs. 4H,I and EV4I,J). By contrast, after 60 min of induction, CXADR increased to 4780 ± 670 copies per AC16-iCAR cell and expression of mature VP1 in infected cells was significantly increased at 6 h (Figs. 4J,K and EV4K,L). These experiments validate the CVB3 permissibility threshold of the model and together with Pinferna, they raise the additional possibility that some human hearts are intrinsically resistant to CVB3 (Fig. 4G).

## Application to molecular subtyping

Transcriptomic profiles are widely used to define disease subtypes (Hoadley et al, 2018; Neff et al, 2021; Ramirez Flores et al, 2021), and functional interpretations may change when gene expression is replaced by inferred protein abundance. As a longstanding example, we selected the intrinsic molecular subtypes of breast cancer defined by a 50-gene classifier (PAM50) for 796 cases with RNA-seq in The Cancer Genome Atlas (Ciriello et al, 2015; Parker et al, 2009; Perou et al, 2000). For consistency, our analysis focused on the 4366 transcripts compatible with protein inference (Fig. 1A), but results were unchanged when using the entire available

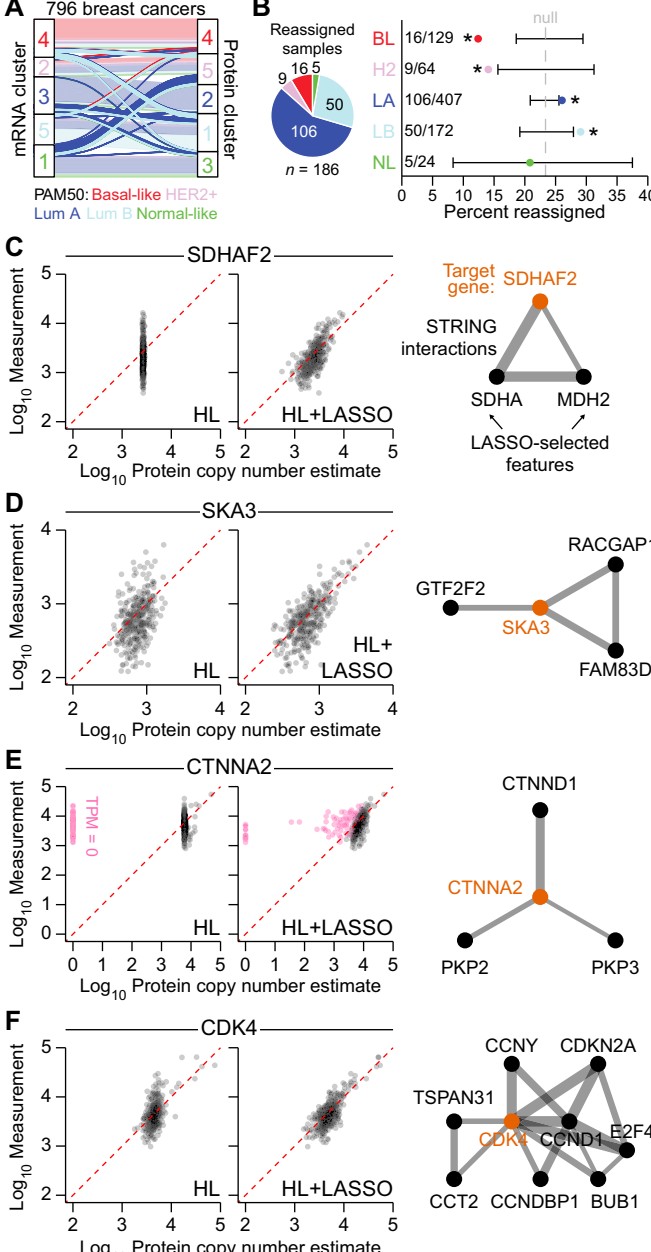

**Figure 5.  Inferred proteomics reassigns luminal A/B transcriptomic subtypes of breast cancer.**

(A) Reorganization of five consensus clusters defined by RNA-seq (left) and Pinferna (right) for breast cancers in The Cancer Genome Atlas (Ciriello et al, 2015). Clusters were determined by Monte Carlo consensus clustering (John et al, 2020) and colored according to the dominant PAM50 subtype of each cluster. Samples that did not change clusters are transparent in the background while samples that changed are opaque in the foreground. Lum A: Luminal A; Lum B: Luminal B. (B) Reassigned samples are predominated by luminal A/B PAM50 subtypes. (Left) Proportion of each subtype among samples that were reassigned. (Right) Percent reassignments for each subtype. The average overall reassignment rate is shown as a null reference (186/796 = 23%; gray dashed) with the 90% hypergeometric confidence interval (black) for each subtype. BL is Basal-like, H2 is HER2 +, LA is Luminal A, LB is Luminal B, and NL is Normal-like. (C–F) Cluster-reorganizing genes are highly dependent on other genes. (Left) Concordance between SWATH-scaled measurements and the HL fit ± LASSO in the meta-assembly. Perfect concordance is given by the red dashed line. Pink points in (E) are samples with TPM = 0 for *CTNNA2*. (Right) STRING interactions (edges) among the target gene (orange) and its LASSO-selected features (black). Edge thickness (gray) reflects the confidence of the interaction as determined by STRING. Thicker lines represent a higher confidence score. Line lengths are arbitrary. Data information: For (A), $n = 796$ breast cancers. For (B), reassignment enrichments were determined by hypergeometric test, *P = 2.2e-4 (BL), 1.9e-2 (H2), 4.1e-2 (LA), and 3.1e-2 (LB). For (C–F), $n = 369$ cell lines.

Among reassigned samples, we noted preferential enrichments in PAM50-classified Luminal A (26%) and Luminal B (29%) tumors (Fig. 5B). Interestingly, Luminal A-reassigned patients had a ~twofold worse overall survival than those that were not reassigned (Appendix Fig. S4A,B). Luminal A/B cases are often intermingled in clusters defined by transcriptomics (Cancer Genome Atlas, 2012), prompting us to look more deeply at their reassignment with Pinferna (Appendix Fig. S5A,B). We surveyed for HL + LASSO genes whose Z-score standardized values changed the most from mRNA to inferred protein and looked within these influential genes for features (other genes) that were STRING interactors ("Methods"; Fig. 2H; Appendix Fig. S5C). Calibration of the mRNA-to-protein relationship for SDHAF2 (a mitochondrial Complex II assembly factor) was dramatically improved with abundance information from other genes, including its interactors, SDHA and MDH2 (Fig. 5C). Similarly, inference of SKA3 (a subunit of the mitotic Ska complex) was influenced by multiple binding partners (Fig. 5D). One of the most notable examples of LASSO modulation was CTNNA2 (an adhesion protein involved in actin regulation). CTNNA2 protein was measured ubiquitously in the meta-assembly, but its mRNA was undetectable in 23% of training samples. In 83% of these, nonzero protein inference was recovered by using abundances including CTNND1 and plakophilin (PKP2, PKP3) interactors (Fig. 5E). For cluster reassignments of breast cancer, the gene with the most interaction-rich feature set was CDK4. CDK4 protein abundance was largely independent of its own mRNA, but a useful calibration was achieved when considering various cyclins and other binding proteins (Fig. 5F). This result is important because CDK4/6 inhibitors are approved to treat luminal breast cancers (Finn et al, 2016; Slamon et al, 2020), but responsiveness has not consistently associated with the abundance of *CDK4* mRNA (Finn et al, 2020; Turner et al, 2019). Newly meaningful subtypes with therapeutic implications might arise when combining transcriptomics with Pinferna to get closer to the functional proteome.

transcriptome (Appendix Fig. S3A–D). Consensus clustering of mRNA profiles identified five ordered and stable groups, which were statistically enriched in PAM50-assigned cases of (1) Normal-like ($P = 3.8e-2$), (2) HER2+ ($P = 2.7e-3$), (3) Luminal A ($P = 3.4e-39$), (4) Basal-like ($P = 1.1e-42$), and (5) Luminal B breast cancer ($P = 1.6e-23$; Fig. 5A; Appendix Fig. S3A,B). When the analysis was repeated with Pinferna estimates after standardization, the smallest number of stable and significant consensus clusters was again five (Appendix Fig. S3E,F). However, the enriched PAM50 assignments were reordered, and 186/796 = 23% of cases changed to a different cluster (Fig. 5A). The aggregate transformations of Pinferna (Fig. 1C–E) thus exceeded a standardized rescaling and considerably altered subgroup composition.

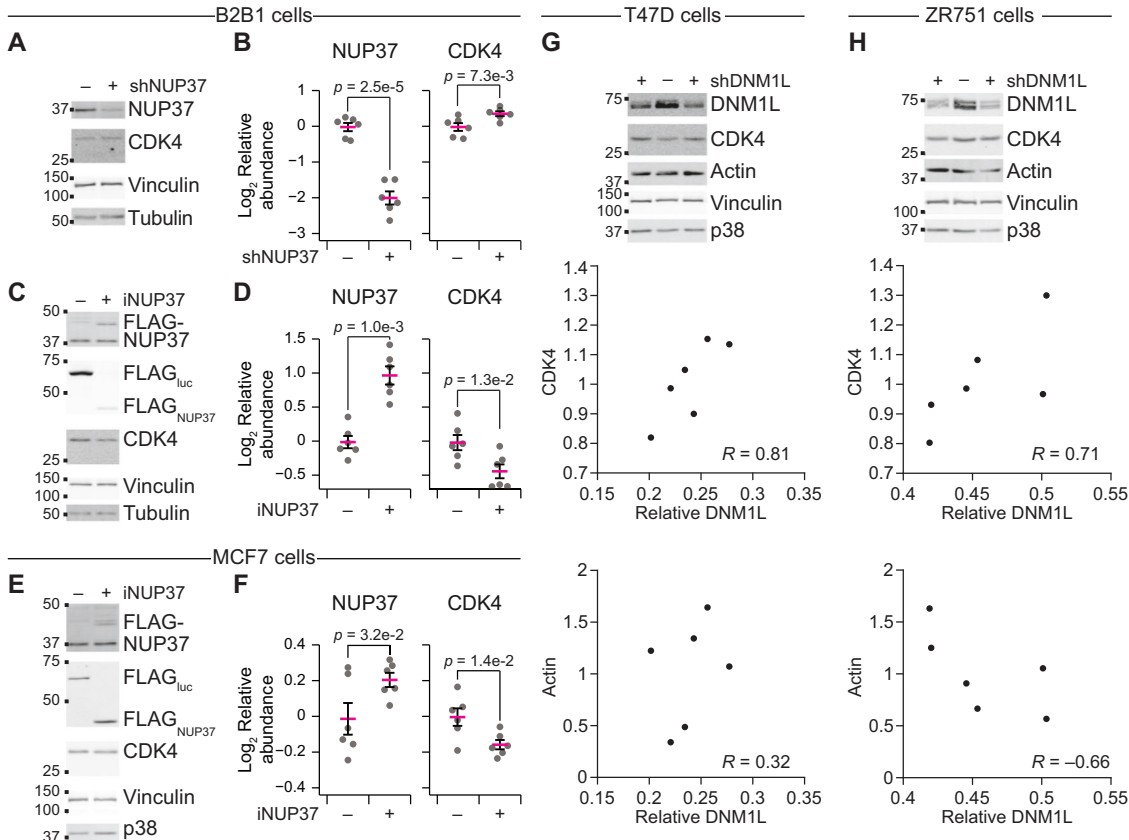

**Figure 6. LASSO features of CDK4 include causal dependencies.**

(A, B) Knockdown of NUP37 increases CDK4 abundance in B2B1 breast epithelial cells (Wang et al, 2023). (C, D) Overexpression of inducible NUP37 (iNUP37) decreases CDK4 abundance in B2B1 breast epithelial cells. (E, F) Overexpression of iNUP37 decreases CDK4 abundance in MCF7 luminal breast cancer cells. (G, H) Residual DNM1L protein after knockdown correlates with CDK4 abundance in T47D (G) and ZR751 (H) luminal breast cancer cells. Actin was used as a housekeeping control not used for loading normalization ("Methods"). Quantities are scaled to shScramble control samples. Data information: For (A, C, E), representative immunoblots for NUP37 and CDK4 are shown with vinculin and tubulin (A, C) or p38 (E) as loading controls and FLAG (C, E) to confirm ectopic expression of iNUP37 or luciferase (luc) control. For (B, D, F), immunoblot results are summarized as the mean total NUP37 (endogenous (B, D, F) and induced (D, F)) or CDK4 ± s.e.m. of $n = 6$ biological replicates. Differences were assessed by one-sided $t$ test. For (G, H), representative immunoblots for DNM1L and CDK4 are shown with vinculin and p38 as loading controls and actin as a housekeeping control. Data are summaries from $n = 6$ biological replicates.

## Test of HL+LASSO-derived biological hypotheses

The LASSO-selected features of Pinferna contain many known interactions (Figs. 2H and 5C–F), raising the possibility that they might also suggest new gene-gene dependencies. We focused on the CDK4 feature set and selected NUP37 (a subunit of the nuclear pore) and DNM1L (a mitochondrial dynamin-like GTPase) for further study. Among 73 features for CDK4, NUP37 was the ninth most negative dependency and DNM1L the fifth most positive (Dataset EV4). We began with a breast epithelial clone (B2B1) that had previously been engineered for inducible overexpression or knockdown of NUP37 (Wang et al, 2023). Compared to induced nontargeting shRNA control, acute NUP37 knockdown caused a detectable increase in CDK4 abundance (Fig. 6A,B). Likewise, inducible NUP37 (iNUP37) overexpression led to a decrease in CDK4 relative to overexpression control (Fig. 6C,D), consistent with the inverse LASSO dependency encoded by Pinferna. We successfully extended the iNUP37 result in B2B1 cells to a luminal breast cancer cell line (MCF7; Fig. 6E,F). Unfortunately, NUP37 knockdown was not effective in MCF7 cells or another luminal breast cancer line (T47D; Appendix Fig. S6),

preventing further interpretations about CDK4. For DNM1L, we pursued similar gain- and loss-of-function approaches but were repeatedly stymied by dosage compensation from the induced DNM1L allele (Appendix Fig. S7). We nevertheless managed to achieve two- to fivefold reductions in DNM1L protein by constitutive knockdown with a validated shRNA (Kashatus et al, 2015; Nagdas et al, 2019). In two luminal breast cancer lines (T47D and ZR751) with DNM1L knockdown, we found that variations in the residual DNM1L protein among replicates correlated positively with the abundance of CDK4 but not a housekeeping control (Fig. 6G,H), as predicted by the positive LASSO dependency. Certainly, not all LASSO-selected features in Pinferna reflect cause-effect relationships, but the NUP37–DNM1L results for CDK4 suggest that some may merit further study.

## Discussion

We have devised a straightforward gene-by-gene formalism that uses mRNA to achieve proportional protein estimates informed by

the best measurements available for each data type. Gene-specific inferences are gleaned from cancer cell lines, but general accuracy and utility is verified in multiple other contexts. Although not exhaustive, our coverage of 4366 mRNA–protein relationships is considerable given that modern SWATH experiments reliably quantify ~5000 proteins (Liu et al, 2019). Encouraged by the robustness of predictions to read depth and alignment details, we provide Pinferna as an open resource (http://janeslab.shinyapps.io/Pinferna; Dataset EV8). The platform is optimal when provided full RNA-seq profiles, but it also accommodates single-gene TPM entries for LASSO-free inference when subset data are exported from public repositories. Besides the 4366 SWATH-calibrated genes, the site contains TMT-only models for an additional 4898 genes that can be rescaled if a proportional calibrator is provided for one or more CCLE samples. The hope is to attract general users to the value in seeing genes and gene profiles of interest through the lens of inferred proteomics (source code is available for developers; "Data availability").

The mRNA-to-protein relationship classes used here distill the major findings of prior models that were finer grained (Eraslan et al, 2019; Schwanhausser et al, 2011; Vogel et al, 2010). Measured mRNA is the net result of its transcription–degradation, with the per-mRNA yield of translation for each gene being the greatest determinant of protein abundance (Schwanhausser et al, 2011; Wilhelm et al, 2014). Together, mRNA abundance and per-mRNA yield define an expected set point for protein abundance as captured by the HL relationship class. For typical in vitro cultures, cell doubling is faster than turnover of most proteins (Buccitelli and Selbach, 2020; Schwanhausser et al, 2011), creating a perpetual state of halving-and-recovery that likely explains why many HL models are log convex. Nonetheless, overall accuracy of Pinferna did not decrease with clinical samples that were less proliferative, suggesting a role for other cyclic perturbations in vivo, such as circadian rhythms (Zhang et al, 2014b). Some genes additionally require protein complexes to persist stably (Buccitelli and Selbach, 2020; Taggart et al, 2020), which creates buffering dependencies on other genes that are coexpressed. The HL + LASSO approach seeks to capture this relationship class by identifying statistical mRNA–protein associations in *trans*. Despite its heavy L1 regularization (Lever et al, 2016), LASSO recovered a significant number of documented protein-protein interactions. Recently, a reciprocal approach to predict relative protein abundance from mRNA was proposed that constrains the search space for each gene to its CORUM–STRING interactors but relaxes the regularization by using elastic net (Lever et al, 2016; Srivastava et al, 2022). These models retain many more features: 158–457 (Srivastava et al, 2022) compared to 1–83 features for HL + LASSO. Future versions may consider hybrid regularizations that penalize CORUM–STRING interactors less during LASSO feature selection. Iterative approaches might also use Pinferna inferences as LASSO features for other genes to approximate biological dependencies more closely. Lastly, we speculate that M genes arise from protein complexes so large that pairwise interactions within a complex completely dictate abundance (Haken, 1987; Schneidman et al, 2006).

Proportional protein estimates are of great practical use for making qualitative determinations in systems-biology models. For example, using inferred protein to simulate heart infections, we clarified that individuals with fewer than ~5000 copies of CXADR per cell were not susceptible to CVB3 (Fig. 4G). Another subfield of relevance is genome-scale metabolic modeling with tailored

derivatives of the generic human metabolic network reconstruction (Brunk et al, 2018). For cell- or tissue-specific modeling, the generic reconstruction is pruned according to which metabolic genes are "not expressed" in a biological context of interest (Agren et al, 2012; Becker and Palsson, 2008). Irrespective of the pruning algorithm, the choice of threshold is made uniformly across all genes in a sample, which defines the resulting model complexity (Opdam et al, 2017). Protein abundances for some metabolic pathways scale linearly with mRNA, but others do not (Mertins et al, 2016; Zhang et al, 2014a), and protein–mRNA set points vary over several orders of magnitude (Eraslan et al, 2019; Wilhelm et al, 2014). For metabolic models concerned with protein fidelity, estimating copy numbers from RNA-seq is a scalable alternative to proteome immunohistochemistry (Agren et al, 2012).

There are limitations in our approach to proportional copy-number estimation. By relying on SWATH for calibration, we lose many of the 9000+ proteins quantified in relative terms by TMT (Nusinow et al, 2020). Calibration data were all collected at steady state; thus, we caution against using RNA-seq obtained shortly after acute perturbations when transcripts and proteins will be most uncoupled (Buccitelli and Selbach, 2020). Signaling proteins rapidly turned over by ubiquitylation (TP53, NFE2L2, NFKBIA) might require other formalisms when they become detectable by SWATH. Broadening Pinferna predictions to non-human samples awaits the availability of robust SWATH libraries in other mammals (Zhong et al, 2020). Last, we recall that no total-protein estimator captures functional state, such as the surface localization of CD40, the tyrosine phosphorylation of PKP2–PKP3, or the kinase activity of CDK4 (Figs. 2C and 5E,F). Despite these caveats, the method adds an immediately useful approach for systems biology that compares favorably against existing alternatives.

RNA-seq delivers more than transcript counts when combined with specialized analytical methods (Gao et al, 2021; La Manno et al, 2018; Newman et al, 2019). Pinferna illustrates how transcriptomics can extend the reach of proteomics in a data-driven way by informing more biological samples, including retrospective ones it would never otherwise have access to.

# Methods

## Methods and protocols

### SWATH alignment and quantification

Raw SWATH data files were obtained from the PRIDE repository (Perez-Riverol et al, 2019) (CAL51, PXD003278 (Liu et al, 2017); U2OS, PXD000954 (Rosenberger et al, 2014); HeLa, PXD00927 (Liu et al, 2019)) and converted to .mzML format using the MSConvertGUI (version 3.0) in the ProteoWizard software suite (Chambers et al, 2012) with the following options: Output format, mzML; Extension, mzML; Binary encoding precision, 64-bit; Write index; Use zlib compression. Peptide fragments were aligned with OpenSwathWorkflow in OpenMS (version 2.4.0) (Rost et al, 2016b) with the following options: -sort_swath_maps, -readOptions normal, -batchSize 1000, -use_ms1_traces, -mz_correction_function quadratic_regression_delta_ppm. Statistical control was performed with PyProphet (version 2.2.5) (Teleman et al, 2015) with the following options: --group_id=transition_group_id, --tric_chromprob.

**Reagents and tools table**

| Reagent/resource | Reference or source | Identifier or catalog number |
|---|---|---|
| **Experimental models** | | |
| AC16 cells (*H. sapiens*) | Davidson et al, 2005 | – |
| AC16-CXADR cells (*H. sapiens*) | Shah et al, 2017 | – |
| B2B1-iLuc cells (*H. sapiens*) | Wang et al, 2023 | – |
| B2B1-iNUP37 cells (*H. sapiens*) | Wang et al, 2023 | – |
| B2B1-ishLuc cells (*H. sapiens*) | Wang et al, 2023 | – |
| B2B1-ishNUP37 cells (*H. sapiens*) | Wang et al, 2023 | – |
| BT20 (*H. sapiens*) | ATCC | Cat # HTB-19 |
| CAMA1 cells (*H. sapiens*) | ATCC | Cat # HTB-21 |
| Coxsackievirus B3, Kandolf strain | Laboratory of Bruce McManus | Genbank # M33854 |
| HCC1395 (*H. sapiens*) | ATCC | Cat # CRL-2324 |
| HCC38 (*H. sapiens*) | ATCC | Cat # CRL-2314 |
| HEK293T/17 cells (*H. sapiens*) | ATCC | Cat # CRL-11268 |
| K562 (*H. sapiens*) | ATCC | Cat # CCL-243 |
| MCF7 cells (*H. sapiens*) | ATCC | Cat # HTB-22 |
| T47D cells (*H. sapiens*) | ATCC | Cat # HTB-133 |
| ZR751 cells (*H. sapiens*) | ATCC | Cat # CRL-1500 |
| **Recombinant DNA** | | |
| pEN_TTmiRc2 | Addgene | #25752 |
| pEN_TT 3xFLAG-NUP37 (*H. sapiens*) | Addgene | #192299 |
| pEN_TT CXADR-V5 (*H. sapiens*) | Addgene (this study) | #220358 |
| pEN_TT DNM1L-V5 (*H. sapiens*) | Addgene (this study) | #220359 |
| pEN_TT luciferase-V5 (*P. pyralis*) | Addgene (this study) | #220360 |
| pLX304 CXADR-V5 (*H. sapiens*) | Addgene | #82723 |
| pSLIK 3xFLAG-luciferase neo (*P. pyralis*) | Addgene | #98392 |
| pSLIK 3xFLAG-NUP37 neo (*H. sapiens*) | Addgene (this study) | #220361 |
| pSLIK CXADR-V5 hygro (*H. sapiens*) | Addgene (this study) | #220362 |
| pSLIK DNM1L-V5 neo (*H. sapiens*) | Addgene (this study) | #220363 |
| pSLIK hygro | Addgene | #25737 |
| pSLIK luciferase-V5 neo (*P. pyralis*) | Addgene (this study) | #220364 |
| pSLIK neo | Addgene | #25735 |
| pSuperior.retro.neo GFP shDNM1L (*H. sapiens*) | Addgene | #220356 |
| pSuperior.retro.neo GFP shScramble | Addgene | #220357 |
| tet pLKO.1-shLuc puro (*P. pyralis*) | Addgene | #136587 |
| tet pLKO.1-shNUP37 v1 puro (*H. sapiens*) | Addgene | #192343 |
| **Antibodies** | | |
| Beta-actin (AC-15) Mouse mAb | Invitrogen | Cat # AM4302 |
| CAR (D3W3G) Rabbit mAb | Cell Signaling Technology | Cat # 16984 |
| CDK4 (D9G3E) Rabbit mAb | Cell Signaling Technology | Cat # 12790 |
| DRP1 (EPR19274) Recombinant Rabbit mAb | Abcam | Cat # ab184247 |
| FLAG (M2) Mouse mAb | Sigma | Cat # F1804 |
| GRB2 (C-23) Rabbit Polyclonal Ab | Santa Cruz Biotechnology | Cat # sc-255 |
| NUP37 (N-13) Rabbit Polyclonal Ab | Santa Cruz Biotechnology | Cat # sc-109348 |
| p38 (C-20) Rabbit Polyclonal Ab | Santa Cruz Biotechnology | Cat # sc-535 |

| Reagent/resource | Reference or source | Identifier or catalog number |
|---|---|---|
| α-Tubulin Chicken Polyclonal Ab | Abcam | Cat # ab89984 |
| α-Tubulin Chicken Polyclonal Ab | LSBio | Cat # LS-C108486 |
| SERPINB6 Rabbit Polyclonal Ab | Proteintech | Cat # 14962-1-AP |
| Vinculin (V284) Mouse mAb | Millipore Sigma | Cat # 05-386 |
| V5 Chicken Polyclonal Ab | Bethyl Laboratories | Cat # A190-118A |
| V5 (SV5-Pk1) Mouse mAb | Invitrogen | Cat # R960-25 |
| VP1 (31A2) Mouse mAb | Mediagnost | Cat # M47 |
| Goat anti-rabbit | Jackson ImmunoResearch | Cat # 111-005-144 |
| Goat anti-mouse | Jackson ImmunoResearch | Cat # 115-005-146 |
| IRDye 800CW-conjugated donkey anti-goat | LI-COR | Cat # 926-32214 |
| IRDye 680LT-conjugated donkey anti-chicken | LI-COR | Cat # 926-68028 |
| IRDye 800CW-conjugated goat anti-rabbit | LI-COR | Cat # 926-32211 |
| IRDye 680LT-conjugated goat anti-rabbit | LI-COR | Cat # 926-68021 |
| IRDye 800CW-conjugated goat anti-mouse | LI-COR | Cat # 926-32210 |
| IRDye 680LT-conjugated goat anti-mouse | LI-COR | Cat # 926-68020 |
| **Oligonucleotides and other sequence-based reagents** | | |
| CXADR forward cloning primer | This study | gcgcACTAGTatggcgctcctgctgtgc |
| CXADR reverse cloning primer | This study | gcgcCAATTGctacgtagaatcgagaccgaggag |
| DNM1L forward cloning primer | This study | gcgcACTAGTatggaggcgctaattcctgt |
| DNM1L reverse cloning primer | This study | gcgcGAATTCTCACGTAGAATCGAGACCGAGGAGAG GGTTAGGGATAGGCTTACCccaaagatgagtctcccggatt |
| Luciferase forward cloning primer | This study | gcgcACTAGTatggaagacgccaaaaacat |
| Luciferase reverse cloning primer | This study | gcgcGAATTCTCACGTAGAATCGAGACCGAGGAGA GGGTTAGGGATAGGCTTACCgatctttccgcccttcttgg |
| **Chemicals, enzymes and other reagents** | | |
| LR Clonase II | ThermoFisher Scientific | Cat # 11791020 |
| Doxycycline Hyclate | Sigma | Cat # D9891 |
| **Software** | | |
| biomaRt v2.52.0 | https://www.bioconductor.org/packages/release/bioc/html/biomaRt.html | |
| ComplexHeatmap v2.12.1 | https://bioconductor.org/packages/release/bioc/html/ComplexHeatmap.html | |
| CPTAC breast CNA | syn11328692 | |
| CPTAC breast RNA-seq | syn11328694 | |
| DescTools v0.99.47 | https://cran.r-project.org/web/packages/DescTools/index.html | |
| ea-utils | https://github.com/ExpressionAnalysis/ea-utils | |
| ggplot2 v3.4.0 | https://ggplot2.tidyverse.org/ | |
| glmnet v4.1-6 | https://cran.r-project.org/web/packages/glmnet/index.html | |
| HISAT2 v2.1.0 | http://daehwankimlab.github.io/hisat2/ | |
| HTSeq v2.0.2 | https://pypi.org/project/HTSeq/ | |
| inflection v1.3.6 | https://cran.r-project.org/web/packages/inflection/index.html | |
| M3C v1.18.0 | https://www.bioconductor.org/packages/release/bioc/html/M3C.html | |
| MATLAB R2022a | https://www.mathworks.com/products/matlab.html | |

| Reagent/resource | Reference or source | Identifier or catalog number |
|---|---|---|
| MATLAB R2022b | https://www.mathworks.com/products/matlab.html | |
| Mclust v6.0.0 | https://cran.r-project.org/web/packages/mclust/index.html | |
| MSConverGUI v3.0 | https://proteowizard.sourceforge.io/download.html | |
| msproteomicstools v0.11.0 | https://pypi.org/project/msproteomicstools/ | |
| numpy v1.26.1 | https://numpy.org/ | |
| OpenMS v2.4.0 | https://openms.de/ | |
| pandas v2.1.2 | https://pandas.pydata.org/ | |
| Proteo-estimator v1.0.5 | https://pypi.org/project/proteo-estimator/ | |
| PyProphet v2.2.5 | https://pypi.org/project/pyprophet/0.23.0/ | |
| Python v3.9.18 | https://www.python.org/ | |
| R v4.2.1 | https://www.r-project.org/ | |
| rbioapi v0.7.7 | https://cran.r-project.org/web/packages/rbioapi/index.html | |
| samtools v1.12 | http://www.htslib.org/ | |
| sratoolkit v2.10.5 | https://anaconda.org/bioconda/sra-tools | |
| STRINGdb v2.8.4 | https://www.bioconductor.org/packages/release/bioc/html/STRINGdb.html | |
| StringTie v2.1.0 | https://ccb.jhu.edu/software/stringtie/ | |
| survival v3.4-0 | https://cran.r-project.org/web/packages/survival/index.html | |
| survminer v0.4.9 | https://cran.r-project.org/web/packages/survminer/index.html | |
| **Other** | | |
| BCA Protein Assay Kit | ThermoFisher Scientific | Cat # 23227 |

Each series of SWATH runs was realigned with TRIC (Rost et al, 2016a) using msproteomicstools (version 0.11.0) with the following options: --method LocalMST, --realign_method lowess, --max_rt_diff 60, --mst:useRTCorrection True, --mst:Stdev_multiplier 3.0, --target_fdr 0.01, --max_fdr_quality 0.05, --alignment_score 0.0005. The top three peptide fragments by intensity (or all peptide fragments if fewer than three) were summed for each protein to estimate relative abundance. Summed intensities were mean averaged across technical replicates when available. To place summed intensities on a common proportional scale, the median abundance of all detected proteins within each sample was centered at 10,000 protein copies per cell (Liu et al, 2019).

### RNA-seq alignment and quantification

For all studies other than Fig. 4, SRA files were obtained from the Sequence Read Archive (SRA) (Leinonen et al, 2011) (HeLa, PRJNA437150; NCI-60, PRJNA433861; Prostate, PRJNA579899) and converted to raw FASTQ files using sratoolkit (version 2.10.5) with fasterq_dump. TruSeq adapters were trimmed using the fastq-mcf function in the ea-utils package with the following options: -q 10, -t 0.01, -k 0. Trimmed datasets were aligned to the human genome (GRCh38) using HISAT2 (version 2.1.0) (Kim et al, 2019) with the following options: --dta (downstream transcriptome assembly) and either --rna-strandedness RF (for paired-end reads generated by the TruSeq strand-specific library; NCI-60 and prostate samples) or --rg-id (for single-end reads generated by

the TruSeq library; HeLa). Output SAM files were converted to BAM files using the sort function in samtools (version 1.12) (Li et al, 2009), and BAM files were indexed to create BAI files using the index function for obtaining counts downstream. Alignments were assembled into transcripts using StringTie (version 2.1.0) (Kovaka et al, 2019) with the -e option restricting assembly to known transcripts in the provided annotation. Counts were obtained using HTSeq (version 2.0.2) (Putri et al, 2022) using BAM files as the input with the following options: -f bam, -r pos, -m intersection-strict, -s reverse, -a 1, -t exon, -i gene_id. Heart RNA-seq data were obtained from dbGaP (GTEx, phs000424.v9.p2), the Sequence Read Archive (MAGNet, SRP237337), and the European Genome-Phenome Archive (EGA, EGAS00001002454). Raw reads from all samples were extracted, aligned, assembled, and counted as described elsewhere (Griffiths et al, 2024).

### Data harmonization

The table of MANE Select identifiers was obtained from the source publication (Morales et al, 2022) and filtered for "MANE Select" genes. The filtered table was appended with UniProt accession codes using biomaRt (version 2.52.0) and GRCh38. The Ensembl BioMart browser was used to obtain HGNC identifiers, Ensembl transcript identifiers (with version numbers for maximum overlap), RefSeq mRNA identifiers, NCBI (formerly Entrez) gene identifiers,

UniProt accession codes, and UniProt gene symbols for *Homo sapiens*. Each row of the MANE Select table was matched to at least two identifiers in the biomaRt table to determine the UniProt accession numbers. When MANE Select annotated a gene symbol as LOC###### and biomaRt contained a more descriptive gene symbol, the biomaRt gene symbol replaced the MANE Select gene symbol and the "Database" column was updated to include "biomaRt symbol" as the source. The harmonization identified 83 genes that are not currently available in UniProt. The final harmonized table of ten identifiers (nine for the 83 genes not in UniProt) for 19,062 genes is available in Dataset EV3.

### Cancer Cell Line Encyclopedia pre-processing

The TMT proteomic dataset was obtained from the source publication (Nusinow et al, 2020) as a CSV file (protein_quant_-current_normalized.csv). After removing proteins annotated as "Fragments", gene symbols were matched to UniProt accession codes by using the harmonized identifier table (Dataset EV3). Protein isoforms with redundant gene symbols were summed. The RNA-seq dataset (Ghandi et al, 2019) was obtained from the Depmap portal, back-transformed from $\log_2$ to TPM, and renamed with the harmonized identifier table (Dataset EV3).

### Meta-assembly, calibration, and inference

**Scaling**: For each gene, SWATH copy-number estimates were divided by the corresponding harmonized TMT data for U2OS and CAL51 cells to calculate U2OS- and CAL51-specific scaling factors. Scaling factors were averaged when possible; otherwise, a single scaling factor was used (Fig. EV1A). The resultant scaling factors were then multiplied across the harmonized TMT data table to yield a SWATH-scaled proteomics dataset of 4384 total proteins across 375 cell lines.

**Regression**: The SWATH-scaled proteomics and RNA-seq transcriptomics datasets were filtered before regression. As recommended (Nusinow and Gygi, 2020), proteomics data from replicates of CAL120 (CAL120_BREAST_TenPx02), SW948 (SW948_LARGE_INTESTINE_TenPx11), and HCT15 (HCT15_LARGE_INTESTINE_TenPx30) were excluded. RNA-seq data were filtered to include only cell lines with SWATH-scaled proteomics available. Datasets were filtered to retain genes for which SWATH-scaled proteomics was available in at least 150 cell lines ($4445/4513 = 98.5\%$ of all SWATH-scaled proteins). Data for regressions are available in Datasets EV1 and EV2.

1. M regression. For each gene, the median was calculated across the SWATH-scaled protein copies per cell (SPC), taking empty entries as missing elements rather than zeros. A 95% confidence interval of the median was estimated by bootstrapping ($n = 1000$ runs).
2. HL regression. For each gene, an HL model was constructed as follows:

$$\text{SPC} = a \bullet \left( \frac{b \bullet TPM}{c + TPM} + TPM \right); \; c > 0,$$

where $a$, $b$, and $c$ were regression coefficients estimated by nonlinear least squares in MATLAB (version R2022a). To prevent discontinuities from division by zero, $c$ was constrained to be greater than

zero. Additionally, a logistic weighting of the cost function was desired to prevent high-abundance cell lines from overleveraging the regression. To achieve both, we devised a two-step procedure in which initial estimates were made with a linear cost function and $c > 0$ constraint using lsqcurvefit with 'FiniteDifferenceType' set to 'Central'. The regression estimates of $a$, $b$, and $c$ were then used as an initial guess for nlinfit with 'RobustWgtFun' set to 'logistic', and the regression was repeated without constraints. If the updated value of $c$ was less than zero, then the updated regression estimates of $a$ and $b$ were used as initial guesses for a second round of lsqcurvefit with a linear cost function and $c > 0$ constraint. A 95% confidence interval of the HL fit was estimated by asymptotic error analysis of the regression coefficients using the $F$ distribution to describe the ratio of the sum-of-squared errors for the ideal and parameter-perturbed model divided by their corresponding degrees of freedom.

3. HL + LASSO regression. Residuals from the HL fit were regressed against all other coding genes in the MANE-harmonized transcriptome (Dataset EV3) by LASSO with the glmnet package (version 4.1-6) in R. To determine the optimal penalty strength parameter ($\lambda$) for each LASSO regression, we used the cv.glmnet function for cross-validation after increasing the function's minimum fractional change in deviance for stopping (fdev) to 0.01. We accounted for differences among cross-validation runs by iterating cv.glmnet 100 times, calculating the BIC for the best $\lambda$ in each iteration, and defining the best $\lambda$ with the lowest BIC as optimal. The optimal $\lambda$ was used with the glmnet function and all observations of the gene to obtain a regularized feature set and linear coefficients. Output of the LASSO regression was added to the HL fit to obtain the HL + LASSO model. LASSO displacements were propagated linearly to the 95% confidence interval of the HL fit, and graphical displays were LOESS smoothed with the geom_smooth function in ggplot2 (version 3.4.0) in R.

**Model selection**: The BIC for each regression was calculated under the assumption of normally distributed random errors as follows:

$$BIC = p \bullet \log(n) - 2 \bullet \log\left( \sum_{i=1}^{n} \frac{1}{\sqrt{2\pi}\sigma} e^{-(y_i - \hat{y}_i)^2 / 2\sigma^2} \right)$$

where

$$\sigma = \sqrt{\frac{\sum_{i=1}^{n} (\hat{y}_i - y_i)^2}{n}}$$

Where $n$ is the number of observations, $p$ is the number of model parameters, $\hat{y}_i$ is the predicted value of the $i^{\text{th}}$ observation, $y_i$ is the measured value of the $i^{\text{th}}$ observation, and *log* is the natural logarithm. For comparison of HL with linear, hyperbolic, and 3-parameter logistic (3PL) alternatives, three alternative models were constructed as follows:

$$\text{Linear}: \text{SPC} = a \bullet TPM$$

$$\text{Hyperbolic}: \text{SPC} = \frac{a \bullet TPM}{b + TPM}; \; b > 0$$

$$3PL : SPC = a - \frac{a}{1 + \left(\frac{TPM}{b}\right)^c} \, ; \, a > 0, \, b > 0$$

where $a$, $b$, and $c$ are regression coefficients that were estimated in MATLAB (version R2022a). Linear models were regressed using fitlm with 'RobustOpts' set to 'logistic' and 'Intercept' set to false. Hyperbolic and 3PL models were fit similarly to HL but with a combination of lsqcurvefit for constrained regression and fitnlm (which calls nlinfit internally) for logistic weighting of the cost function and the estimation of log-likelihood, which was used for BIC calculation with the aicbic function. BIC weights (BICw) were calculated as follows:

$$BICw_i = \frac{\exp\left[-\frac{1}{2}(BIC_i - BIC_{\min})\right]}{\sum_{k=1}^{K} \exp\left[-\frac{1}{2}(BIC_k - BIC_{\min})\right]}$$

where $BIC_i$ is the BIC for the $i$th model and $BIC_{min}$ is the minimum BIC in the group of models (Wagenmakers and Farrell, 2004).

### Gene ontology analysis

Enrichments of M, HL, and HL + LASSO genes for biological processes were evaluated with the GO knowledgebase (Gene Ontology et al, 2023). Genes in each class (Dataset EV4) were separately tested on the GO Enrichment Analysis web page (https://geneontology.org/) and searched for "Biological Process" at a false-discovery rate of 5%. The complete list of enrichments is available in Dataset EV5.

### Concavity analysis

The concavity of HL fits was assessed with the check_curve function of the inflection package (version 1.3.6) in R. HL curves for TPM > 5 (~1 copy per cell) were analyzed after log transformation of $x$ and $y$ coordinates.

### Quantitative immunoblotting

Relative and absolute protein abundances were estimated by quantitative immunoblotting as previously described (Janes, 2015) with antibodies recognizing the following protein epitopes: β-actin (Invitrogen, AM4302, 1:5000), CAR (Cell Signaling Technology, 16984, 1:1000), CDK4 (Cell Signaling Technology, 12790, 1:1000), DNM1L (Abcam, ab184247, 1:1000), FLAG (Sigma, F1804, 1:5000), GRB2 (Santa Cruz, sc-255, 1:1000), NUP37 (Santa Cruz, sc-109348, 1:1000), p38 (Santa Cruz, sc-535, 1:5000), SERPINB6 (Proteintech, 14962-1-AP, 1:2000), tubulin (Abcam, ab89984, 1:20,000 and LSBio, LS-C108486, 1:5000), V5 (Bethyl Laboratories, A190-118A, 1:5000), vinculin (Millipore Sigma, 05-386, 1:10,000), and VP1 (Mediagnost, M47, 1:2000). For absolute quantification of CXADR, AC16-CAR extracts with 5.5 ± 0.3 million CXADR per cell were used (Lopacinski et al, 2021; Shah et al, 2017), and a tertiary detection scheme was used with unlabeled goat anti-rabbit (Jackson ImmunoResearch, 111-005-144, 1:1000) for 1 h followed by IRDye 800CW-conjugated donkey anti-goat (LI-COR, 926-32214, 1:20,000). For ZR751 cells (Appendix Fig. S7A), a tertiary detection scheme was also used to detect V5 (Invitrogen, R960-25, 1:1000), with unlabeled goat anti-mouse (Jackson ImmunoResearch, 115-005-146, 1:1000) for 1 h followed by IRDye 800CW-conjugated donkey anti-goat (LI-COR, 926-32214,

1:20,000). Before normalizing to control conditions, cell types were normalized to the following loading controls: vinculin–tubulin (B2B1, BT20, CAMA1 (Fig. 2E), HCC38, HCC1395, K562, T47D (Fig. 2E), and ZR751 (Fig. 2E; Appendix Fig. 7A,B)), vinculin–p38 (CAMA1 (Appendix Fig. 7C,D) and T47D (Appendix Fig. 7C,E)), vinculin–GRB2 (AC16 (Figs. 4I,K and EV4J, L)), vinculin–tubulin–p38–GRB2 (T47D (Fig. 6G) and ZR751 (Fig. 6H)), and no loading normalization (AC16 (Figs. 4H,J and EV4I,K) and MCF7).

### Feature weight distributions

To obtain LASSO feature weights, each LASSO coefficient for a gene was multiplied by the mean TPM of the feature averaged across the 369 cell lines in the meta-assembly (including zeros). Feature weights for all HL + LASSO genes were concatenated and filtered for subunits of the proteasome (PSM-prefixed gene names) or ribosome (RPS- or RPL-prefixed gene names), allowing duplicates if the feature appeared in more than one gene. Distributions were plotted as smoothed densities with the geom_density function in ggplot2 (version 3.4.0).

### STRING interactions and maps

STRING interactions were obtained with the STRINGdb (version 2.8.4) and rbioapi (version 0.7.7) packages in R. Sessions were initialized with STRINGdb$new and the following arguments: species = 9606 (Homo sapiens), version = 11.5, score_threshold = 400 (medium-confidence interactions). HL + LASSO genes were mapped with the string_db$map function, and up to 1000 medium-confidence interactions were retrieved with the rba_string_interaction_partners function and harmonized with the gene identifier table (Dataset EV3). For comparison, LASSO-selected features were substituted with an identical number of randomly selected genes to recalculate the number of interactions. The substitution–recalculation step was iterated 10,000 times to build a null distribution.

### STRING visualization

Interaction maps were drafted on the STRING database web site (Szklarczyk et al, 2023) under Search>Multiple proteins. The default output maps were altered as follows: meaning of network edges = confidence, minimum required interaction score = medium confidence (0.400), disable 3D bubble design, and disable structure previews inside network bubbles. Interaction maps were exported as vectorized SVG files for further stylistic refinement.

### Assembly of test datasets

**HeLa**: After using the latest analytical procedures, SWATH data from two HeLa derivatives (Kyoto L8 and CCL2 L13) in the original study (Liu et al, 2019) did not pass the internal calibration step of the OpenSwathWorkflow alignment and were omitted here. Pre-quantified SWATH data for HeLa derivatives were downloaded from https://helaprot.shinyapps.io/crosslab/ and normalized to the median copy number of proteins co-quantified in the meta-assembly (10,000 copies per cell). Before HeLa RNA-seq down-sampling (Fig. EV3E), RNA-seq data from the HeLa derivatives were averaged. Raw counts were converted to counts per million (CPM), and the counts per million-normalized reads for each gene were averaged across all derivatives. Then, the average CPM was converted back to an averaged count by multiplying the average read depth and rounding to the nearest integer. The average counts

for each gene were downsampled 100 times using rbinom in R, with the number of trials equal to the downsampled read depth (25 million to 50,000) and the probability of success equal to the number of average counts for that gene divided by the total number of average counts. Downsampled counts were converted to TPM and used with Pinferna to predict the mean-averaged SWATH data from the HeLa derivatives.

**NCI-60**: Pre-aligned SWATH data for the NCI-60 panel of cell lines (Guo et al, 2019) were downloaded from CellMiner as a processed dataset (Protein: SWATH (Mass spectrometry)—Peptide), quantified for protein, harmonized as described above, and normalized to the median copy number of proteins co-quantified in the meta-assembly (14,000 copies per cell). Pre-aligned RNA-seq data for the NCI-60 panel of cell lines (Reinhold et al, 2019) were downloaded from CellMiner as a processed dataset (RNA: RNA-seq - composite expression), summarized as TPM, and harmonized as described above. The following NCI-60 lines excluded from the meta-assembly were used as test data: SF268 (brain), SF539 (brain), SNB-19 (brain), SNB-75 (brain), U251 (brain), Hs 578T (breast), COLO 205 (colon), HCC2998 (colon), KM12 (colon), CCRF-CEM (leukemia), HL-60(TB) (leukemia), MOLT-4 (leukemia), SR (leukemia), EKVX (non-small cell lung), HOP-62 (non-small cell lung), HOP-92 (non-small cell lung), NCI-H322M (non-small cell lung), M14 (melanoma), Malme-3M (melanoma), MDA-MB-435 (melanoma), NCI-ADR-RES (ovarian), OV-CAR5 (ovarian), SK-OV-3 (ovarian), DU145 (prostate), ACHN (renal), RXF 393 (renal), SN12C (renal), TK-10 (renal), and UO-31 (renal).

**Prostate**: Pre-aligned SWATH data for normal and malignant prostate (Charmpi et al, 2020) were downloaded from PRIDE (PXD004589), quantified for protein, harmonized for gene names as described above, and normalized to the median copy number of proteins co-quantified in the meta-assembly (14,000 copies per cell). Harmonization of samples was more challenging because of different patient-coding schemes for the SWATH and RNA-seq datasets. We obtained metadata annotation for RNA-seq from the SRA Run Selector (PRJNA579899) and then reconciled these identifiers with the PXD004589 identifiers using a key personally communicated by Wenguang Shao (Shao et al, 2019). Tumor-normal pairs were retained in the harmonized dataset if the tumor grade annotations were consistent between SWATH and RNA-seq. The final patient annotations and cross-referencing key is available in Dataset EV6.

**Heart**: The GTEx test dataset from the v8 final data release consisted of 432 left ventricle and 429 atrial appendage autopsy samples from 561 healthy donors (Consortium, 2020). Both GTEx heart tissue sites were considered separately in the analysis. The MAGNet test dataset consisted of 200 cardiomyopathy and 166 healthy control samples (Liu et al, 2021). The EGA test dataset consisted of 149 cardiomyopathy and 113 healthy control samples (Heinig et al, 2017). After RNA-seq alignment as described above, the 1489 samples were concatenated without batch correction before the analysis.

**Breast cancer**: Pre-aligned RNA-seq data for ductal and lobular neoplasms in The Cancer Genome Atlas (TCGA) were downloaded from the Genomic Data Commons portal. The samples were intersected by TCGA identifiers with the published samples profiled by RNA-seq and classified by PAM50 (Ciriello et al, 2015), yielding 796 samples in the test dataset.

### Assembly of CPTAC test datasets

**Breast cancer**: iTRAQ proteomics were downloaded from the Proteomic Data Commons portal (PDC000173; TCGA_Breast_-BI_Proteome.itraq.tsv). iTRAQ proteomics, transcriptomics, and Pinferna predictions were intersected by TCGA identifiers ($n = 99$ samples) and by gene names to yield a common set of samples and genes for comparisons.

**Ovarian cancer**: iTRAQ proteomics were downloaded from the Proteomic Data Commons portal (PDC000113; TCGA_Ovarian_J-HU_Proteome.itraq.tsv). Pre-aligned RNA-seq for Cystic, Mucinous and Serous Neoplasms in The Cancer Genome Atlas were downloaded from the Genomic Data Commons portal and used to make predictions with Pinferna. iTRAQ proteomics, transcriptomics, and Pinferna predictions were intersected by TCGA identifiers ($n = 93$ samples) and by gene names to yield a common set of samples and genes for comparisons.

### CPTAC comparisons

Proteomic or transcriptomic abundances were $\log_2$ transformed and centered by subtracting the mean abundance of each gene from the values of that gene across all samples so that relative comparisons could be made between CPTAC proteomics and RNA-seq data or Pinferna predictions. The Euclidean distance was calculated between the RNA-seq data or Pinferna predictions and the measured CPTAC proteomics for each sample.

### Alternative methods for protein abundance estimation

**PaxDb**: The following aggregated proteomics data were downloaded from PaxDb (Wang et al, 2015) as averaged protein parts per million (ppm): for NCI-60 comparisons, *H. sapiens*—Cell line (Integrated); for general tissue comparisons, *H. sapiens*—Whole organism (Integrated); for prostate tissue comparisons, *H. sapiens*—Prostate gland (Integrated). PaxDb entries less than 0.01 ppm were excluded, and the filtered data were harmonized as described above. Last, each aggregated dataset was normalized to the median copy number of proteins co-quantified in the meta-assembly: for Cell line (Integrated), 8000 copies per cell; for Whole organism (Integrated), 8000 copies per cell; for Prostate gland (Integrated), 9000 copies per cell. Before use, PaxDb outputs were shift- and scale-calibrated to the meta-assembly (see below). Note that PaxDb does not use information from RNA-seq and thus makes a single prediction of protein abundance for each integrated context. Cell line-specific information was not available for NCI-60 lines other than U251.

**PTR**: Protein-transcript ratios in the original publication (Eraslan et al, 2019) were not calculated on the scale of copies per cell. To convert, the protein abundances used for PTR estimation were normalized to the median copy number of proteins co-quantified in the meta-assembly (9000 copies per cell). Using the renormalized protein abundances, we rederived PTRs from the associated RNA-seq fragments per kilobase per million mapped reads (FPKM) as follows: PTR = $\log_{10}$(protein abundance) – $\log_{10}$(FPKM) (Eraslan et al, 2019). PTRs were calculated for each of 29 tissues and in a general manner by using the median PTR. Before use, PTR outputs were shift- and scale-calibrated to the meta-assembly and TPM inputs (see below). NCI-60 predictions used the PTR specific to each cell line's tissue of origin, which was not available for leukemic, breast, or melanoma lines; therefore, these lines were omitted from PTR predictions.

**ProteoEstimator**: Proteoestimator v1.0.5 was installed as a Python package (https://pypi.org/project/proteo-estimator/), and training datasets for breast cancer samples were obtained from Mi Yang (Yang et al, 2020). RNA training data was subset to the same genes as in the ProteoEstimator RNA training dataset for breast cancer (15,115 genes – syn11328694) and likewise for copy number amplifications (CNAs) (16,884 genes – syn11328692). Any genes that were not found in the test datasets were added with an absent count of 0 for RNA or a diploid count of 2 for CNA. ProteoEstimator uses RNA log-transformed Z-scores for predictions. Accordingly, we log-transformed the TPM data from test datasets with a pseudocount, and then calculated the Z-score as (x – mean)/standard deviation. To provide CNA estimates for the NCI-60, processed array comparative genomic hybridization data were downloaded from CellMiner (Reinhold et al, 2012). For prostate cancer CNA estimates, processed exome-seq data were downloaded from (Charmpi et al, 2020). CNA estimates were log-transformed as a copy number ratio compared to normal: $\log_2(x/2)$. The function predict_protein_abundances was used to generate predictions for the NCI-60 or prostate samples. Outputs from ProteoEstimator were intersected with the gene set of Pinferna and back-transformed from $\log_2$ space for relative comparisons of each gene. Relative abundances were calibrated to a proportional scale by median-centering the abundance distribution for each gene about a randomized measurement for that gene in the meta-assembly.

**Shift-scaling of uncalibrated methods**: Before making predictions, PaxDb and PTR were used with RNA-seq (in TPM) of all cell lines in the meta-assembly, and predicted outputs ($X_{uncalibrated}$) were calibrated to SWATH-scaled TMT proteomics of the meta-assembly ($Y_{measurement}$) as follows: $Y_{measurement} = aX_{uncalibrated}^b$, where $a$ is the shift parameter and $b$ is the scaling parameter estimated by maximizing ΔCDF of the calibration. For PaxDb, $a = 1.46$ and $b = 0.886$. For PTR, $a = 112$ and $b = 0.453$.

### Randomized measurements, median null model, and ΔCDF

Randomized SWATH measurement profiles were constructed by randomly sampling a gene-specific copy number estimate for each gene in the proteome and iterating 100 times without replacement. The 100 randomized-measurement distributions were compared to Pinferna and ordered by the K-S statistic, with the median distribution selected as the null model for formal K-S hypothesis testing. To compare prediction methods to the median null, the area between the two was integrated by difference (ΔCDF) using the AUC function in the DescTools (version 0.99.47) package in R.

### Transcription-translation model

The system of ordinary differential equations was solved in MATLAB R2022a using ode15s. The simulation was performed 100 times allowing parameters to vary lognormally about their central nondimensionalized estimate with a coefficient of variation of 10%. After each simulation, 10 time points were chosen randomly and stored, for a total of 1000 points at the end of the simulation.

### CVB3 model

A mass-action model of CVB3 infection (MODEL2110250001) was modified from its published version (Lopacinski et al, 2021) to accommodate CD55 and CXADR abundances as input parameters. Each simulated infection was initialized with abundances for CD55–CXADR and 10 plaque-forming units of CVB3. In silico infections proceeded for 24 h and were iterated 100 times with 5% lognormal coefficient of variation between runs. The median virion output was stored, and overall viral load (measured as mature CVB3 virions and after excluding cases of zero viral load) was fit with a Gaussian mixture model using the Mclust function in the Mclust (version 6.0.0) package in R. The best mixture model by BIC was a three-component model of unequal variance, which classified each sample based on the probability of the sample falling into that component. For the randomized-measurement case (Fig. EV4H), protein estimates were obtained by setting the median TPM value of the heart samples to a randomly selected protein set point in the meta-assembly and linearly scaling the other samples around that set point. This process was replicated 100 times to create a 1489 sample × 100 replicate matrix of randomized measurements for CD55 and CXADR. Each CD55–CXADR pair was passed to the CVB3 model and simulated with MATLAB (version R2022b) as before 100 times each for a total of 14,890,000 simulations, which were threaded to 100 cores over 10 nodes on the Rivanna high-performance computing cluster of the University of Virginia. Taking viral load at 24 h of infection as the phenotype, we classified [CVB3 virions] = 0 nM as resistant, 0 nM < [CVB3 virions] <36 nM as sublytic, and [CVB3 virions] ≥36 nM as lytic.

### Plasmids

pDONR223 DNM1L was obtained from the human ORFeome v5.1 (Yang et al, 2011). DNM1L was PCR amplified with a C-terminal V5 tag and SpeI–EcoRI restriction sites and cloned into SpeI–MfeI-digested pEN_TTmiRc2 (Addgene #25752) to yield pEN_TT DNM1L-V5 (Addgene #220359). Luciferase was PCR amplified from pEN_TT 3xFLAG-luciferase (Addgene #98391) with a C-terminal V5 tag and SpeI–EcoRI restrictions sites and cloned into SpeI–MfeI-digested pEN_TTmiRc2 (Addgene #25752) to yield pEN_TT luciferase-V5 (Addgene #220360). CXADR-V5 was PCR amplified from pLX304 CXADR-V5 (Addgene #82723) with SpeI–MfeI restriction sites and cloned into SpeI–MfeI-digested pEN_TTmiRc2 (Addgene #25752) to yield pEN_TT CXADR-V5 (Addgene #220358). pEN_TT DNM1L-V5, luciferase-V5, and 3xFLAG-NUP37 (Addgene #192299) donor vectors were recombined into pSLIK neo (Addgene #25735) by LR recombination with LR Clonase II (ThermoFisher, 11791020) to obtain pSLIK DNM1L-V5 neo (Addgene #220363), pSLIK luciferase-V5 neo (Addgene #220364), and pSLIK 3xFLAG-NUP37 neo (Addgene #220361). pEN_TT CXADR-V5 donor vector was recombined into pSLIK hygro (Addgene #25737) by LR recombination to obtain pSLIK CXADR-V5 hygro (Addgene #220362). tet pLKO.1-shNUP37 v1 puro (Addgene #192343), tet pLKO.1-shLuciferase puro (Addgene #136587), pSLIK 3xFLAG-luciferase neo (Addgene #98392), pSuperior.retro.neo GFP shDNM1L (Addgene #220356), and pSuperior.retro.neo GFP shScramble (Addgene #220357) were obtained from authors of the source publications (Bajikar et al, 2017; Kashatus et al, 2015; Nagdas et al, 2019; Pereira et al, 2020; Wang et al, 2023).

### Viral transduction

pSLIK and tet pLKO.1 plasmids were packaged as lentivirus in HEK293T cells by calcium phosphate precipitation with psPAX2

(Addgene #12260) and pMD2.G (Addgene #12259) as described (Wang et al, 2011). pSuperior.retro plasmids were packaged as retrovirus with pCL-ampho (Naviaux et al, 1996) in HEK293T cells by calcium phosphate precipitation as described (Wang et al, 2011). Cells were transduced once with lentivirus or retrovirus plus 8 µg/ml polybrene in six-well dishes and allowed to grow for 48 h. Transduced cells were transferred to 10-cm dishes and selected with 2 µg/ml puromycin (NUP37 knockdown in MCF10A, MCF7, and T47D cells), 1500 µg/ml G418 (inducible NUP37 in MCF7 cells, and DNM1L knockdown or inducible DNM1L in CAMA1, T47D, and ZR751 cells), or 100 µg/ml hygromycin (inducible CXADR in AC16 cells) until control plates had cleared.

### CXADR counting

Parental AC16, AC16-iCAR, or AC16-CAR cells were seeded at 50,000 cells/cm$^2$ in six-well dishes for 24 h. Parental and iCAR cells were treated with 1 µg/ml doxycycline (Sigma, D9891) for 30 or 60 min. Cells were trypsinized, centrifuged, resuspended in phosphate-buffered saline, and counted on a Countess 3 Automated Cell Counter (ThermoFisher). Counted cells were re-centrifuged, resuspended in 1× Laemmli sample buffer to a concentration of 5000 cells/µl, and passed through a 25-gauge needle before immunoblotting.

### CVB3 infection

Parental AC16 or AC16-iCAR cells were seeded at 50,000 cells/cm$^2$ in six-well dishes for 24 h then treated with 1 µg/ml doxycycline for 30 or 60 min. Before CVB3 infection, 75% of the cell culture medium was removed, and cells were infected with CVB3 at multiplicity of infection = 5 for 1 h. During the infection, the plates were incubated at 37 °C and rocked every 10–15 min to ensure even coverage of the virus. After 1 h, the media was aspirated, and cells were refed with fresh AC16 growth medium lacking selection antibiotics. Cells were cultured an additional 5 h for a 6-h infection. Cells were lysed in radioimmunoprecipitation buffer plus protease–phosphatase inhibitors: 50 mM Tris-HCl (pH 7.5), 150 mM NaCl, 1% (v/v) Triton X-100, 0.5% (w/v) sodium deoxycholate, 0.1% (w/v) SDS, 5 mM EDTA, 10 µg/ml aprotinin, 10 µg/ml leupeptin, 1 µg/ml pepstatin, 1 µg/ml microcystin-LR, 200 µM Na$_3$VO$_4$, and 1 mM PMSF. Protein concentrations of clarified extracts were determined with the BCA Protein Assay Kit (ThermoFisher, 23227).

### Monte Carlo consensus clustering (M3C)

Consensus clustering of the breast cancer dataset was performed with all 19,062 genes or all 4366 genes inferable by Pinferna. LASSO-modulated protein inferences below zero copies per cell were set to zero. All data were log$_2$ transformed, and genes with zero variance were eliminated before Z-score standardization. Datasets were clustered using the M3C function from the M3C (version 1.18.0) package with the following options: removeplots = T, iters = 10, objective = "entropy", clusteralg = "spectral". Clustering statistics lie within the M3C object, and five consensus clusters were selected based on maximum or near-maximum cluster stability and significance.

### Hierarchical clustering

Breast cancer RNA-seq was log$_2$ transformed and row standardized. For inferred proteomic profiles, M genes were removed before standardization because of zero variance. Data were clustered by Euclidean distance with Ward's linkage using the Heatmap function in the ComplexHeatmap (version 2.12.1) package. Columns were clustered within a subtype defined either by transcriptomics or proteomics. To identify genes that changed disproportionately between mRNA and protein, the Z-scores of the species were subtracted: $Z_{diff} = Z\text{-score}_{Protein} - Z\text{-score}_{mRNA}$. To identify genes of interest that drove luminal reassignments, we filtered for genes with a $|Z_{diff}| \geq 3$ and ranked by the frequency of occurrence, focusing on genes with $|Z_{diff}| \geq 3$ in six or more samples that had undergone subtype reassignment.

### Proportional hazards modeling

Survival analyses were performed with the survfit and coxph functions of the package survival (version 3.4-0) in R. Survival plots were generated using the function ggsurvplot of the package survminer (version 0.4.9) in R.

### NUP37–DNM1L perturbations

For inducible NUP37 or DNM1L experiments, cells were seeded at 25,000 cells/cm$^2$ in six-well dishes for 24 h and then treated with 1 µg/ml doxycycline for 24 h. For NUP37 knockdown, cells were seeded at 12,500 cells/cm$^2$ in six-well dishes for 24 h and then treated with 1 µg/ml doxycycline for 48 h. For ectopic DNM1L knockdown, cells were seeded at 12,500 cells/cm$^2$ in six-well dishes and cultured for 72 h. Cells were lysed in radioimmunoprecipitation buffer plus protease–phosphatase inhibitors, and protein concentrations of clarified extracts were determined as described for CVB3 infection.

## Data availability

Code is available on GitHub for the Pinferna web browser (https://github.com/JanesLab/Pinferna/) and the transcription-translation model (https://github.com/JanesLab/TranscriptionTranslation/).

The source data of this paper are collected in the following database record: biostudies:S-SCDT-10_1038-S44320-024-00064-3.

## Peer review information

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

## Acknowledgements

We thank Jeff Saucerman for suggesting BIC weights, William Shao for testing the Pinferna web browser, the University of Virginia Research Computing group for computational resources and technical guidance, Jennifer Kashatus for help with DNM1L reagents, and Kenley Ellis for editing the manuscript. This study was supported by grants from the National Institutes of Health (U54-CA274499 to KAJ, DFK and R50-CA265089 to LW) and the David & Lucile Packard Foundation (2009-34710 to KAJ), a Cardiovascular Training Grant Fellowship (T32-HL007284 to AJS), a Sture G. Olsson Fellowship (to AJS), and a Human Frontier Science Program Fellowship (LT000469/2021-L to CDG).

## Author contributions

**Andrew J Sweatt**: Conceptualization; Data curation; Software; Formal analysis; Funding acquisition; Validation; Investigation; Visualization; Methodology; Writing—original draft; Writing—review and editing. **Cameron D Griffiths**: Data curation; Funding acquisition; Validation; Investigation; Writing—review and editing. **Sarah M Groves**: Data curation; Software; Formal analysis; Methodology; Writing—review and editing. **B Bishal Paudel**: Formal analysis;

Writing—review and editing. **Lixin Wang**: Resources; Funding acquisition; Validation. **David F Kashatus**: Resources; Supervision; Funding acquisition; Validation; Writing—review and editing. **Kevin A Janes**: Conceptualization; Formal analysis; Supervision; Funding acquisition; Validation; Investigation; Visualization; Writing—original draft; Project administration; Writing—review and editing.

Source data underlying figure panels in this paper may have individual authorship assigned. Where available, figure panel/source data authorship is listed in the following database record: biostudies:S-SCDT-10_1038-S44320-024-00064-3.

## Disclosure and competing interests statement
The authors declare no competing interests.

# Expanded View Figures

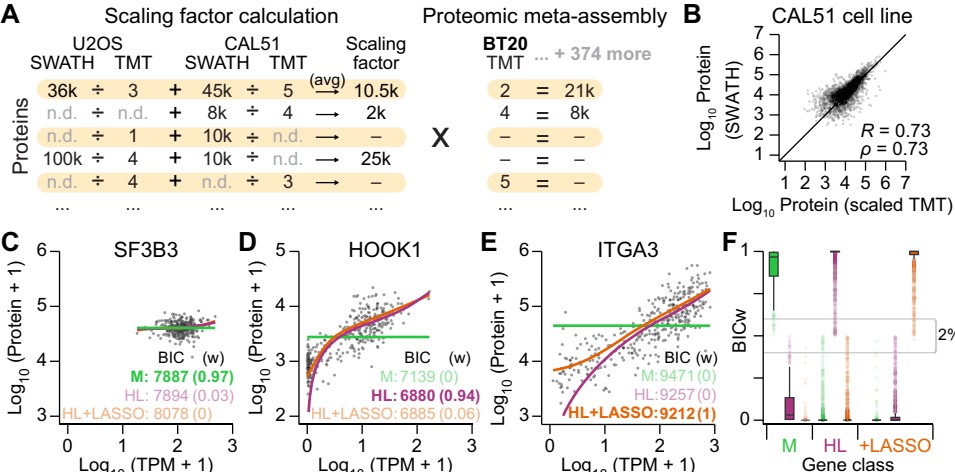

**Figure EV1.   Calibration for the proteomic meta-assembly and model selection examples.**

(**A**) SWATH–TMT scaling factor estimation and calibration of the proteomic meta-assembly. Scaling factors were estimated for each protein by dividing the SWATH intensity by the TMT intensity in U2OS and CAL51 when both data types were available and averaging (avg) the two ratios when possible. The gene-specific scaling factors were used to convert the entire TMT dataset (bold) to proportional protein abundances. (**B**) The reciprocal cross-calibration to that shown in Fig. 1B. Step 1 of Fig. 1A was performed with U2OS data alone and the SWATH-scaled TMT proteomics of CAL51 cells compared with data obtained directly by SWATH. (**C–E**) Model selection of the representative genes shown in Fig. 1C–E. Proportional protein copies per cell were regressed against the mRNA abundance normalized as transcripts per million (TPM). Data are fit with M, HL, and HL + LASSO models. The BIC was used to discriminate the best model for each fit, and BIC weights (w) are shown in parentheses to indicate the relative best-model probability. The lowest BIC is indicated. (**F**) BIC weights (BICw) for each model fit to M, HL, or HL + LASSO ( + LASSO) genes. Range of weights for genes with model ambiguity (two models with a BICw ≥0.4) are boxed (gray) with the percentage indicated. Data information: For (**B**), Pearson's R and Spearman's ρ are shown. For (**C–E**), $n = 369$ cancer cell lines. For (**F**), box-and-whisker plots show the median BICw (horizontal line), interquartile range (IQR, box), and an additional 1.5 IQR extension from the box edge (whiskers) from $n = 4366$ genes.

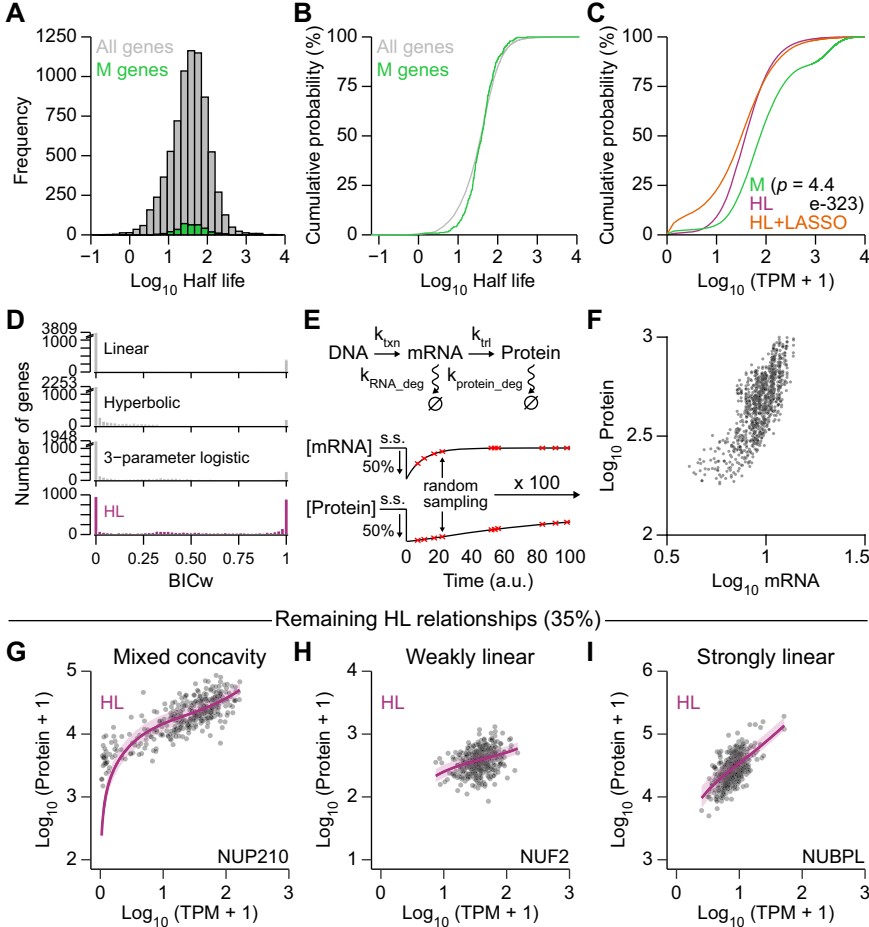

**Figure EV2. Extended characterization of M and HL mRNA-to-protein relationship classes.**

(A, B) M genes do not exhibit longer half-lives compared to other relationship classes. Half-lives for proteins were obtained from (Zecha et al, 2018) and plotted by frequency (A) or as an empirical cumulative distribution function (B) for all genes (gray) and M genes (green). (C) M genes are more abundant by mRNA compared to other relationship classes. mRNA abundance was normalized as TPM and placed on a log scale with a pseudocount of 1. Cumulative distribution functions are shown for M genes (green), HL genes (purple), and HL + LASSO genes (orange). (D) Distribution of BIC weights (BICw) for models encoding linear, hyperbolic, three-parameter logistic, and HL relationships shown in Fig. 2B. (E, F) Log-convex patterns arise when mRNA and protein abundances recover from transient perturbations to steady-state values. A simple transcription–translation model (E) was reduced by 50% and randomly sampled at 10 time points (red) during the return to steady state (s.s.). For the model, the following dimensionless rate parameters were used: $k_{txn} = 1$; $k_{trl} = 1$; $k_{RNA\_deg} = 0.1$; $k_{protein\_deg} = 0.01$. The model was simulated 100 times with lognormally distributed parameter noise (coefficient of variation = 10%) and the joint observation of mRNA and protein abundances ($n = 10$ time points x 100 simulations) is shown in (F). (G–I) Examples of other HL relationships besides those of Fig. 2C, D: mixed concavity (G), weakly linear (H), and strongly linear (I) relationships. Data information: For (A, B), $n = 7029$ genes (gray) and 334 genes (green). For (C), $n = 395$ genes (green), 2569 genes (purple), and 1402 genes (orange). Distributions were compared by K-S test with Šidák correction for multiple-hypothesis testing. For (D), $n = 4366$ genes.

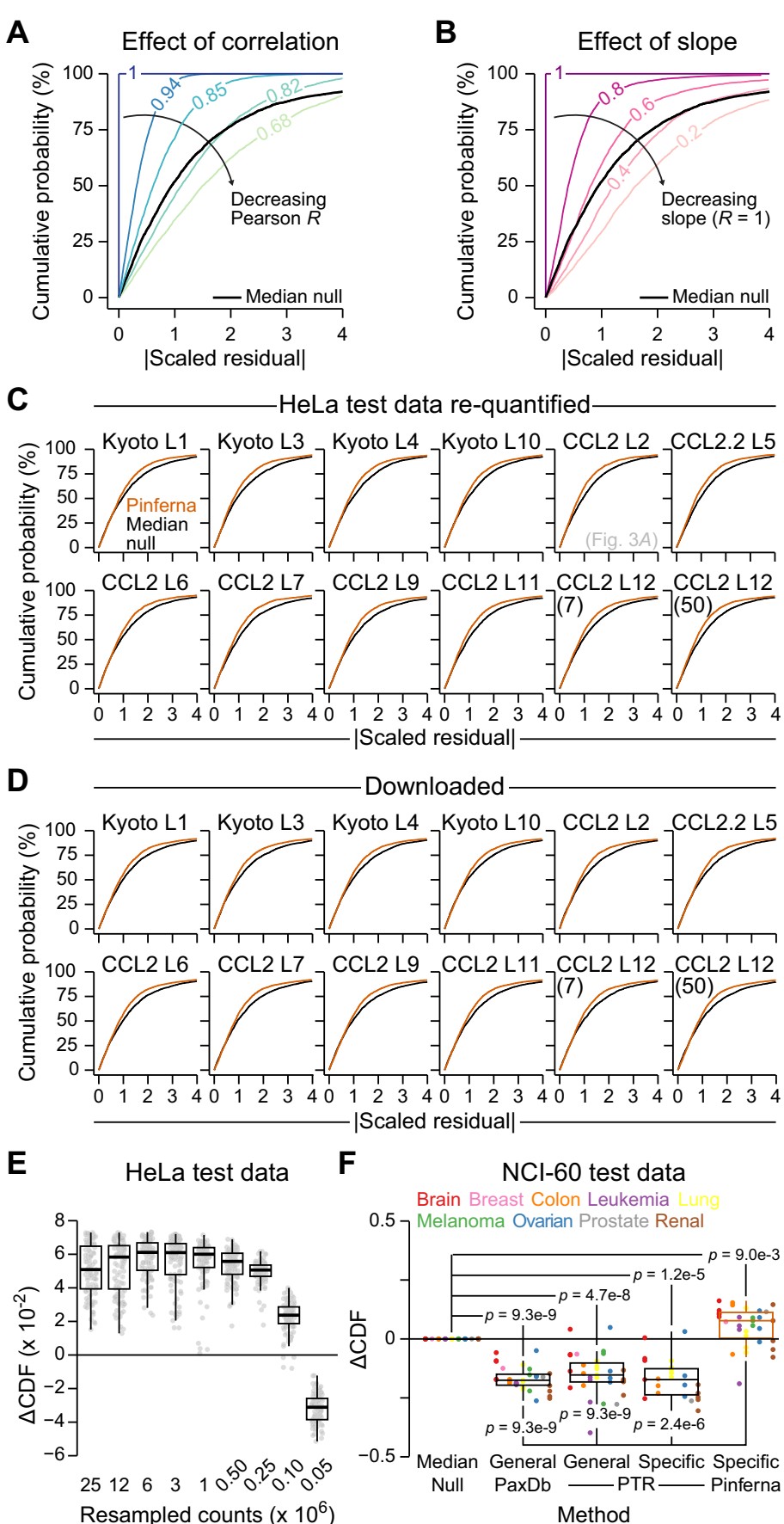

**Figure EV3.  Robustness of Pinferna predictions.**

(**A, B**) Cumulative distribution of scaled residuals degrades with decreasing correlation (**A**) and slope (**B**) between predicted and measured values. A synthetic dataset was created by taking the median TPM and protein value for each gene among 12 HeLa cell lines (Liu et al, 2019) and using perturbations of the synthetic data as a prediction. The median null for this dataset relative to Pinferna is shown for reference. To disrupt correlations (**A**), Gaussian white noise with increasing variance was added to the measured HeLa protein abundances to achieve the indicated Pearson correlations. Measured protein abundances were separately multiplied by the indicated slope while retaining a linear relationship (**B**). (**C, D**) Prediction accuracy is not dependent on SWATH analytical details. Cumulative distribution plots comparing Pinferna and median-null predictions for paired RNA-seq–SWATH datasets of 12 HeLa derivatives (PRJNA437150; PXD009273) named as in the publication (Liu et al, 2019). Proteins were re-quantified from raw SWATH data processed exactly like the meta-assembly ("Methods") (**C**); or, protein quantities were taken directly from the publication (Liu et al, 2019) (**D**). Results from the re-quantified CCL2 L2 derivative are reprinted from Fig. 3A. (**E**) Prediction accuracy is not heavily dependent on RNA-seq read depth. Count-based RNA-seq data for the HeLa lines was averaged and iteratively downsampled (gray) and TPM values re-estimated before making proteome-wide copy-number predictions with Pinferna. See Fig. 3B for an explanation of ΔCDF. (**F**) Prediction accuracy is not dependent on RNA-seq analytical details. Pinferna predictions were made using TPM values taken directly from the original publication (Reinhold et al, 2019), and ΔCDF values were calculated for NCI-60 cell lines excluded from model training (Fig. 1A) and organized by cancer type. Data information: For (**E**), $n = 100$ iterations. For (**F**), $n = 5$ brain, 1 breast, 3 colon, 4 leukemia, 4 lung, 3 melanoma, 3 ovarian, 1 prostate, 5 renal cell lines. Differences between groups were assessed by paired sign-rank tests with Šidák correction. For (**E, F**), box-and-whisker plots show the median ΔCDF (horizontal line), interquartile range (IQR; box), and an additional 1.5 IQR extension (whiskers) of the data.

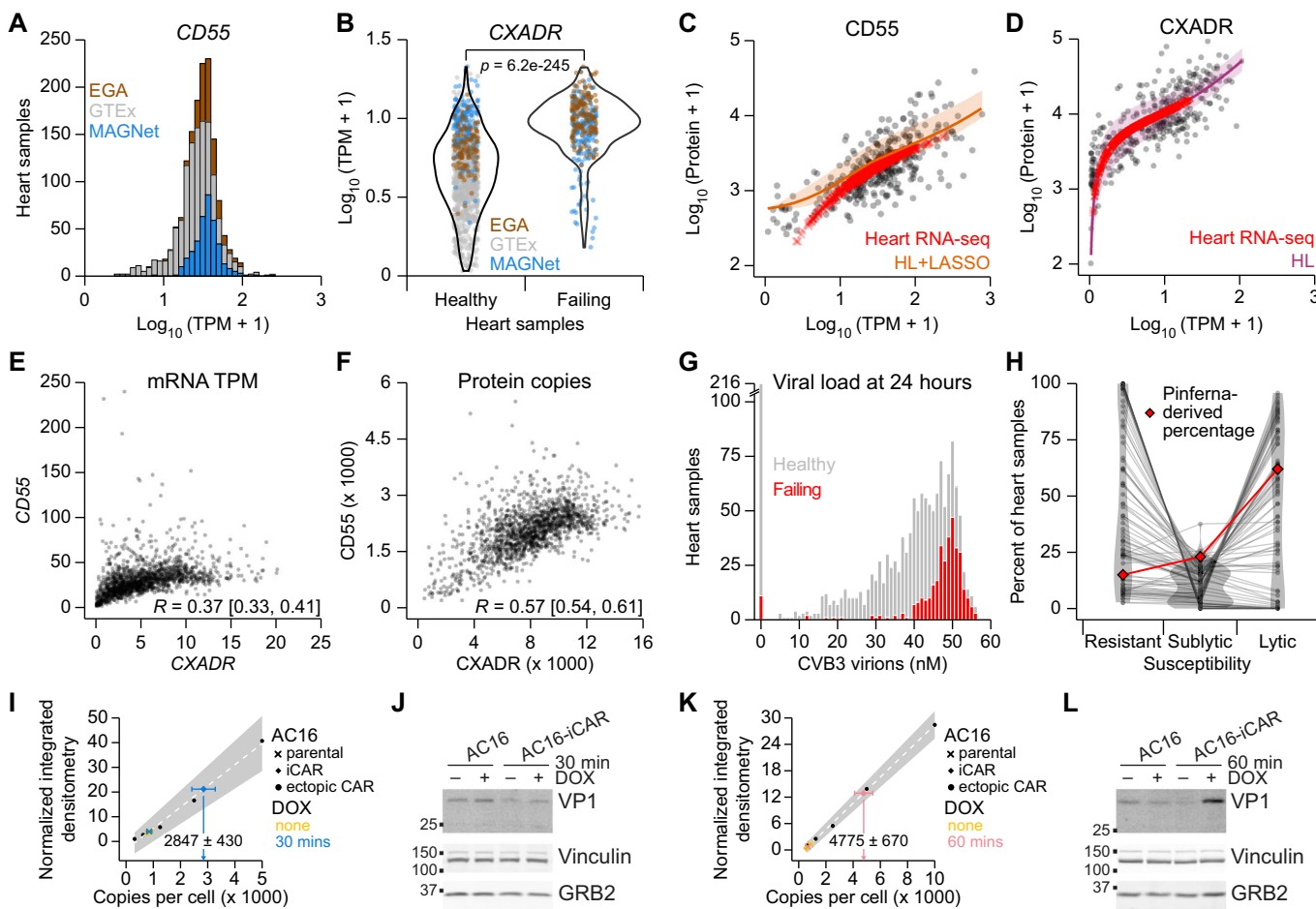

**Figure EV4. Calibrated protein inferences of CD55 and CXADR yield disease-related predictions of coxsackievirus B3 (CVB3) susceptibility.**

(A) Distribution of *CD55* abundance (Dataset EV7) separated by source data: EGA (brown; EGAS00001002454), GTEx (gray; phs000424.v9.p2), MAGNet (blue; GSE141910). (B) *CXADR* is upregulated in failing hearts. *CXADR* abundance (Dataset EV7) was stratified by heart health and colored by source data as in (A). (C, D) Calibration plots (orange (C); purple (D)) and Pinferna predictions for CD55 (C) and CXADR (D) in human heart samples (Heart RNA-seq; red). CD55 deviations from the smoothed best fit are caused by cardiac-specific features in the HL + LASSO regressions. (E, F) CD55–CXADR coregulation is increased at the protein level (F) compared to the mRNA level (E). (G) Replotted histogram of Fig. 4C separated by heart health (Dataset EV7). (H) Predicted prevalence of susceptibility groups based on randomized measurements. After linearly scaling to randomized abundances of CD55 and CXADR, CVB3 infections were simulated for 1489 heart samples as in Fig. 4B. The 24-h end states were quantified by the percentage of samples with resistant ([CVB3 virions] = 0 nM), sublytic (0 nM < [CVB3 virions] < 36 nM), and lytic ([CVB3 virions] ≥ 36 nM) phenotypes (Fig. 4C). Pinferna-derived percentages are overlaid in red. Results from the randomized simulations are connected, and densities in each group are shown by a violin plot in the background. (I–L) AC16 cardiomyocytes were stably transduced with doxycycline (DOX)-inducible CXADR-V5 (iCAR), induced for 30 min (I, J) or 60 min (K, L) and quantified for CXADR (I, K) or infected with CVB3 (multiplicity of infection = 5) for 6 h and immunoblotted for VP1 with vinculin and GRB2 loading controls (J, L). Representative immunoblots of (I, K) are shown in Fig. 4H, J, and immunoblots of (J, L) are quantified in Fig. 4I, K. Data information: For (B–F), n = 1489 heart samples. For (B), differences between groups were assessed by rank-sum test. For (C, D), best-fit calibrations ± 95% confidence intervals are overlaid on the proteomic–transcriptomic data from n = 369 cancer cell lines. For (E, F), the Pearson R is shown with 95% confidence interval in brackets calculated by the Fisher Z transformation. For (H), n = 100 randomizations. For (I, K), n = 4 biological replicates calibrated against a five-point standard curve of AC16-CAR cells fit as a linear model.

