## [Peer Review File · Molecular Systems Biology]

Proteome-wide copy-number estimation from transcriptomics

Andrew Sweatt, Cameron Griffiths, Sarah Groves, B. Paudel, Lixin Wang, David Kashatus, and Kevin Janes

Corresponding author(s): Kevin Janes (kjanes@virginia.edu)

Review Timeline:

Submission Date:	2nd Aug 23
Editorial Decision:	26th Sep 23
Appeal Received:	6th Oct 23
Editorial Decision:	9th Oct 23
Revision Received:	5th May 24
Editorial Decision:	25th Jun 24
Appeal Received:	5th Jul 24
Editorial Decision:	30th Jul 24
Revision Received:	22nd Aug 24
Accepted:	2nd Sep 24

Editors: Maria Polychronidou and Jingyi Hou

Transaction Report:

26th Sep 2023

RE: Manuscript MSB-2023-11919, Proteome-wide copy-number estimation from transcriptomics

Dear Dr Janes,

Thank you again for submitting your work to Molecular Systems Biology. We have now heard back from the three referees who agreed to evaluate your study. As you will see below, the reviewers raise substantial concerns on your work, which unfortunately preclude its publication in Molecular Systems Biology.

The reviewers appreciate that the study addresses a relevant topic. However, they point out that the main conclusions are not well supported and that the relevance of Pinferna for future applications remains unclear. Specifically, they mention that in absence of follow up validations and demonstrations of how Pinferna goes beyond alternative approaches both in terms of performance and in terms of revealing new biological insights the study remains limited. As such, they indicated that they do not support publication in Molecular Systems Biology.

Taken together and given the substantial concerns raised by the reviewers, which go beyond the scope of a major revision, I am afraid that we cannot offer to publish the study. I am sorry that the review of your work did not result in a more favorable outcome on this occasion, but I hope that you will not be discouraged from submitting future work to Molecular Systems Biology. In any case, thank you for the opportunity to examine this work.

Kind regards,

Maria

Maria Polychronidou, PhD
Senior Editor
Molecular Systems Biology

Reviewer #1:

In their manuscript entitled "Proteome-wide copy-number estimation from transcriptomics" Sweatt and co-workers develop and apply an approach to protect protein copy numbers from RNAseq data. To this end, they take advantage of existing TMT mass spec and matching RNAseq data of 375 cancer cell lines. By scaling these relative protein abundances to protein intensities derived from a SWATH experiment they arrive at protein abundance estimates across cell lines. Regression analysis of mRNA and protein abundance estimates allowed them to (i) select which of three alternative models (median, HL, HL+LASSO) describes the mRNA to protein relationship best and (ii) to determine gene-specific model parameters. They show that this approach can predict protein levels from RNAseq data better than a null model based on observed protein abundances only. Applying this approach to a mathematical model of coxsackievirus infection of cardiomyocytes provided meaningful results for the receptors CD55 and CXADR. Moreover, applying the method to breast cancer RNAseq data resulted in reclassification of RNA subtypes according to the inferred proteomic data.

Overall, this is a well-written paper that addresses a key question in gene expression control. A particular strength is that the approach presented is quite simple, well-described and makes intuitive sense. The finding that mRNA to protein relationships can often be described by a hyperbolic-to-linear model is interesting. Also, it is intriguing to see that LASSO identifies a number of "trans" relationships between mRNAs and proteins that make sense. These observations and the Pinferna predictions themselves will be valuable for the scientific community.

Having said this, I think the paper needs to be improved in a number of ways before it can become acceptable for publication in MSB.

1. Validation and application of Pinferna: The described application of Pinferna to modelling viral infection and breast cancer are interesting but not really convincing. What do we really learn from the "1489 systems-biology models of coxsackievirus B3 infection susceptibility"? Also, does the reclassification of breast cancer samples based on Pinferna-derived protein copy number estimates reflect reality better than the RNA-based classification? This is something the authors could actually test using existing matched RNAseq and proteomic datasets for breast cancer (<https://www.nature.com/articles/nature18003>).

2. There is a community-based assessment of different methods to predict protein levels from mRNA levels:

<https://doi.org/10.1016/j.cels.2020.06.013> This is a paper the authors fail to mention. Similar to point 1 above it would be important to benchmark the performance of Pinferna relative to the best-performing model in the community-based resource.

3. There are a few misconceptions related to absolute protein quantification, DDA and DIA methods. The authors should rephrase these sections. Page 3: "Unfortunately, multiplex labeling yields peptide-specific relative quantities that cannot examine absolute differences among proteins within a sample (Pappireddi et al, 2019)." This and a few following statements indicate that the authors have a wrong perception of the fundamental principles of protein quantification by MS: Multiplexed labelling methods such as TMT (but also MS1 labelling methods like MTRAQ, SILAC, dimethyl etc) can provide accurate relative quantification of proteins across samples. These can be used to scale absolute protein abundance (for example, estimates obtained for a single sample) across samples. In fact, it is not difficult to use MS1 peptide intensities from TMT datasets to compute absolute protein abundance estimates. These estimates can then be scaled based on TMT-ratios, exactly the way the authors did for SWATH. "This difficulty is surmounted by data-independent acquisition methods like sequential window acquisition of all theoretical mass spectra (SWATH), which analyzes all precursor ions in a series of mass-to-charge ratio windows (Gillet et al, 2012). After data acquisition, the most sensitive peptide(s) of a protein are summed by intensity, and the resulting data are centered at a reasonable per-cell average (104 copies) to yield absolute estimates of the detectable proteome. SWATH is newer, harder to adapt to different cell lineages, and lower throughput. Consequently, the leading proteomics repository contains ~tenfold fewer SWATH depositions than label-based depositions as of mid-2023 (Perez-Riverol et al, 2019). The need for commoditized SWATH-like protein estimates may forever outpace the ability to generate them directly." SWATH (or other DIA approaches) are not intrinsically superior for absolute protein quantification. Also, there is no such thing as a "most sensitive peptide" - peptides do not have sensitivities. Different peptides have different "flyabilities" in the mass spectrometer (reflecting different ionisation efficiencies etc.). Also, while it is true that SWATH datasets are underrepresented in repositories, I do not see the value in "commoditized SWATH-like estimates": Absolute protein abundance estimates can be obtained from mass spec data in a number of different ways (Top3, iBAQ, APEX etc.). This is true for both DIA and DDA datasets. Some of these algorithms perform better than others, a critical assessment of them can be found here:

<https://doi.org/10.1002/pmic.201300135>. SWATH is not special. In fact, the authors might want to also consider iBAQ or Top3-based estimates available for over 10,000 proteins in proteomicsDB (<https://www.proteomicsdb.org/>).

4. The authors tend to oversimplify the complexity of mRNA and protein correlations. Page 6, "Linear mRNA-protein relationships adequately recapitulate protein expression among genes within a sample, but they are poor at distinguishing protein differences among samples for any given gene (Buccitelli & Selbach, 2020; Fortelny et al, 2017; Tasaki et al, 2022; Wilhelm et al, 2014)." First, proteins cannot be expressed - only genes can. The authors should write "protein abundance differences". Second, this implies that across-gene within sample mRNA-to-protein correlations are always higher than within gene across sample mRNA-to-protein correlations. This is not the case. Page 6 "The current thinking is that the steady-state abundance of mRNA and its intrinsic translation rate create a general "set point" for protein expression, which is buffered or tuned according to the abundance of complexes that stably contain the protein (Buccitelli & Selbach, 2020; Taggart et al, 2020)." Protein-level buffering is a more widely observed phenomenon that is not only observed for members of multiprotein complexes. Therefore, while it is correct that "our working inventory of protein-protein interactions and stable complexes in mammalian cells is far from saturation", stating that this "has thus far prevented a bottom-up reconstruction of mRNA-to-protein relationships" is an oversimplification: There are many other factors involved - protein-protein interactions is just one of them.

5. Page 7, Accuracy test (Fig. 3 A): I agree with the conclusions drawn by the authors. However, I think it is important to clearly state that the improvement of Pinferna compared to the null models is modest.

Additional point:

- Page 3: "Isobaric labeling approaches such as tandem mass tagging (TMT) now quantify 11+ multiplex samples and are the method of choice for proteogenomics (Ellis et al, 2013; Thompson et al, 2003)." This statement is outdated. TMT 18-plex is now available.

Reviewer #2:

In the manuscript by Sweatt et al., the relationship between proteomics and RNAseq data is explored. The authors introduce a model aimed at predicting protein abundance from RNAseq and subsequently apply this model to different systems. While the effort is commendable, I believe the manuscript necessitates significant revisions before being suitable for publication in any journal. I doubt that reasonable efforts can modify this manuscript to meet the standards of journals like MolSysBio. I have serious reservations about the model's technical implementation and its predictions. The model's overall utility remains ambiguous. Merely applying it to different systems without proper validation seems inadequate.

Major concerns:

1. Data Integration: The choice of integrating SWATH data as executed in the study is perplexing and might adversely affect the overall analysis. It's known that absolute abundance data can be directly procured from TMT measurements by quantifying MS1 intensities and then partitioning these intensities based on relative quantification from TMT signals. This method isn't commonly

pursued because MS intensities show only weak correlations with absolute protein abundances - a characteristic also seen with SWATH data.

Nevertheless, the integration of SWATH data with multiplexed data has potential advantages: A) To clarify the relationship between MS1 intensity and protein abundance in DDA experiments, as the association isn't linear (e.g., Schwanhaueser et al.). Still, it might be more effective to use internal standards since the linearity of this relationship in SWATH data is also questionable. B) To address the limited dynamic range of multiplexed proteomics experiments, which SWATH data might overcome, especially for large protein abundance changes.

2. It is unclear to what extent the developed model learns the relationship between protein and mRNA abundance changes versus between mRNA and protein measurement artifacts. e.g., the used TMT data typically has an inherent <100x dynamic range. Shown ratios from one TMT experiment above 50x are essentially meaningless. At the same time, much of the RNAseq data shown seems to fall in the very low regime, e.g., in figure 2D, where sampling noise dominates. Data lacking these artifacts, e.g., SWATH or very deep RNAseq, should be used to evaluate and mitigate.

3. The manuscript fails to provide a convincing demonstration of the model's usefulness. In principle, there is value in developing the kind of model the authors describe. However, the model uses many parameters; just slightly beating simple alternatives is not particularly exciting. To make this an interesting study, I would like to see some strong, non-obvious predictions that are rigorously tested.

Reviewer #3:

Sweatt and co-authors present a computational model (Pinferna) for predicting absolute protein levels (protein copy numbers) from mRNA levels. Three types of models are fitted for each protein: just using the median (M), fitting a non-linear model using the encoding mRNA of the protein as a predictor (HL) and by additionally accounting for transcripts of other genes as predictors (HL-LASSO). The authors test Pinferna on a range of datasets with matched transcriptomics and proteomics datasets. Having a tool to predict protein levels from mRNA levels is definitely helpful. This work is timely and relevant. However, before publication a number of points need to be addressed.

Major points

1. As far as we can see the comparison to competing tools is unfair. Pinferna works on calibrated median protein levels that match to the data that is being used in this study. This is important for predicting absolute protein levels. Competing tools aim to predict the variation of protein levels between conditions or samples rather than absolute levels. Therefore, they also first need to be calibrated accordingly. The fact that competing methods perform significantly(!) worse than random guessing is suspicious (Figure 3). The competitors performance is significantly worse than using the protein medians. While this shows that the models are not calibrated to absolute protein levels, it indicates also that they contain information that is anticorrelated with the actual levels which might be informative.

2. Related: for most applications in biology it is much more important to correctly predict the (relative) differences in protein levels between conditions (e.g. healthy versus disease, ko versus wild-type, etc.). Pinferna focuses on absolute levels, which is legitimate. But the authors should better investigate how well Pinferna also predicts protein level variation.

3. Some of the claims are simply not supported by sufficient evidence. For example, in the analysis of coxsackievirus infections the authors write about changes in infection time and rate as if they had actually observed those differences in human subjects. However, in reality these are just simulations that may or may not be correct. Likewise with the breast cancer study: wordings like "CTNNA2 protein was ubiquitous in the meta-assembly" are misleading, because these are predictions that could not be validated.

4. Using their model the authors reclassify a number of breast cancer samples. They should provide some evidence that this reclassification actually improves the grouping of cancers. For example, are the survival times of the new grouping more consistent than before? E.g. the authors could show Kaplan-Meier Curves to address this question.

Minor points:

5. p3, introduction: 'but mapping the transcriptome to the abundance of proteins is complex' is a weird phrasing

6. p3: 'The need for commoditized SWATH-like protein estimates may forever outpace the ability to generate them directly.' A little pessimistic?

7. p4 "estimating absolute copy numbers from mRNA is historically fraught with uncertainty" This is a very florid way of saying "it is difficult".

8. p4: Need to define what they mean with 'intrinsic translation rate'.

9. p5 LASSO: are only coding transcripts (mRNA) considered here? Or also non-coding transcripts and if yes, which ones?

10. p5: Discussion of M-proteins is too short. I would like to see a thorough discussion of the different types of M, HL and HL-LASSO proteins. E.g. what do the GO enrichments of all three classes look like? Does it biochemically make sense that the respective types of proteins end up in those classes?

11. p5 bottom: Taggart et al. 2020: also cite Goncalves et al. 2017, PMID 29032074

12. p6: 'HL accommodated rare log-concave down relationships that occurred when protein abundance saturated at high transcript abundances ($p < 10^{-15}$; Fig 2C).' What does the pvalue refer to?

13. p6: 'Loss of transcript dependence arises biologically when a protein subunit surpasses the abundance of the complex in which it resides (Taggart et al, 2020; Wuhr et al, 2014), as observed for M genes across their entire measured range of mRNA.' Not only then. In fact, many of the RNA-protein relationships modelled by HL-LASSO may be due to such stoichiometric constraints.
14. p6: 'Specific RPS- or RPL-prefixed transcripts' The encoded proteins are members of the small and large subunits, the names are not the important part.
15. p6: 'p << 10⁻⁴' just write the p-value accurately
16. p7: 'To be useful for new samples, gene-specific model predictions should be more accurate than guesses based on past copy-number estimates of the protein in other settings.' It also has to be better than using 1:1 relationships with all mRNAs of one sample, i.e. assuming the differences between protein levels are the same as between transcript levels or having one reference level per gene for both layers.
17. p7: The sampling process should be better explained. The authors frequently use the term 'randomized measurements', but I think they did not actually randomize any values. If I understood the procedure they actually performed a random sampling of the measurements, by randomly sampling from all observed measurements of one given protein. Thereby they maintain all distributional properties of the protein, especially the mean, median and variance. If this is what the authors did, they should also describe it as such. Otherwise better explain.
18. p7: "Accuracy estimates were comparable when reported protein abundances were used instead of protein re-quantifications performed exactly as done for the training data". I don't understand this sentence. What means 'reported abundances'? If these were the actual measurement the accuracy should be perfect and the analysis trivial. And what means "protein re-quantifications performed exactly as done for the training data"? Does it mean they applied their model? I have no idea what this sentence is trying to convey.
19. p7: 'The results bolster recent claims that typical single-cell RNA-seq data sequenced at ~50,000 reads per cell poorly reflect protein abundances (Brunner et al, 2022; Reimegard et al, 2021) and separately indicate that Pinferna's bulk predictions of protein from mRNA are robust to algorithmic details.' This statement is too broad. For sure some proteins can be accurately quantified/predicted.
20. p7: PTR with proper calibration might outperform pinferna in the prostate data.
21. p9: 'Using the Pinferna estimates of CD55 and CXADR as initial conditions, we created a series of individualized model variants to create a virtual cohort of the human population.' I have no idea what this means.
22. p12: "individuals with fewer than ~5000 copies of CXADR per cell were not susceptible to CVB3" You don't know this. This is just a model prediction. See also my point 3 above.
23. p12 last paragraph: The analogy with bootstrapping does not work. Bootstrapping works by sampling the existing data. Here, the authors propose to use additional (independent) data to improve protein predictions and in this case even different types of data (RNA instead of protein).
24. Materials and Methods: add the supplementary methods here as well. I don't see any point in splitting them off in a separate document.
25. p15: GO methods need to be much more detailed
26. Figure 2c: Why is the difference between Pinferna and the median less significant than the comparison of Pinferna versus PTR even though the mean differences are bigger? (similar other cases exist)

** As a service to authors, EMBO Press offers the possibility to directly transfer declined manuscripts to another EMBO Press title or to the open access journal Life Science Alliance launched in partnership between EMBO Press, Rockefeller University Press and Cold Spring Harbor Laboratory Press. The full manuscript and if applicable, reviewers' reports, are automatically sent to the receiving journal to allow for fast handling and a prompt decision on your manuscript. For more details of this service, and to transfer your manuscript please click on Link Not Available. **

RESPONSE TO REVIEWERS

Reviewer #1:

In their manuscript entitled "Proteome-wide copy-number estimation from transcriptomics" Sweatt and co-workers develop and apply an approach to protect protein copy numbers from RNAseq data. To this end, they take advantage of existing TMT mass spec and matching RNAseq data of 375 cancer cell lines. By scaling these relative protein abundances to protein intensities derived from a SWATH experiment they arrive at protein abundance estimates across cell lines. Regression analysis of mRNA and protein abundance estimates allowed them to (i) select which of three alternative models (median, HL, HL+LASSO) describes the mRNA to protein relationship best and (ii) to determine gene-specific model parameters. They show that this approach can predict protein levels from RNAseq data better than a null model based on observed protein abundances only. Applying this approach to a mathematical model of coxsackievirus infection of cardiomyocytes provided meaningful results for the receptors CD55 and CXADR. Moreover, applying the method to breast cancer RNAseq data resulted in reclassification of RNA subtypes according to the inferred proteomic data.

Overall, this is a well-written paper that addresses a key question in gene expression control. A particular strength is that the approach presented is quite simple, well-described and makes intuitive sense. The finding that mRNA to protein relationships can often be described by a hyperbolic-to-linear model is interesting. Also, it is intriguing to see that LASSO identifies a number of "trans" relationships between mRNAs and proteins that make sense. These observations and the Pinferna predictions themselves will be valuable for the scientific community.

Based on this summary, we interpret Reviewer #1 to be provisionally enthusiastic about the manuscript's overall strength, interest level, and scientific value. Other than linguistic criticisms, the major concerns are i) failure to demonstrate that Pinferna moves archival RNA-seq datasets tangibly closer to measured proteome profiles, and ii) lack of comparison with the best-in-class method of a DREAM challenge. We have already addressed (i) and describe concrete plans for (ii).

Having said this, I think the paper needs to be improved in a number of ways before it can become acceptable for publication in MSB.

1. Validation and application of Pinferna: The described application of Pinferna to modelling viral infection and breast cancer are

Fig R1. Pinferna moves RNA-seq data closer to TMT proteomics. **(A)** Comparison with breast CPTAC data ($n = 99$; Mertins *et al*, 2016). **(B)** Comparison with ovarian CPTAC data ($n = 93$; Zhang *et al*, 2016). Log₂-centered Euclidean distances from TMT data were compared by sign-rank test.

interesting but not really convincing. What do we really learn from the "1489 systems-biology models of coxsackievirus B3 infection susceptibility"? Also, does the reclassification of breast cancer samples based on Pinferna-derived protein copy number estimates reflect reality better than the RNA-based classification? This is something the authors could actually test using existing matched RNAseq and proteomic datasets for breast cancer (<https://www.nature.com/articles/nature18003>).

To evaluate whether Pinferna inferences are closer than RNA-seq to measured proteomics, we obtained two paired RNA-seq/TMT proteomics datasets from CPTAC (Mertins *et al*, 2016; Zhang *et al*, 2016). Using the RNA-seq data, we found that log₂-centered Euclidean distances of Pinferna estimates were significantly closer to the paired TMT proteomics data (Fig R1). Additional evidence for the informativeness of the Pinferna-based classification is provided in the response to Point #4 of Reviewer #3.

2. There is a community-based assessment of different methods to predict protein levels from mRNA

levels: <https://doi.org/10.1016/j.cels.2020.06.013> This is a paper the authors fail to mention. Similar to point 1 above it would be important to benchmark the performance of Pinferna relative to the best-performing model in the community-based resource.

In the first submission, we tried to be very consistent in framing Pinferna as a method for inferring absolute copy numbers. This application required us to develop the Δ CDF of the |scaled residual| as a novel performance metric, which accounts for absolute deviations weighted by absolute variances in protein abundance. The cited DREAM challenge is different in that its goal was to relate variations in protein abundance to variations in transcript abundance by Pearson correlation independent of absolute scale (Yang *et al*, 2020). Nevertheless, we have devised a way to compare the DREAM challenge winner against Pinferna. We will use the Dockerized version of the winning HongyangLi_YuanfangGuan model and then center each gene-specific prediction on a randomized measurement for that sample to calculate Δ CDF as in Fig 3. Doing such a re-calibration for PaxDb and PTR moves Δ CDF values closer to the median null, but Pinferna remains superior (Fig R2).

Fig R2. Recalibration of competing methods to randomized measurements does not surpass the performance of Pinferna. Reassessment of (A) Fig 3C, (B) Fig 3D, and (C) Fig 3E from the original submission.

We would gladly clarify our understanding in these passages if permitted to revise. In the first submission, we crafted a very succinct introduction to get the reader to our results quickly. However, we can easily provide a more complete description of alternative methods for quantification. The poor results of PaxDb, which relies on spectral counting, is consistent with the results of (Ahrne *et al*, 2013). We appreciate the information about proteomicsDB and have used this resource as an orthogonal validation in response to Point #3 of Reviewer #3 (Lautenbacher *et al*, 2022).

4. The authors tend to oversimplify the complexity of mRNA and protein correlations. Page 6, "Linear mRNA-protein relationships adequately recapitulate protein expression among genes within a sample, but they are poor at distinguishing protein differences among samples for any given gene (Buccitelli & Selbach, 2020; Fortelny *et al*, 2017; Tasaki *et al*, 2022; Wilhelm *et al*, 2014)." First, proteins cannot be expressed - only genes can. The authors should write "protein abundance differences". Second, this implies that across-gene within sample mRNA-to-protein correlations are always higher than within gene across sample mRNA-to-protein correlations. This is not the case. Page 6 "The current thinking is that the steady-state abundance of mRNA and its intrinsic translation rate create a general "set point" for protein expression, which is buffered or tuned according to the abundance of complexes that stably contain the protein (Buccitelli & Selbach, 2020; Taggart *et al*, 2020)." Protein-level buffering is a more widely observed phenomenon that is not only observed for members of multiprotein complexes. Therefore, while it is correct that "our working inventory of protein-protein interactions and stable complexes in mammalian cells is far from saturation", stating that this "has thus far prevented a bottom-up reconstruction of mRNA-to-protein relationships" is an oversimplification: There are many other factors involved - protein-protein interactions is just one of them.

Our response here to Reviewer #1 is very similar to Point #3—we commit to providing a longer and more nuanced introduction if permitted to revise.

5. Page 7, Accuracy test (Fig 3 A): I agree with the conclusions drawn by the authors. However, I think it is important to clearly state that the improvement of Pinferna compared to the null models is modest.

We are happy that Reviewer #1 agrees with the evidence of our assertions. In addition, we would like to expand upon how Δ CDF should be interpreted, as it is a new metric. No matter the method, we find that 20–

3. There are a few misconceptions related to absolute protein quantification, DDA and DIA methods. The authors should rephrase these sections. Page 3: "Unfortunately, multiplex labeling yields peptide-specific relative quantities that cannot examine absolute differences among proteins within a sample (Pappireddi *et al*, 2019)." This and a few following statements indicate that the authors have a wrong perception of the fundamental principles of protein quantification by MS: Multiplexed labelling methods such as TMT (but also MS1 labelling methods like MTRAQ, SILAC, dimethyl etc) can provide accurate relative quantification of proteins across samples. These can be used to scale absolute protein abundance (for example, estimates obtained for a single sample) across samples. In fact, it is not difficult to use MS1 peptide intensities from TMT datasets to compute absolute protein abundance estimates. These estimates can then be scaled based on TMT-ratios, exactly the way the authors did for SWATH.

"This difficulty is surmounted by data-independent acquisition methods like sequential window acquisition of all theoretical mass spectra (SWATH), which analyzes all precursor ions in a series of mass-to-charge ratio windows (Gillet *et al*, 2012). After data acquisition, the most sensitive peptide(s) of a protein are summed by intensity, and the resulting data are centered at a reasonable per-cell average (104 copies) to yield absolute estimates of the detectable proteome. SWATH is newer, harder to adapt to different cell lineages, and lower throughput. Consequently, the leading proteomics repository contains ~tenfold fewer SWATH depositions than label-based depositions as of mid-2023 (Perez-Riverol *et al*, 2019). The need for commoditized SWATH-like protein estimates may forever outpace the ability to generate them directly." SWATH (or other DIA approaches) are not intrinsically superior for absolute protein quantification. Also, there is no such thing as a "most sensitive peptide" - peptides do not have sensitivities. Different peptides have different "flyabilities" in the mass spectrometer (reflecting different ionisation efficiencies etc.). Also, while it is true that SWATH datasets are underrepresented in repositories, I do not see the value in "commoditised SWATH-like estimates": Absolute protein abundance estimates can be obtained from mass spec data in a number of different ways (Top3, iBAQ, APEX etc.). This is true for both DIA and DDA datasets. Some of these algorithms perform better than others, a critical assessment of them can be found here: <https://doi.org/10.1002/pmic.201300135>. SWATH is not special. In fact, the authors might want to also consider iBAQ or Top3-based estimates available for over 10,000 proteins in proteomicsDB (<https://www.proteomicsdb.org/>).

35% of genes will be predicted accurately and ~10% of genes will be predicted inaccurately in a given sample (Fig 3A and B). Methods distinguish themselves by the 60% of genes in the middle, which range in absolute performance (|Scaled residual|) from about 0.5–3. Approximating the median null as an elliptical arc over that range, the maximum theoretical Δ CDF is $(0.9 - 0.3) \times (3 - 0.5) \times (1 - \pi/4) = 0.35$. Thus, the performance of Pinferna (Δ CDF = 0.15–0.2) is ~50% of the way to the practical optimum.

Additional point:

- Page 3: *"Isobaric labeling approaches such as tandem mass tagging (TMT) now quantify 11+ multiplex samples and are the method of choice for proteogenomics (Ellis et al, 2013; Thompson et al, 2003)." This statement is outdated. TMT 18-plex is now available.*

Yes, we were aware of this increased plexity but could not find publications at the time that used 18-plex beyond proof-of-concept. We wrote "11+" to emphasize that 11-plex experiments are routine and higher plexity studies will soon become so.

Reviewer #2:

In the manuscript by Sweatt et al., the relationship between proteomics and RNAseq data is explored. The authors introduce a model aimed at predicting protein abundance from RNAseq and subsequently apply this model to different systems. While the effort is commendable, I believe the manuscript necessitates significant revisions before being suitable for publication in any journal. I doubt that reasonable efforts can modify this manuscript to meet the standards of journals like MolSysBio. I have serious reservations about the model's technical implementation and its predictions. The model's overall utility remains ambiguous. Merely applying it to different systems without proper validation seems inadequate.

We recognize that Reviewer #2 is the most pessimistic of the three and provided the fewest overall comments. The concerns about technical implementation are immediately addressed by literature evidence or follow-up analyses of data already in hand. Deficiencies in “proper validation” come from our misunderstanding about the scope of a Method article for *Molecular Systems Biology*. Before submitting, we read the Instructions for Authors:

Method articles describe new methods that will be relevant to a broad audience and represent a clear advance over existing methodologies. New methods should have a clear potential to generate novel discoveries or open new areas of research. Estimations of performance, accuracy and systematic benchmarking should be included when appropriate.

and consulted with four recent Method articles available at the time of submission (Bachman, Gyori & Sorger, 2023; Cuomo *et al*, 2022; Kamal *et al*, 2023; Martin *et al*, 2022). None of these resources indicated to us that novel discoveries must be secondarily validated by methods other than the one reported in the manuscript. We are certainly not opposed to doing such secondary validation; rather, we seek to explain why such validation was omitted from the original submission. Our impression was that new methods needed to be compared rigorously against competing alternatives using a validated reference. For Pinferna, the validated references were three SWATH-MS datasets that were entirely separate from the training data (Fig 3C–E). The potential to generate novel discoveries was illustrated by applying Pinferna to kinetic modeling of virus infection (Fig 4) and cancer subtyping (Fig 5). Below, we describe concrete validation plans for each of these applications.

Major concerns:

1. *Data Integration: The choice of integrating SWATH data as executed in the study is perplexing and might adversely affect the overall analysis. It's known that absolute abundance data can be directly procured from TMT measurements by quantifying MS1 intensities and then partitioning these intensities based on relative quantification from TMT signals. This method isn't commonly pursued because MS intensities show only weak correlations with absolute protein abundances - a characteristic also seen with SWATH data. Nevertheless, the integration of SWATH data with multiplexed data has potential advantages: A) To clarify the relationship between MS1 intensity and protein abundance in DDA experiments, as the association isn't linear (e.g., Schwanhaueser et al.). Still, it might be more effective to use internal standards since the linearity of this relationship in SWATH data is also questionable. B) To address the limited dynamic range of multiplexed proteomics experiments, which SWATH data might overcome, especially for large protein abundance changes.*

We agree with Reviewer #2 that the best absolute measurements by mass spectrometry involve SRM/MRM-type studies with internal standards, but these do not scale to the proteome. However, the comment about the linearity or reproducibility of SWATH-MS is puzzling to us given our reading of the field. A highly influential paper by the Aebersold group along with ten other laboratories showed that SWATH-MS was linear over ~4.5 orders of magnitude with an inter-site coefficient of variation of ~20% (Collins *et al*, 2017). We confirmed this performance (as well as that of TMT) in Fig 1B, where SWATH-MS data from CAL51 cells (PXD003278) were used to scale the CCLE-wide TMT dataset (Nusinow *et al*, 2020). The aggregated SWATH-scaled TMT proteome of U2OS cells from (Nusinow *et al*, 2020) was then compared with an independent SWATH-MS dataset of U2OS cells (PXD000954), and the concordance was excellent ($R = 0.77$, $\rho = 0.76$; Fig 1B).

2. *It is unclear to what extent the developed model learns the relationship between protein and mRNA abundance changes versus between mRNA and protein measurement artifacts. e.g., the used TMT data typically has an inherent <100x dynamic range. Shown ratios from one TMT experiment above 50x are essentially meaningless. At the same time, much of the RNAseq data shown seems to fall in the very low regime, e.g., in figure 2D, where sampling noise dominates. Data lacking these artifacts, e.g., SWATH or very deep RNAseq, should be used to evaluate and mitigate.*

This is an excellent point that we would like to address from multiple angles. First, the originating experiments—TMT ten-plex data on 375 CCLE cells lines were collected in groups of nine with a “bridge” sample comprised of 11 pooled cell lines (Nusinow *et al*, 2020). The aggregated dataset may therefore have an apparent dynamic range of greater than 50–100 if, for example, one sample is 20-fold more abundant than

the bridge and another sample (in a separate run) is 20-fold less abundant than the bridge. To gain a sense of how prevalent this situation might be, we evaluated the dynamic-range statistics of the SWATH-scaled TMT proteome (Dataset EV4). 75% of all genes in the TMT training set have a min-max dynamic range within 50x and 85% of them have a min-max dynamic range within 100x. The number of genes jumps to 98.5% within 50x and 99.7% within 100x when considering the ratio of the 5th and 95th percentiles of the training data. Thus, even if a handful of observations on the extremes were deemed problematic, they would not substantively impact the Pinferna calibration (Fig 1A, Steps 2 and 3).

A second concern relates to whether Pinferna models are reflecting the noise floor of the TMT approach—when labeled peptides have low abundance, they are more affected by the constant background of coeluting peptides, giving rise to “ratio compression” at the lowest fold-change values (Savitski *et al*, 2013). For SERPINB6 specifically (Fig 2D), we observe no instances of protein “dropout” that would suggest that the lowest ~12.5% of the observations (TPM < 30) are at the noise floor. **If permitted to revise**, we will validate the TMT estimates of this target by quantitatively immunoblotting for SERPINB6 (Janes, 2015). We have already identified breast cancer lines in the lab that span the full range of SERPINB6 abundances (HCC1395, BT20, HCC1937, MDAMB436, HCC1806, HCC1500, HCC70, MDAMB231), and several high-quality antibodies are commercially available.

3. The manuscript fails to provide a convincing demonstration of the model's usefulness. In principle, there is value in developing the kind of model the authors describe. However, the model uses many parameters; just slightly beating simple alternatives is not particularly exciting. To make this an interesting study, I would like to see some strong, non-obvious predictions that are rigorously tested.

Please note that Reviewer #2 sees the potential value of Pinferna if the method can go beyond making accurate copy-number predictions of the proteome. The “slightly beating simple alternatives” concern is addressed in the response to Point #5 of Reviewer #1 above. We have developed a highly feasible two-pronged plan for “strong, non-obvious predictions,” one related to kinetic modeling of virus infection (Fig 4) and the other related to cancer subtyping (Fig 5). The plan for Fig 4 is described in the response to Point #22 of Reviewer #3 below; we describe the Fig 5 plan here.

The initial assessment of HL+LASSO models (Fig 3E and F) supported that LASSO features of a gene contained biological meaning, but it only implied that some of these features (i.e., other genes) were causative. We will build off of the CDK4 observation and its densely connected network of known interactions appearing as LASSO features (Fig 5F). The known CDK4 interaction mechanisms captured by Pinferna are already diverse. E2F4 is a putative substrate of CDK4 and provides a positive LASSO coefficient possibly by protein stabilization (Scime *et al*, 2008). Conversely, CCND1 provides a negative LASSO coefficient and is a direct protein-protein binding partner with a short half-life that increases CDK4 turnover (Matsushima *et al*, 1992). TSPAN31 also adds a negative LASSO coefficient but appears to do so by a miRNA-like process, whereby it acts as an antisense transcript binding the 3' UTR of CDK4 (Xia *et al*, 2020). **If permitted to revise**, we will

examine two unexpected genes among the LASSO features for CDK4: the nucleoporin NUP37 (positive coefficient) and the mitochondrial dynamin-like GTPase DNM1L (negative coefficient). We already have cloned inducible NUP37 perturbations and engineered breast epithelial lines (Fig R3A and B) for another project (Wang *et al*, 2023).

Separately, our collaborators have knocked out DNM1L in a colorectal cancer cell line (Fig R3C). We will measure CDK4 mRNA (by qPCR) and CDK4 protein (by immunoblotting) in these cells and test whether the CDK4 mRNA-to-protein relationship is altered by NUP37 and DNM1L. There is no obvious link between a cyclin-dependent kinase (CDK4) and a nucleoporin (NUP37) or a mitochondrial GTPase (DNM1L), making these predictions particularly stringent.

Fig R3. Genetic and cell-line resources immediately available for a major revision. **(A–B)** An MCF10A (breast epithelial) derivative expressing inducible NUP37 or shNUP37 (Wang *et al*, 2023). **(C)** HCT116 (colorectal cancer) clones targeted with sgDNM1L, with homozygous knockout clones highlighted in green.

Reviewer #3:

Sweatt and co-authors present a computational model (Pinferna) for predicting absolute protein levels (protein copy numbers) from mRNA levels. Three types of models are fitted for each protein: just using the median (M), fitting a non-linear model using the encoding mRNA of the protein as a predictor (HL) and by additionally accounting for transcripts of other genes as predictors (HL-LASSO). The authors test Pinferna on a range of datasets with matched transcriptomics and proteomics datasets. Having a tool to predict protein levels from mRNA levels is definitely helpful. This work is timely and relevant. However, before publication a number of points need to be addressed.

Our interpretation of this summary is that Reviewer #3 is very enthusiastic about the topic and its timely relevance but raises two issues: i) skepticism about the overall performance of Pinferna, which we believe relates to a misunderstanding of the Δ CDF performance metric and how calibration-vs.-test sets were separated; ii) inadequate secondary validation of Pinferna-derived predictions. The response to Reviewer #3 clarifies the performance metric along with some additional procedural details. It also adds secondary experimental plans that are complementary to those described in the response to Point #3 of Reviewer #2.

Major points

1. As far as we can see the comparison to competing tools is unfair. Pinferna works on calibrated median protein levels that match to the data that is being used in this study. This is important for predicting absolute protein levels. Competing tools aim to predict the variation of protein levels between conditions or samples rather than absolute levels. Therefore, they also first need to be calibrated accordingly. The fact that competing methods perform significantly(!) worse than random guessing is suspicious (Figure 3). The competitors performance is significantly worse than using the protein medians. While this shows that the models are not calibrated to absolute protein levels, it indicates also that they contain information that is anticorrelated with the actual levels which might be informative.

We should clarify a couple points of misunderstanding:

- i) Although Pinferna is calibrated to an average absolute abundance measurement from SWATH-MS of two cell lines (U2OS and CAL51; Fig EV1A), it has no SWATH-MS information about the test data (Fig 3), which are normalized independently. We would emphasize this point in a revision.
- ii) We selected PaxDb and PTR as competing methods because they claim to make absolute protein predictions (Eraslan *et al*, 2019; Wang *et al*, 2015). Techniques that only predict relative changes were avoided because of the calibration uncertainty noted by Reviewer #3. However, we have come up with a way to compare “best-case” calibrations across absolute and relative methods, which are described in the response to Point #2 of Reviewer #1.
- iii) The calibrations of PaxDb and PTR predictions were done exactly as with SWATH-MS, which centers the full distribution of proteins at 10,000 copies. Not all proteins detectable by SWATH-MS were amenable to predictions by PaxDb and PTR. Therefore, we calibrated the prediction set to the median copy number (based on the SWATH-MS calibration) for the proteins that could be predicted by these alternative methods. We will revisit our explanation of these details in the Materials and Methods.
- iv) Reviewer #3 is correct that a below-median Δ CDF value can indicate a gross miscalibration, which is exactly what we wanted to drive home with the comparisons in Fig 3. Because PaxDb and PTR were calibrated identically to SWATH-MS, it implies that the aggregate distribution of thousands of predicted proteins is skewed or displaced from what it should be.
- v) Taking this comment together with Point #5 of Reviewer #1, it appears we did a poor job of introducing Δ CDF as a novel metric of absolute prediction accuracy: Δ CDF is not equivalent to a Pearson correlation. A set of predictions could be highly correlated with measured values, but if the slope of the relationships is far from one, the Δ CDF will be poor. Reciprocally, a cloud of predictions and observations (each with 10% error) would have a poor correlation, but the Δ CDF would be high because the errors are small relative to the absolute value itself. **If permitted to revise**, we would dwell more on the Δ CDF and its characteristics with an Expanded View Figure including simulated examples that provide some intuition.

2. Related: for most applications in biology it is much more important to correctly predict the (relative) differences in protein levels between conditions (e.g. healthy versus disease, ko versus wild-type, etc.). Pinferna focuses on absolute levels, which is legitimate. But the authors should better investigate how well Pinferna also predicts protein level variation.

We appreciate that Reviewer #3 sees legitimate value in predicting absolute copy numbers. To assess performance by relative differences, we calculated the \log_2 -centered Euclidean distances between PTR or

Fig R4. Fold change values among NCI-60 and prostate samples are predicted more accurately by Pinferna than PTR. Log₂-centered Euclidean distances from TMT data were compared by sign-rank test.

Pinferna and the measured SWATH-MS values for each sample. Fold-change values for Pinferna were significantly closer to the SWATH-MS ground truth for all test data (Fig R4).

3. Some of the claims are simply not supported by sufficient evidence. For example, in the analysis of coxsackievirus infections the authors write about changes in infection time and rate as if they had actually observed those differences in human subjects. However, in reality these are just simulations that may or may not be correct. Likewise with the breast cancer study: wordings like "CTNNA2 protein was ubiquitous in the meta-assembly" are misleading, because these are predictions that could not be validated.

We agree with Reviewer #3 that we did not validate this assertion, because it was not an interpretive claim of a Pinferna prediction but rather a description of the TMT data that we started with. If deemed important for a revision, we would gladly obtain a CTNNA2 antibody and immunoblot cell lines on the low end of the RNA-seq TPM spectrum for this gene (we have identified several lines already in hand). Immediately, when we consult ProteomicsDB—a resource recommended by Reviewer #1 (Lautenbacher *et al*, 2022)—we find independent evidence that CTNNA2 protein is detectable across all cell lines assembled (Fig R5).

Fig R5. Screenshot of CTNNA2 protein abundance among cell lines quantified by MS1-iBAQ and aggregated in ProteomicsDB. We expand upon the coxsackievirus B3 (CVB3) infection model in the response to Point #22 below.

Fig R6. Pinferna-reclassified Luminal A cases identify a high-risk subpopulation.

Minor points:

5. p3, introduction: 'but mapping the transcriptome to the abundance of proteins is complex' is a weird phrasing
6. p3: 'The need for commoditized SWATH-like protein estimates may forever outpace the ability to generate them directly.' A little pessimistic?
7. p4 "estimating absolute copy numbers from mRNA is historically fraught with uncertainty" This is a very florid way of saying "it is difficult".
8. p4: Need to define what they mean with 'intrinsic translation rate'.
9. p5 LASSO: are only coding transcripts (mRNA) considered here? Or also non-coding transcripts and if yes, which ones?

These minor changes are all easily implemented in a revision.

10. p5: Discussion of M-proteins is too short. I would like to see a thorough discussion of the different types of M, HL and HL-LASSO proteins. E.g. what do the GO enrichments of all three classes look like? Does it biochemically make sense that the respective types of proteins end up in those classes?

We provided the GO enrichments for all categories in Table EV6. We did not expand on the other enrichment categories because HL and HL+LASSO categories comprise such large fractions of the Pinferna proteome (59% and 32% respectively). Those enrichments in Table EV6 likely reflect proteins reliably detected by TMT and SWATH-MS rather than the subcategories themselves.

11. p5 bottom: Taggart et al. 2020: also cite Goncalves et al. 2017, PMID 29032074

Easily implemented in a revision.

12. p6: 'HL accommodated rare log-concave down relationships that occurred when protein abundance saturated at high transcript abundances ($p < 10^{-15}$; Fig 2C).' What does the pvalue refer to?

Our apologies—this p value carried over from an early draft where we tested whether log-concave down genes had significantly higher abundance than other HL genes. We would remove this p value if permitted to revise.

13. p6: 'Loss of transcript dependence arises biologically when a protein subunit surpasses the abundance of the complex in which it resides (Taggart et al, 2020; Wuhr et al, 2014), as observed for M genes across their entire measured range of mRNA.' Not only then. In fact, many of the RNA-protein relationships modelled by HL-LASSO may be due to such stoichiometric constraints.

14. p6: 'Specific RPS- or RPL-prefixed transcripts' The encoded proteins are members of the small and large subunits, the names are not the important part.

Both easily implemented in a revision.

15. p6: ' $p < 10^{-4}$ ' just write the p-value accurately

This p value was estimated computationally from 10^4 random draws of the STRING database. Given the null distribution of Fig 2F, it would require a prohibitive number of randomizations to estimate the p value exactly. We believe that the current number of randomizations is sufficient to reach the conclusion that the number of observed STRING interactors is highly unlikely to arise by chance (Huber, 2019).

16. p7: 'To be useful for new samples, gene-specific model predictions should be more accurate than guesses based on past copy-number estimates of the protein in other settings.' It also has to be better than using 1:1 relationships with all mRNAs of one sample, i.e. assuming the differences between protein levels are the same as between transcript levels or having one reference level per gene for both layers.

We can easily add this comparator in a revision. The result will be poor no matter the metric. We illustrated a variation of this idea in Fig EV4H.

17. p7: *The sampling process should be better explained. The authors frequently use the term 'randomized measurements', but I think they did not actually randomize any values. If I understood the procedure they actually performed a random sampling of the measurements, by randomly sampling from all observed measurements of one given protein. Thereby they maintain all distributional properties of the protein, especially the mean, median and variance. If this is what the authors did, they should also describe it as such. Otherwise better explain.*

If, by “values,” Reviewer #3 means measurements of different proteins, then Reviewer #3 is correct. We sought to retain the statistical properties of each protein as a stringent reference. We would attempt to clarify the sampling process along with the interpretation of Δ CDF in a revision as described in the Point #1 response to Reviewer #3.

18. p7: *"Accuracy estimates were comparable when reported protein abundances were used instead of protein re-quantifications performed exactly as done for the training data". I don't understand this sentence. What means 'reported abundances'? If these were the actual measurement the accuracy should be perfect and the analysis trivial. And what means "protein re-quantifications performed exactly as done for the training data"? Does it mean they applied their model? I have no idea what this sentence is trying to convey.*

We should have been more explicit here. “Reported protein abundances” are the processed quantitative data taken directly from the publication or deposition. “Protein re-quantifications” were our best efforts to reproduce the quantification from raw data files based on the methods described in the paper. Ideally, the two should be identical, but they never are in practice because depositors do not provide containerized versions of their processing algorithms. Package versions get updated with different dependencies, yielding numbers that are close but not identical. Our point is that these small differences (both for public RNA-seq data and SWATH-MS data) have a negligible impact on the performance of Pinferna.

19. p7: *'The results bolster recent claims that typical single-cell RNA-seq data sequenced at ~50,000 reads per cell poorly reflect protein abundances (Brunner et al, 2022; Reimegard et al, 2021) and separately indicate that Pinferna's bulk predictions of protein from mRNA are robust to algorithmic details.' This statement is too broad. For sure some proteins can be accurately quantified/predicted.*

We can revise this statement to reflect our intent: 50,000 reads per cell is insufficient to glean absolute protein abundances with Pinferna. The reason is that TPM estimates become highly unstable when calculated from so few reads. Unstable TPM values yield inaccurate inferences from the Pinferna models.

20. p7: *PTR with proper calibration might outperform pinferna in the prostate data.*

We show results from a best-case calibration in the response to Point #2 of Reviewer #1 (Fig R2). The re-calibration actually causes PTR to perform somewhat worse.

21. p9: *'Using the Pinferna estimates of CD55 and CXADR as initial conditions, we created a series of individualized model variants to create a virtual cohort of the human population.' I have no idea what this means.*

We were too terse in the first submission. The model of CVB3 infection takes (among other inputs) the abundance of cellular CD55 and CXADR as the starting point for an in silico infection (Lopacinski *et al*, 2021). We used Pinferna to convert heart RNA-seq TPMs of CD55 and CXADR to copies per cell and then initialized 1489 different in silico models for CVB3 infection based on the inferred CD55–CXADR protein abundances. Thus, model outcomes describe the diversity of responses predicted based only on inferred protein differences in CD55 and CXADR in this human-derived “virtual” population (Laubenbacher, Sluka & Glazier, 2021).

22. p12: "individuals with fewer than ~5000 copies of CXADR per cell were not susceptible to CVB3" You don't know this. This is just a model prediction. See also my point 3 above.

This is a model prediction we would happily test in vitro if given the opportunity to revise. Previously, we showed that a cardiomyocyte-derived cell line (AC16) is not permissive for CVB3 infection unless CXADR (also known as CAR) is overexpressed (Shah *et al*, 2017) (Fig R7A). In a later publication (Lopacinski *et al*, 2021), we performed RNA-seq on AC16 cells with or without CXADR overexpression—using a rudimentary HL model based on limited data at the time, we estimated that parental AC16 cells harbored about 2300 CXADR copies per cell (Fig R7B). Using Pinferna, the improved protein estimate is 4478 copies per cell, which is still less than the stated threshold. **If permitted to revise**, we will perform an absolute quantification of endogenous CXADR by immunoblotting with recombinant standards as in (Lopacinski *et al*, 2021). Then, we will mildly express ectopic CXADR or sort endogenous subpopulations with the goal of evaluating the 5000 copy-per-cell threshold stemming from Pinferna and the in silico model. This experimental effort will add another non-obvious prediction that is rigorously tested (Point #3 of Reviewer #2).

Fig R7. AC16 cardiomyocytes are not infected with ~2300 endogenous copies per cell of CXADR/CAR. **(A)** AC16 cells lacking overexpressed CXADR/CAR-V5 do not express viral protein 1 (VP1) or undergo apoptosis in response to CVB3 infection (Shah *et al*, 2017). **(B)** Early HL-based protein estimates of CXADR/CAR based on RNA-seq of AC16 cells (Lopacinski *et al*, 2021).

23. p12 last paragraph: The analogy with bootstrapping does not work. Bootstrapping works by sampling the existing data. Here, the authors propose to use additional (independent) data to improve protein predictions and in this case even different types of data (RNA instead of protein).

Easily ceded in a revision.

24. Materials and Methods: add the supplementary methods here as well. I don't see any point in splitting them off in a separate document.

25. p15: GO methods need to be much more detailed

Both easily implemented in a revision.

26. Figure 2c: Why is the difference between Pinferna and the median less significant than the comparison of Pinferna versus PTR even though the mean differences are bigger? (similar other cases exist)

We thank Reviewer #3 for bringing this to our attention. The Fig 2 caption states that differences between groups were assessed by rank-sum test. Comparisons with median nulls involve a large number of tied ranks, requiring an approximation of the statistic rather than the exact calculation, which is possible with other comparisons in the subpanel. Considering that predictions from each method are paired by sample, we will revise the statistical analysis and change to a binomial test that will not have the same complication. The p values will still not be proportional to the perceived effect size because the binomial test is a nonparametric method assessing directional differences. However, it will avoid complications with ties and the overall conclusions should be unaltered.

References

- Ahrne E, Molzahn L, Glatter T, Schmidt A (2013) Critical assessment of proteome-wide label-free absolute abundance estimation strategies. *Proteomics* **13**: 2567-2578
- Bachman JA, Gyori BM, Sorger PK (2023) Automated assembly of molecular mechanisms at scale from text mining and curated databases. *Mol Syst Biol* **19**: e11325
- Collins BC, Hunter CL, Liu Y, Schilling B, Rosenberger G, Bader SL, Chan DW, Gibson BW, Gingras AC, Held JM, Hirayama-Kurogi M, Hou G, Krisp C, Larsen B, Lin L, Liu S, Molloy MP, Moritz RL, Ohtsuki S, Schlapbach R et al (2017) Multi-laboratory assessment of reproducibility, qualitative and quantitative performance of SWATH-mass spectrometry. *Nat Commun* **8**: 291
- Cuomo ASE, Heinen T, Vagiaki D, Horta D, Marioni JC, Stegle O (2022) CellRegMap: a statistical framework for mapping context-specific regulatory variants using scRNA-seq. *Mol Syst Biol* **18**: e10663
- Eraslan B, Wang D, Gusic M, Prokisch H, Hallstrom BM, Uhlen M, Asplund A, Ponten F, Wieland T, Hopf T, Hahne H, Kuster B, Gagneur J (2019) Quantification and discovery of sequence determinants of protein-per-mRNA amount in 29 human tissues. *Mol Syst Biol* **15**: e8513
- Huber W (2019) Reporting p Values. *Cell Syst* **8**: 170-171
- Janes KA (2015) An analysis of critical factors for quantitative immunoblotting. *Sci Signal* **8**: rs2
- Kamal A, Arnold C, Claringbould A, Moussa R, Servaas NH, Kholmatov M, Daga N, Nogina D, Mueller-Dott S, Reyes-Palomares A, Palla G, Sigalova O, Bunina D, Pabst C, Zaugg JB (2023) GRaNIE and GRaNPA: inference and evaluation of enhancer-mediated gene regulatory networks. *Mol Syst Biol* **19**: e11627
- Laubenbacher R, Sluka JP, Glazier JA (2021) Using digital twins in viral infection. *Science* **371**: 1105-1106
- Lautenbacher L, Samaras P, Muller J, Grafberger A, Shraideh M, Rank J, Fuchs ST, Schmidt TK, The M, Dallago C, Wittges H, Rost B, Krcmar H, Kuster B, Wilhelm M (2022) ProteomicsDB: toward a FAIR open-source resource for life-science research. *Nucleic Acids Res* **50**: D1541-D1552
- Lopacinski AB, Sweatt AJ, Smolko CM, Gray-Gaillard E, Borgman CA, Shah M, Janes KA (2021) Modeling the complete kinetics of coxsackievirus B3 reveals human determinants of host-cell feedback. *Cell Syst* **12**: 304-323 e313
- Martin PCN, Kim H, Lovkvist C, Hong BW, Won KJ (2022) Vesalius: high-resolution in silico anatomization of spatial transcriptomic data using image analysis. *Mol Syst Biol* **18**: e11080
- Matsushime H, Ewen ME, Strom DK, Kato JY, Hanks SK, Roussel MF, Sherr CJ (1992) Identification and properties of an atypical catalytic subunit (p34PSK-J3/cdk4) for mammalian D type G1 cyclins. *Cell* **71**: 323-334
- Mertins P, Mani DR, Ruggles KV, Gillette MA, Clauser KR, Wang P, Wang X, Qiao JW, Cao S, Petralia F, Kawaler E, Mundt F, Krug K, Tu Z, Lei JT, Gatza ML, Wilkerson M, Perou CM, Yellapantula V, Huang KL et al (2016) Proteogenomics connects somatic mutations to signalling in breast cancer. *Nature* **534**: 55-62
- Nusinow DP, Szpyt J, Ghandi M, Rose CM, McDonald ER, 3rd, Kalocsay M, Jane-Valbuena J, Gelfand E, Schweppe DK, Jedrychowski M, Golji J, Porter DA, Rejtar T, Wang YK, Kryukov GV, Stegmeier F, Erickson BK, Garraway LA, Sellers WR, Gygi SP (2020) Quantitative Proteomics of the Cancer Cell Line Encyclopedia. *Cell* **180**: 387-402 e316
- Savitski MM, Mathieson T, Zinn N, Sweetman G, Doce C, Becher I, Pachi F, Kuster B, Bantscheff M (2013) Measuring and managing ratio compression for accurate iTRAQ/TMT quantification. *J Proteome Res* **12**: 3586-

Scime A, Li L, Ciavarra G, Whyte P (2008) Cyclin D1/cdk4 can interact with E2F4/DP1 and disrupts its DNA-binding capacity. *J Cell Physiol* **214**: 568-581

Shah M, Smolko CM, Kinicki S, Chapman ZD, Brautigan DL, Janes KA (2017) Profiling Subcellular Protein Phosphatase Responses to Coxsackievirus B3 Infection of Cardiomyocytes. *Mol Cell Proteomics* **16**: S244-S262

Wang L, Paudel BB, McKnight RA, Janes KA (2023) Nucleocytoplasmic transport of active HER2 causes fractional escape from the DCIS-like state. *Nat Commun* **14**: 2110

Wang M, Herrmann CJ, Simonovic M, Szklarczyk D, von Mering C (2015) Version 4.0 of PaxDb: Protein abundance data, integrated across model organisms, tissues, and cell-lines. *Proteomics* **15**: 3163-3168

Xia Y, Deng Y, Zhou Y, Li D, Sun X, Gu L, Chen Z, Zhao Q (2020) TSPAN31 suppresses cell proliferation in human cervical cancer through down-regulation of its antisense pairing with CDK4. *Cell Biochem Funct* **38**: 660-668

Yang M, Petralia F, Li Z, Li H, Ma W, Song X, Kim S, Lee H, Yu H, Lee B, Bae S, Heo E, Kaczmarczyk J, Stepniak P, Warchol M, Yu T, Calinawan AP, Boutros PC, Payne SH, Reva B et al (2020) Community Assessment of the Predictability of Cancer Protein and Phosphoprotein Levels from Genomics and Transcriptomics. *Cell Syst* **11**: 186-195 e189

Zhang H, Liu T, Zhang Z, Payne SH, Zhang B, McDermott JE, Zhou JY, Petyuk VA, Chen L, Ray D, Sun S, Yang F, Chen L, Wang J, Shah P, Cha SW, Aiyetan P, Woo S, Tian Y, Gritsenko MA et al (2016) Integrated Proteogenomic Characterization of Human High-Grade Serous Ovarian Cancer. *Cell* **166**: 755-765

10th Oct 2023

Manuscript Number: MSB-2023-11919R-Q

Title: Proteome-wide copy-number estimation from transcriptomics

Dear Kevin,

Thank you for your message regarding our decision on your manuscript MSB-2023-11919. I have now had the chance to read the manuscript and your point-by-point response to the reviewers' comments and I have discussed them with the team. As I will explain below, we would not be opposed to considering a revised and extended manuscript addressing the issues raised by the reviewers.

During the review, the reviewers acknowledged that the addressed topic is relevant and that Pinferna seems potentially useful for the field. However, they indicated that as it stands the superiority of Pinferna in terms of performance and potential applications was not well supported. They also raised several technical concerns related to these issues, a prominent one being that the fact that competing methods perform worse than random seems worrisome. Moreover, they pointed out that the study would benefit from some follow up analyses supporting the relevance of Pinferna and validating some of its predictions (e.g. non intuitive ones). We think that the additional analyses and clarifications outlined in your preliminary point by point response seem potentially promising for addressing the reviewers' concerns, including those summarised above.

As such, we would invite you to submit a revised manuscript, which will be sent back to the reviewers so that they can assess if their concerns have been satisfactorily addressed.

The revised manuscript will be subjected to peer review and as you probably understand, we can give no guarantee about the eventual acceptability of the study. If you do decide to resubmit the extended/revised manuscript, we would ask you to enclose with your resubmission a point-by-point response to the points raised.

Let me know in case you have any questions or if there is anything you would like to discuss in further detail.

Kind regards,

Maria

Maria Polychronidou, PhD
Senior Editor
Molecular Systems Biology

We realize that it is difficult to revise to a specific deadline. In the interest of protecting the conceptual advance provided by the work, we recommend a revision within 3 months (7th Jan 2024). Please discuss the revision progress ahead of this time with the editor if you require more time to complete the revisions. Use the link below to submit your revision:

IMPORTANT: When you send your revision, we will require the following items:

1. the manuscript text in LaTeX, RTF or MS Word format
2. a letter with a detailed description of the changes made in response to the referees. Please specify clearly the exact places in the text (pages and paragraphs) where each change has been made in response to each specific comment given
3. three to four 'bullet points' highlighting the main findings of your study
4. a short 'blurb' text summarizing in two sentences the study (max. 250 characters)
5. a 'thumbnail image' (550px width and max 400px height, Illustrator, PowerPoint or jpeg format), which can be used as 'visual title' for the synopsis section of your paper.
6. Please include an author contributions statement after the Acknowledgements section (see <https://www.embopress.org/page/journal/17444292/authorguide>)
7. Please complete the CHECKLIST available at (<https://bit.ly/EMBOPressAuthorChecklist>). Please note that the Author Checklist will be published alongside the paper as part of the transparent process (<https://www.embopress.org/page/journal/17444292/authorguide#transparentprocess>).
8. When assembling figures, please refer to our figure preparation guideline in order to ensure proper formatting and readability in print as well as on screen:

See also figure legend guidelines: <https://www.embopress.org/page/journal/17444292/authorguide#figureformat>

9. Please note that corresponding authors are required to supply an ORCID ID for their name upon submission of a revised manuscript (EMBO Press signed a joint statement to encourage ORCID adoption).

(<https://www.embopress.org/page/journal/17444292/authorguide#editorialprocess>)

Currently, our records indicate that the ORCID for your account is 0000-0002-8028-6138.

Link Not Available

The system will prompt you to fill in your funding and payment information. This will allow Wiley to send you a quote for the article processing charge (APC) in case of acceptance. This quote takes into account any reduction or fee waivers that you may be eligible for. Authors do not need to pay any fees before their manuscript is accepted and transferred to the publisher.

EMBO Press participates in many Publish and Read agreements that allow authors to publish Open Access with reduced/no publication charges. Check your eligibility: <https://authorservices.wiley.com/author-resources/Journal-Authors/open-access/affiliation-policies-payments/index.html>

*** PLEASE NOTE *** As part of the EMBO Press transparent editorial process initiative (see our Editorial at <https://dx.doi.org/10.1038/msb.2010.72>), Molecular Systems Biology publishes online a Review Process File with each accepted manuscripts. This file will be published in conjunction with your paper and will include the anonymous referee reports, your point-by-point response and all pertinent correspondence relating to the manuscript. If you do NOT want this File to be published, please inform the editorial office at msb@embo.org within 14 days upon receipt of the present letter.

RESPONSE TO REVIEWERS**Reviewer #1:**

In their manuscript entitled "Proteome-wide copy-number estimation from transcriptomics" Sweatt and co-workers develop and apply an approach to protect protein copy numbers from RNAseq data. To this end, they take advantage of existing TMT mass spec and matching RNAseq data of 375 cancer cell lines. By scaling these relative protein abundances to protein intensities derived from a SWATH experiment they arrive at protein abundance estimates across cell lines. Regression analysis of mRNA and protein abundance estimates allowed them to (i) select which of three alternative models (median, HL, HL+LASSO) describes the mRNA to protein relationship best and (ii) to determine gene-specific model parameters. They show that this approach can predict protein levels from RNAseq data better than a null model based on observed protein abundances only. Applying this approach to a mathematical model of coxsackievirus infection of cardiomyocytes provided meaningful results for the receptors CD55 and CXADR. Moreover, applying the method to breast cancer RNAseq data resulted in reclassification of RNA subtypes according to the inferred proteomic data.

Overall, this is a well-written paper that addresses a key question in gene expression control. A particular strength is that the approach presented is quite simple, well-described and makes intuitive sense. The finding that mRNA to protein relationships can often be described by a hyperbolic-to-linear model is interesting. Also, it is intriguing to see that LASSO identifies a number of "trans" relationships between mRNAs and proteins that make sense. These observations and the Pinferna predictions themselves will be valuable for the scientific community.

We thank Reviewer #1 for their provisional enthusiasm.

Having said this, I think the paper needs to be improved in a number of ways before it can become acceptable for publication in MSB.

1. Validation and application of Pinferna: The described application of Pinferna to modelling viral infection and breast cancer are interesting but not really convincing. What do we really learn from the "1489 systems-biology models of coxsackievirus B3 infection susceptibility"? Also, does the reclassification of breast cancer samples based on Pinferna-derived protein copy number estimates reflect reality better than the RNA-based classification? This is something the authors could actually test using existing matched RNAseq and proteomic datasets for breast cancer (<https://www.nature.com/articles/nature18003>).

The 1489 systems-biology models of coxsackievirus B3 infection predicted that the threshold for infectivity is ~5000 copies of the viral receptor CXADR per cell (Fig. 4G). In the revision, we engineered non-permissive cardiomyocytes to express ~5000 copies of CXADR per cell (quantified by immunoblotting) and found that it rendered these cells permissive to coxsackievirus B3 (Fig. 4J,K of the revision). When the same cells expressed only ~2800 copies of CXADR, they remained refractory to infection (Fig. 4H,I of the revision). These new experiments validate predictions of the systems-biology model and the protein inferences of Pinferna.

To compare matched RNAseq and proteomic datasets, we obtained the breast cancer dataset of (Mertins *et al*, 2016) suggested by Reviewer #1 along with an ovarian proteogenomic dataset from (Zhang *et al*, 2016). Relative abundance changes were compared by log₂ centering the RNAseq–proteomics data and Pinferna predictions for each gene to yield a fold-change value across samples in the dataset. Then, we quantified Euclidean distances between groups to evaluate how close predicted fold-change values were to measured values. For both datasets, Pinferna predictions were significantly closer to the TMT relative proteomics than were the RNAseq profiles (Appendix Fig. S2 of the revision). Additional evidence for the informativeness of the Pinferna-based breast classification is provided in the response to Point #4 of Reviewer #3.

2. There is a community-based assessment of different methods to predict protein levels from mRNA levels: <https://doi.org/10.1016/j.cels.2020.06.013> This is a paper the authors fail to mention. Similar to point 1 above it would be important to benchmark the performance of Pinferna relative to the best-performing model in the community-based resource.

Thank you for bringing the DREAM challenge to our attention (Yang *et al*, 2020). The goal of this challenge was to predict relative changes in protein abundance from relative changes in transcript abundance, which is different from Pinferna's prediction of absolute copy numbers per cell from RNA transcripts per million (TPM). Nevertheless, we devised a way to compare the DREAM challenge winner (ProteoEstimator) against Pinferna by centering each relative gene-specific prediction on a randomized measurement of the protein and calculating Δ CDF as described in the first submission. Fig. 3C–E of the revision now contains the Δ CDF performance estimates of ProteoEstimator (Yang *et al*, 2020), which are significantly worse than Pinferna in all three cases.

*3. There are a few misconceptions related to absolute protein quantification, DDA and DIA methods. The authors should rephrase these sections: Page 3: "Unfortunately, multiplex labeling yields peptide-specific relative quantities that cannot examine absolute differences among proteins within a sample (Pappireddi *et al*, 2019)." This and a few following statements indicate that the authors have a wrong*

perception of the fundamental principles of protein quantification by MS: Multiplexed labelling methods such as TMT (but also MS1 labelling methods like MTRAQ, SILAC, dimethyl etc) can provide accurate relative quantification of proteins across samples. These can be used to scale absolute protein abundance (for example, estimates obtained for a single sample) across samples. In fact, it is not difficult to use MS1 peptide intensities from TMT datasets to compute absolute protein abundance estimates. These estimates can then be scaled based on TMT-ratios, exactly the way the authors did for SWATH. "This difficulty is surmounted by data-independent acquisition methods like sequential window acquisition of all theoretical mass spectra (SWATH), which analyzes all precursor ions in a series of mass-to-charge ratio windows (Gillet et al, 2012). After data acquisition, the most sensitive peptide(s) of a protein are summed by intensity, and the resulting data are centered at a reasonable per-cell average (10⁴ copies) to yield absolute estimates of the detectable proteome. SWATH is newer, harder to adapt to different cell lineages, and lower throughput. Consequently, the leading proteomics repository contains ~tenfold fewer SWATH depositions than label-based depositions as of mid-2023 (Perez-Riverol et al, 2019). The need for commoditized SWATH-like protein estimates may forever outpace the ability to generate them directly." SWATH (or other DIA approaches) are not intrinsically superior for absolute protein quantification. Also, there is no such thing as a "most sensitive peptide" - peptides do not have sensitivities. Different peptides have different "flyabilities" in the mass spectrometer (reflecting different ionisation efficiencies etc.). Also, while it is true that SWATH datasets are underrepresented in repositories, I do not see the value in "commoditized SWATH-like estimates": Absolute protein abundance estimates can be obtained from mass spec data in a number of different ways (Top3, iBAQ, APEX etc.). This is true for both DIA and DDA datasets. Some of these algorithms perform better than others, a critical assessment of them can be found here: <https://doi.org/10.1002/pmic.201300135>. SWATH is not special. In fact, the authors might want to also consider iBAQ or Top3-based estimates available for over 10,000 proteins in proteomicsDB (<https://www.proteomicsdb.org/>).

The points by Reviewer #1 are well taken. We meant to convey that multiplex labeling does not directly provide information about absolute protein abundance. Also, apologies for the abuse of terminology—by “most sensitive”, we meant “most ionizable”. We have revised this passage in the Introduction as follows (key edits in magenta):

Multiplex labeling yields peptide-specific relative quantities, but other analytical methods are needed to give information about absolute differences among proteins within a sample (Ahrne et al, 2013; Pappireddi et al, 2019). Intensity-based quantification of MS1 precursor ions is possible across proteins within a sample, but stochasticity in the ions selected for peptide identification creates sparsity challenges when data from many runs must be combined. One robust alternative is to use data-independent acquisition methods like sequential window acquisition of all theoretical mass spectra (SWATH), which analyzes all precursor ions in a series of mass-to-charge ratio windows (Gillet et al, 2012). After data acquisition, the best-ionizing peptide(s) of a protein are summed by intensity, and the resulting data are centered at a reasonable per-cell average (10⁴ copies) to estimate absolute copy numbers for the detectable proteome. SWATH is more reproducible but also more computationally intensive, harder to set up, and lower throughput (Collins et al, 2017; Ludwig et al, 2018). The need for absolute protein estimates may forever outpace the ability to generate them directly.

We appreciate the information about proteomicsDB and have used this resource as an orthogonal validation in response to Point #3 of Reviewer #3 (Lautenbacher et al, 2022).

4. The authors tend to oversimplify the complexity of mRNA and protein correlations. Page 6, "Linear mRNA-protein relationships adequately recapitulate protein expression among genes within a sample, but they are poor at distinguishing protein differences among samples for any given gene (Buccitelli & Selbach, 2020; Fortelny et al, 2017; Tasaki et al, 2022; Wilhelm et al, 2014)." First, proteins cannot be expressed - only genes can. The authors should write "protein abundance differences". Second, this implies that across-gene within sample mRNA-to-protein correlations are always higher than within gene across sample mRNA-to-protein correlations. This is not the case. Page 6 "The current thinking is that the steady-state abundance of mRNA and its intrinsic translation rate create a general "set point" for protein expression, which is buffered or tuned according to the abundance of complexes that stably contain the protein (Buccitelli & Selbach, 2020; Taggart et al, 2020)." Protein-level buffering is a more widely observed phenomenon that is not only observed for members of multiprotein complexes. Therefore, while it is correct that "our working inventory of protein-protein interactions and stable complexes in mammalian cells is far from saturation", stating that this "has thus far prevented a bottom-up reconstruction of mRNA-to-protein relationships" is an oversimplification: There are many other factors involved - protein-protein interactions is just one of them.

We have revised this passage to provide a more nuanced description of mRNA-protein correlations as follows (key edits in magenta):

Protein abundance differences may track somewhat linearly with mRNA when considering all genes within a sample, but it is often difficult for mRNA to predict quantitative protein differences among samples for any given gene (Buccitelli & Selbach, 2020; Fortelny et al, 2017; Tasaki et al, 2022; Wilhelm et al, 2014). The latter is important for systems biology when using transcriptome profiles to instantiate personalized models of function (Lewis et al, 2021; Montagud et al, 2022; Pereira et al, 2020). The current thinking is that the steady-state abundance of mRNA and its translation rate create a general "set point" for protein copies, which are buffered or tuned according to the protein's characteristics and its (de)stabilizing interactions with other proteins in a cell context (Buccitelli & Selbach, 2020; Taggart et al, 2020). Unfortunately, our working knowledge of these characteristics and interactions in mammalian cells remains incomplete (Giurgiu et al, 2019; Richards et al, 2021), which has thus far prevented a bottom-up reconstruction of mRNA-to-protein relationships that are absolute and conditional.

5. Page 7, Accuracy test (Fig 3 A): I agree with the conclusions drawn by the authors. However, I think it is important to clearly state

that the improvement of Pinferna compared to the null models is modest.

We are happy that Reviewer #1 agrees with the evidence of our assertions. In addition, we would like to expand upon how Δ CDF should be interpreted, as it is a new metric. No matter the method, we find that 20–35% of genes will be predicted accurately and ~10% of genes will be predicted inaccurately in a given sample (Fig. 3A,B). Methods distinguish themselves by the 60% of genes in the middle, which range in absolute performance ($|\text{Scaled residual}|$) from about 0.5–3. Approximating the median null as an elliptical arc over that range, the maximum theoretical Δ CDF is $(0.9 - 0.3)(3 - 0.5)(1 - \pi/4) = 0.32$. Thus, the performance of Pinferna (Δ CDF = 0.15–0.2) is >50% of the way to the practical maximum. We have edited the text as follows:

Although modestly positive, the Δ CDF value for Pinferna was >50% of the practical maximum for this metric (Appendix Text S1).

and added Appendix Text S1 that provides more explanation of the Δ CDF for readers.

Additional point:

- Page 3: "Isobaric labeling approaches such as tandem mass tagging (TMT) now quantify 11+ multiplex samples and are the method of choice for proteogenomics (Ellis *et al*, 2013; Thompson *et al*, 2003)." This statement is outdated. TMT 18-plex is now available.

Thank you, we have revised the sentence as follows (key edits in magenta):

Isobaric labeling approaches such as tandem mass tagging (TMT) now quantify up to 18 multiplex samples and are the method of choice for proteogenomics (Ellis *et al*, 2013; Li *et al*, 2021; Thompson *et al*, 2003).

Reviewer #2:

In the manuscript by Sweatt et al., the relationship between proteomics and RNAseq data is explored. The authors introduce a model aimed at predicting protein abundance from RNAseq and subsequently apply this model to different systems. While the effort is commendable, I believe the manuscript necessitates significant revisions before being suitable for publication in any journal. I doubt that reasonable efforts can modify this manuscript to meet the standards of journals like MolSysBio. I have serious reservations about the model's technical implementation and its predictions. The model's overall utility remains ambiguous. Merely applying it to different systems without proper validation seems inadequate.

The manuscript has undergone significant revisions to assuage the core concerns of Reviewer #2. The perceived deficiencies in “proper validation” came from our misunderstanding about the scope of a Method article for *Molecular Systems Biology*. Before submitting, we read the Instructions for Authors:

Method articles describe new methods that will be relevant to a broad audience and represent a clear advance over existing methodologies. New methods should have a clear potential to generate novel discoveries or open new areas of research. Estimations of performance, accuracy and systematic benchmarking should be included when appropriate.

and consulted with four recent Method articles available at the time of submission (Bachman *et al*, 2023; Cuomo *et al*, 2022; Kamal *et al*, 2023; Martin *et al*, 2022). None of these resources indicated to us that novel discoveries must be secondarily validated by methods other than the one reported in the manuscript. Our impression was that new methods needed to be compared rigorously against competing alternatives using a validated reference. For Pinferna, the validated references were three SWATH-MS datasets that were entirely separate from the training data (Fig. 3C–E). The potential to generate novel discoveries was illustrated by applying Pinferna to kinetic modeling of virus infection (Fig. 4) and cancer subtyping (Fig. 5). We are certainly not opposed to doing such secondary validation and include it in the revised manuscript, as described below.

Major concerns:

1. *Data Integration: The choice of integrating SWATH data as executed in the study is perplexing and might adversely affect the overall analysis. It's known that absolute abundance data can be directly procured from TMT measurements by quantifying MS1 intensities and then partitioning these intensities based on relative quantification from TMT signals. This method isn't commonly pursued because MS intensities show only weak correlations with absolute protein abundances - a characteristic also seen with SWATH data. Nevertheless, the integration of SWATH data with multiplexed data has potential advantages: A) To clarify the relationship between MS1 intensity and protein abundance in DDA experiments, as the association isn't linear (e.g., Schwanhueser et al.). Still, it might be more effective to use internal standards since the linearity of this relationship in SWATH data is also questionable. B) To address the limited dynamic range of multiplexed proteomics experiments, which SWATH data might overcome, especially for large protein abundance changes.*

We agree with Reviewer #2 that the best absolute measurements by mass spectrometry involve SRM/MRM-type studies with internal standards, but these do not scale to the proteome. However, the comment about the linearity or reproducibility of SWATH-MS is puzzling to us given our reading of the field. A highly influential paper by the Aebersold group along with ten other laboratories showed that SWATH-MS was linear over ~4.5 orders of magnitude with an inter-site coefficient of variation of ~20% (Collins *et al*, 2017). We confirmed this performance (as well as that of TMT) in Fig 1B, where SWATH-MS data from CAL51 cells (PXD003278) were used to scale the CCLE-wide TMT dataset (Nusinow *et al*, 2020). The aggregated SWATH-scaled TMT proteome of U2OS cells from (Nusinow *et al*, 2020) was then compared with an independent SWATH-MS dataset of U2OS cells (PXD000954), and the concordance was excellent ($R = 0.77$, $\rho = 0.76$; Fig. 1B). This introductory passage has been revised in response to Point #3 of Reviewer #1.

2. *It is unclear to what extent the developed model learns the relationship between protein and mRNA abundance changes versus between mRNA and protein measurement artifacts. e.g., the used TMT data typically has an inherent <100x dynamic range. Shown ratios from one TMT experiment above 50x are essentially meaningless. At the same time, much of the RNAseq data shown seems to fall in the very low regime, e.g., in figure 2D, where sampling noise dominates. Data lacking these artifacts, e.g., SWATH or very deep RNAseq, should be used to evaluate and mitigate.*

This is an excellent point that we would like to address from multiple angles. First, the originating experiments—TMT ten-plex data on 375 CCLE cells lines were collected in groups of nine with a “bridge” sample comprised of 11 pooled cell lines (Nusinow *et al*, 2020). The aggregated dataset may therefore have an apparent dynamic range of greater than 50–100 if, for example, one sample is 20-fold more abundant than the bridge and another sample (in a separate run) is 20-fold less abundant than the bridge. To gain a sense of how prevalent this situation might be, we evaluated the dynamic-range statistics of the SWATH-scaled TMT proteome (Dataset EV4). 75% of all genes in the TMT training set have a min-max dynamic range within 50x

and 85% of them have a min-max dynamic range within 100x. The number of genes jumps to 98.5% within 50x and 99.7% within 100x when considering the ratio of the 5th and 95th percentiles of the training data. Thus, even if a handful of observations on the extremes were deemed problematic, they would not substantively impact the Pinferna calibration (Fig. 1A, Steps 2 and 3).

Figure 2. Reprinted subpanels (E,F) of the revision.

Overall, we noted excellent concordance between the TMT estimates and the immunoblots, although there was some evidence of ratio compression for one cell line with the lowest SERPINB6 (Fig. 2E,F in the revision; reprinted left). We have revised the text as follows (key edits in magenta):

More common were log-concave up HL relationships in which protein abundance increased only at higher mRNA abundances (Fig. 2D). *Some of this behavior was attributable to the ratio compression that occurs when low-abundance proteins are quantified by TMT (Savitski et al, 2013), which we confirmed by quantitative immunoblotting (Fig. 2E,F).*

3. *The manuscript fails to provide a convincing demonstration of the model's usefulness. In principle, there is value in developing the kind of model the authors describe. However, the model uses many parameters; just slightly beating simple alternatives is not particularly exciting. To make this an interesting study, I would like to see some strong, non-obvious predictions that are rigorously tested.*

We address the “slightly beating simple alternatives” concern in the response to Point #5 of Reviewer #1 above and Appendix Text S1 of the revision. We adopted a two-pronged plan for “strong, non-obvious predictions,” one related to kinetic modeling of virus infection (Figs. 4H–K and EV4I–L of the revision) and the other related to determinants of breast cancer subtyping (new Fig. 6 of the revision). The results for kinetic modeling of virus infection are described in the response to Point #22 of Reviewer #3 below; we describe Fig. 6 here.

The initial assessment of HL+LASSO models (Fig. 3G,H of the revision) supported that LASSO features of a gene contained biological meaning, but it only implied that some of these features (i.e., other genes) were causative. We built off the CDK4 observation and its densely connected network of known interactions appearing as LASSO features (Fig. 5F). The existing CDK4 interaction mechanisms captured by Pinferna are already diverse. E2F4 is a putative substrate of CDK4 and provides a positive LASSO coefficient possibly by protein stabilization (Scime et al, 2008). Conversely, CCND1 provides a negative LASSO coefficient and is a direct protein-protein binding partner with a short half-life that increases CDK4 turnover (Matsushima et al, 1992). TSPAN31 also adds a negative LASSO coefficient but appears to do so by a miRNA-like process, whereby it acts as an antisense transcript binding the 3' UTR of CDK4 (Xia et al, 2020). The LASSO features of Pinferna thus contain diverse mechanisms of protein-abundance regulation.

In the revision, we tested two unexpected genes among the LASSO features for CDK4: the nucleoporin NUP37 (negative coefficient) and the mitochondrial dynamin-like GTPase DNM1L (positive coefficient; Dataset EV5). Using NUP37-engineered breast epithelial cells from a prior study (Wang et al, 2023) as well as a luminal breast cancer line engineered for this revision, we confirmed that NUP37 abundance inversely affects CDK4 abundance (Fig. 6A–F of the revision). Teaming up with local DNM1L expert (and revision co-author) David Kashatus, we pursued analogous perturbations of DNM1L in different luminal breast cancer settings. Achieving substantive increases in DNM1L proved impossible because of dosage compensation from endogenous DNM1L (Appendix Fig. S7 of the revision). However, using a validated shDNM1L hairpin from the Kashatus lab (Kashatus et al, 2015; Nagdas et al, 2019), we achieved two- to fivefold reduction and found that the extent of DNM1L knockdown correlated positively with CDK4 (Fig. 6G,H of the revision). There is no obvious precedent for a cyclin-dependent kinase (CDK4) residing downstream of a nucleoporin (NUP37) or a mitochondrial GTPase (DNM1L), making these LASSO predictions particularly stringent.

Reviewer #3:

Sweatt and co-authors present a computational model (Pinferna) for predicting absolute protein levels (protein copy numbers) from mRNA levels. Three types of models are fitted for each protein: just using the median (M), fitting a non-linear model using the encoding mRNA of the protein as a predictor (HL) and by additionally accounting for transcripts of other genes as predictors (HL-LASSO). The authors test Pinferna on a range of datasets with matched transcriptomics and proteomics datasets. Having a tool to predict protein levels from mRNA levels is definitely helpful. This work is timely and relevant. However, before publication a number of points need to be addressed.

We thank Reviewer #3 for their provisional enthusiasm.

Major points

1. As far as we can see the comparison to competing tools is unfair. Pinferna works on calibrated median protein levels that match to the data that is being used in this study. This is important for predicting absolute protein levels. Competing tools aim to predict the variation of protein levels between conditions or samples rather than absolute levels. Therefore, they also first need to be calibrated accordingly. The fact that competing methods perform significantly(!) worse than random guessing is suspicious (Figure 3). The competitors performance is significantly worse than using the protein medians. While this shows that the models are not calibrated to absolute protein levels, it indicates also that they contain information that is anticorrelated with the actual levels which might be informative.

We apologize for the confusion and must clarify a couple points of misunderstanding. First, although Pinferna is calibrated to an average absolute abundance measurement from SWATH-MS of two cell lines (U2OS and CAL51; Fig EV1A), it has no SWATH-MS information about the test data (Fig 3), which are normalized independently. Second, we selected PaxDb and PTR as competing methods because they claim to make absolute protein predictions (Eraslan *et al*, 2019; Wang *et al*, 2015). They were calibrated similarly to the median copy number for the proteins quantified in both the TMT-SWATH meta-assembly and the competing method (Methods). Third, although the first submission originally avoided competing tools that predict variation of protein levels, the revision adds a best-performing tool in this category (ProteoEstimator), which was calibrated to the median null (Methods). The results of this comparison are described in Response #2 to Reviewer #1 and Fig. 3C–E of the revision.

Fourth, Reviewer #3 is correct that a below-median Δ CDF value indicates a gross bias in the predictions, which is exactly what we wanted to drive home with the comparisons in Fig. 3. Because PaxDb and PTR were calibrated identically to SWATH-MS, it implies that the aggregate distribution of thousands of predicted proteins is skewed or displaced from what it should be. However, it is not true that a negative Δ CDF indicates an anti-correlation like a Pearson or Spearman statistic. A set of predictions could be highly correlated with measured values, but if the slope of the relationships is far from one, the Δ CDF will be poor. Reciprocally, a cloud of predictions and observations (each with 10% error) would have a poor correlation, but the Δ CDF would be high because the errors are small relative to the absolute value itself. To clarify, we added graphical displays of how a |Scaled residual| CDF changes under different error regimes (Fig. EV3A,B of the revision), and we expanded upon the characteristics of Δ CDF under limiting cases (Appendix Text S1 of the revision).

2. Related: for most applications in biology it is much more important to correctly predict the (relative) differences in protein levels between conditions (e.g. healthy versus disease, ko versus wild-type, etc.). Pinferna focuses on absolute levels, which is legitimate. But the authors should better investigate how well Pinferna also predicts protein level variation.

We appreciate that Reviewer #3 sees legitimate value in predicting absolute copy numbers. To assess performance by relative differences, we \log_2 centered each dataset (PTR, Pinferna, SWATH-MS) by gene to yield fold-change values for each gene relative to other samples in the dataset. We then calculated the Euclidean distances between PTR or Pinferna predictions and the measured SWATH-MS values for each sample to test if the fold-change value is similar between the predicted and measured values. Fold-change profiles (across all genes) for Pinferna were significantly closer to the SWATH-MS ground truth for all test cases (Appendix Fig. S1 of the revision).

3. Some of the claims are simply not supported by sufficient evidence. For example, in the analysis of coxsackievirus infections the authors write about changes in infection time and rate as if they had actually observed those differences in human subjects. However, in reality these are just simulations that may or may not be correct. Likewise with the breast cancer study: wordings like "CTNNA2 protein was ubiquitous in the meta-assembly" are misleading, because these are predictions that could not be validated.

We expand upon the coxsackievirus B3 (CVB3) infection model in the response to Point #22 below, which

provides additional evidence for the claim. Regarding CTNNA2, we agree with Reviewer #3 that we did not validate this assertion, because it was not an interpretive claim of a Pinferna prediction but rather a description of the TMT data that we started with. We revised this passage to read, “was **measured ubiquitously**” and add that when we consulted ProteomicsDB—a resource recommended by Reviewer #1 (Lautenbacher *et al*, 2022)—we find independent evidence that CTNNA2 protein is detectable across all cell lines assembled (Fig. R1).

Figure R1. Screenshot of CTNNA2 protein abundance among cell lines quantified by MS1-iBAQ and aggregated in ProteomicsDB.

4. Using their model the authors reclassify a number of breast cancer samples. They should provide some evidence that this reclassification actually improves the grouping of cancers. For example, are the survival times of the new grouping more consistent than before? E.g. the authors could show Kaplan-Meier Curves to address this question.

We thank Reviewer #3 for this excellent suggestion. Focusing on the two luminal subtypes with the largest number of reassignments, we stratified cases based on whether they had been reclassified by Pinferna or not. Strikingly, the Pinferna-reclassified cases of Luminal A breast cancer had significantly worse outcomes, suggesting that RNA-seq had misassigned them to the low-risk Luminal A group. This analysis is included in Appendix Fig. S4 of the revision.

Minor points:

5. p3, introduction: ‘but mapping the transcriptome to the abundance of proteins is complex’ is a weird phrasing

We have revised this passage to read, “but **relating transcript copies** to the abundance of proteins is complex”.

6. p3: ‘The need for commoditized SWATH-like protein estimates may forever outpace the ability to generate them directly.’ A little pessimistic?

This sentence was slightly edited in the response to Point #3 to Reviewer #1. Of course, we dream of a day when quantitative mass spectrometry is as widely used and as accessible as RNA-seq, but we must also plan for the possibility that such a day will never come. For reference, we include a trajectory of human RNA-seq and TMT- or SWATH-based mass spectrometry depositions over the past decade (Fig. R2). The two orders-of-magnitude separation between transcriptomics and proteomics shows no signs of abating.

Figure R2. Public depositions of RNA-seq and TMT or SWATH mass spectrometry since 2013.

7. p4 “estimating absolute copy numbers from mRNA is historically fraught with uncertainty” This is a very florid way of saying “it is difficult”.

We have simplified the passage to read, “estimating absolute copy numbers from mRNA is **challenging**”.

8. p4: Need to define what they mean with ‘intrinsic translation rate’.

We were simply referring to the translation rate of that mRNA and have removed “intrinsic” in the revision.

9. p5 LASSO: are only coding transcripts (mRNA) considered here? Or also non-coding transcripts and if yes, which ones?

We apologize for the ambiguity. In the revised Appendix Supplementary Methods, we now clarify that:

Residuals from the HL fit were regressed against all other coding genes in the MANE-harmonized transcriptome (Dataset EV3) by LASSO...

A future long-term goal is to enhance Pinferna with miRNA expression and seed sequences of genes to determine if performance is significantly improved over that reported here.

10. p5: Discussion of M-proteins is too short. I would like to see a thorough discussion of the different types of M, HL and HL-LASSO proteins. E.g. what do the GO enrichments of all three classes look like? Does it biochemically make sense that the respective types of proteins end up in those classes?

We provided the GO enrichments for all categories in Dataset EV6 of the original submission. We did not expand on the other enrichment categories because HL and HL+LASSO categories comprise such large fractions of the Pinferna proteome (59% and 32% respectively). Those enrichments in Dataset EV5 of the revision likely reflect proteins reliably detected by TMT and SWATH-MS rather than the subcategories themselves.

11. p5 bottom: Taggart et al. 2020: also cite Goncalves et al. 2017, PMID 29032074

Thank you, we have appended this citation.

12. p6: 'HL accommodated rare log-concave down relationships that occurred when protein abundance saturated at high transcript abundances ($p < 10^{-15}$; Fig 2C).' What does the pvalue refer to?

Our apologies—this p value carried over from an early draft where we tested whether log-concave down genes had significantly higher abundance than other HL genes. We have removed this p value.

13. p6: 'Loss of transcript dependence arises biologically when a protein subunit surpasses the abundance of the complex in which it resides (Taggart et al, 2020; Wuhr et al, 2014), as observed for M genes across their entire measured range of mRNA.' Not only then. In fact, many of the RNA-protein relationships modelled by HL-LASSO may be due to such stoichiometric constraints.

We agree but mentioned only M genes in this passage because HL+LASSO had not yet been similarly discussed in the text yet. To simplify the revision, we deleted “as observed for M genes across their entire measured range of mRNA”.

14. p6: 'Specific RPS- or RPL-prefixed transcripts' The encoded proteins are members of the small and large subunits, the names are not the important part.

We have revised this passage to read, “Specific subunits of the ribosome and proteasome were also enriched...”

15. p6: ' $p \ll 10^{-4}$ ' just write the p-value accurately

This p value was estimated computationally from 10^4 random draws of the STRING database. Given the null distribution (Fig. 2H in the revision), it would require a prohibitive number of randomizations to estimate the p value exactly. We believe that the current number of randomizations is sufficient to reach the conclusion that the number of observed STRING interactors is highly unlikely to arise by chance (Huber, 2019). We have clarified in the text that the p value is estimated.

16. p7: 'To be useful for new samples, gene-specific model predictions should be more accurate than guesses based on past copy-number estimates of the protein in other settings.' It also has to be better than using 1:1 relationships with all mRNAs of one sample, i.e. assuming the differences between protein levels are the same as between transcript levels or having one reference level per gene for both layers.

In the first submission, we tested for linear relationships by calibration and found they were strongly disfavored

Figure R3. Δ CDF performance of a 1:1 relationship between mRNA TPM and protein abundance. TPM distributions were centered at the same median protein copies as the sample tested.

all observed measurements of one given protein. Thereby they maintain all distributional properties of the protein, especially the mean, median and variance. If this is what the authors did, they should also describe it as such. Otherwise better explain.

Yes, we struggled with a concise term to describe these null simulations (e.g., “randomized measurement” was suggested by one of our internal reviewers after they did not like our first choice of “empirical guessing”). Reviewer #3 is correct that “randomized measurement” is a random sample taken from all observed measurements of one given protein. We have revised this passage in the text as follows (key edits in magenta):

Therefore, we assessed Pinferna predictions against a null model of “randomized measurements” built by randomly selecting an abundance for each protein from measurements of that protein in the meta-assembled dataset originally used for training (Fig. 1A).

We agree with Reviewer #3 that it is important to retain the distributional properties of each protein.

18. p7: "Accuracy estimates were comparable when reported protein abundances were used instead of protein re-quantifications performed exactly as done for the training data". I don't understand this sentence. What means 'reported abundances'? If these were the actual measurement the accuracy should be perfect and the analysis trivial. And what means "protein re-quantifications performed exactly as done for the training data"? Does it mean they applied their model? I have no idea what this sentence is trying to convey.

We should have been more explicit here. “Reported protein abundances” are the processed quantitative data taken directly from the publication or deposition. “Protein re-quantifications” were from raw data files using the latest quantification methods described in the paper. We have revised this passage in the text as follows (key edits in magenta):

Accuracy estimates were comparable when using the reported post-processed protein abundances of (Liu *et al*, 2019) instead of their raw data quantified with the updated analysis pipeline of the meta-assembly (Methods; Fig. EV3C,D).

Package versions get updated with different dependencies, yielding numbers that are close but not identical. Our point is that these small differences (both for public RNA-seq data and SWATH-MS data) have a negligible impact on the performance of Pinferna.

19. p7: 'The results bolster recent claims that typical single-cell RNA-seq data sequenced at ~50,000 reads per cell poorly reflect protein abundances (Brunner et al, 2022; Reimegard et al, 2021) and separately indicate that Pinferna's bulk predictions of protein from mRNA are robust to algorithmic details.' This statement is too broad. For sure some proteins can be accurately quantified/predicted.

We have revised this passage in the text as follows (key edits in magenta):

The results bolster recent claims that typical single-cell RNA-seq data sequenced at ~50,000 reads per cell incompletely reflect the proteome (Brunner *et al*, 2022; Reimegard *et al*, 2021)

The reason is that TPM estimates become highly unstable when calculated from so few reads. Unstable TPM values yield inaccurate inferences from the Pinferna models.

20. p7: PTR with proper calibration might outperform pinferna in the prostate data.

compared to HL (Fig. EV2D). Also, PTR can be thought of as a gene-specific linear relationship for the entire proteome (Fig. 3C–E). In addition, here we tested a direct 1:1 relationship between mRNA TPM and protein (Fig. R3); as expected, the predictor performed poorly for all comparisons in Fig. 3C–E.

17. p7: The sampling process should be better explained. The authors frequently use the term 'randomized measurements', but I think they did not actually randomize any values. If I understood the procedure they actually performed a random sampling of the measurements, by randomly sampling from

We removed the notion of best-case calibration by looking at fold-change values between PTR and Pinferna. For both the prostate data and the NCI-60 samples, Pinferna is closer by log₂-centered Euclidean distance than either the general or tissue-specific implementations of PTR (Appendix Fig. S1).

21. p9: 'Using the Pinferna estimates of CD55 and CXADR as initial conditions, we created a series of individualized model variants to create a virtual cohort of the human population.' I have no idea what this means.

Apologies that we were too terse in the first submission. The model of CVB3 infection takes (among other inputs) the abundance of cellular CD55 and CXADR as the starting point for an *in silico* infection (Lopacinski *et al*, 2021). We used Pinferna to convert heart RNA-seq TPMs of CD55 and CXADR to copies per cell and then initialized 1489 different *in silico* models for CVB3 infection based on the inferred CD55–CXADR protein abundances. We have revised this passage in the text as follows (key edits in magenta):

We re-parameterized an existing model of CVB3 infection (Lopacinski *et al*, 2021) with the Pinferna estimates of CD55 and CXADR to instantiate model variants that simulate a human population.

Model outcomes are meant to approximate the diversity of responses predicted based only on inferred protein differences in CD55 and CXADR in this human-derived “virtual” population (Laubenbacher *et al*, 2021).

22. p12: "individuals with fewer than ~5000 copies of CXADR per cell were not susceptible to CVB3" You don't know this. This is just a model prediction. See also my point 3 above.

We took this comment from Reviewer #3 as an opportunity to address a broader concern raised in Point #3 of Reviewer #2 that non-obvious predictions were not rigorously tested in the manuscript. To tackle the claim that the susceptibility to CVB3 infection occurs at ~5000 copies of CXADR per cell, we turned to a cardiomyocyte-derived cell line (AC16) that is not permissive for CVB3 infection unless CXADR (also known as CAR) is overexpressed (Shah *et al*, 2017). Using epitope-tagged CXADR calibrated with recombinant standards, we found that parental AC16 cells express ~850 copies of CXADR per cell (Fig. 4I,K of the revision). We engineered AC16 cells with a doxycycline-inducible CXADR (iCAR) that provided temporal controls over the abundance of the viral receptor. When iCAR was induced for 30 minutes, we quantified ~2800 copies of CXADR per cell and no detectable viral infection compared to controls (Figs. 4H,I and EV4I,J of the revision). However, at 60 minutes of iCAR induction, we quantified 4780 ± 670 copies of CXADR per cell and significant increases in viral protein synthesis indicative of early infection (Figs. 4J,K and EV4K,L of the revision). These experiments bolster the model prediction, the claim about viral susceptibility, and the quantity of non-obvious predictions that are rigorously tested (see also Fig. 6 and our response to Point #3 of Reviewer #2).

23. p12 last paragraph: The analogy with bootstrapping does not work. Bootstrapping works by sampling the existing data. Here, the authors propose to use additional (independent) data to improve protein predictions and in this case even different types of data (RNA instead of protein).

Very well, we have revised the concluding paragraph as follows (key edits in magenta):

RNA-seq delivers more than transcript counts when combined with specialized analytical methods (Gao *et al*, 2021; La Manno *et al*, 2018; Newman *et al*, 2019). Pinferna illustrates how transcriptomics can extend the reach of proteomics in a data-driven way by informing more biological samples, including retrospective ones it would never otherwise have access to.

24. Materials and Methods: add the supplementary methods here as well. I don't see any point in splitting them off in a separate document.

We were following the *Molecular Systems Biology* Instructions for Authors page:

In cases where detailed methods cannot be described within the length limits of the article, additional Materials and Methods can be included as part of the Appendix.

We will defer to the Editor about combining the Methods into one place. Currently, Methods that only pertain to Appendix or Extended View figures are included in the Appendix.

25. p15: GO methods need to be much more detailed

We have expanded the Methods as follows (key edits in magenta):

Gene ontology analysis

Enrichments of M, HL, and HL+LASSO genes for biological processes were evaluated with the GO knowledgebase (Gene Ontology *et al*, 2023). Genes in each class (Dataset EV5) were separately tested on the GO Enrichment Analysis web page (<https://geneontology.org/>) and searched for “Biological Process” at a false-discovery rate of 5%. The complete list of enrichments is available in Dataset EV6.

26. *Figure 2c: Why is the difference between Pinferna and the median less significant than the comparison of Pinferna versus PTR even though the mean differences are bigger? (similar other cases exist)*

We thank Reviewer #3 for bringing this to our attention. The Fig. 2 caption of the first submission stated that differences between groups were assessed by rank-sum test. Comparisons with median nulls involve many tied ranks, requiring an approximation of the statistic rather than the exact calculation, which is possible with other comparisons in the subpanel. Considering that predictions from each method are paired by sample, we revised the statistical analysis and changed to a sign-rank test that will not have the same complication. The p values will still not be proportional to the perceived effect size because the sign-rank test is a nonparametric method assessing directional differences. However, it will avoid complications with ties and the overall conclusions are unaltered.

References

- Ahrne E, Molzahn L, Glatter T, Schmidt A (2013) Critical assessment of proteome-wide label-free absolute abundance estimation strategies. *Proteomics* **13**: 2567-2578
- Bachman JA, Gyori BM, Sorger PK (2023) Automated assembly of molecular mechanisms at scale from text mining and curated databases. *Mol Syst Biol* **19**: e11325
- Brunner AD, Thielert M, Vasilopoulou C, Ammar C, Coscia F, Mund A, Hoerning OB, Bache N, Apalategui A, Lubeck M, Richter S, Fischer DS, Raether O, Park MA, Meier F, Theis FJ, Mann M (2022) Ultra-high sensitivity mass spectrometry quantifies single-cell proteome changes upon perturbation. *Mol Syst Biol* **18**: e10798
- Buccitelli C, Selbach M (2020) mRNAs, proteins and the emerging principles of gene expression control. *Nat Rev Genet* **21**: 630-644
- Collins BC, Hunter CL, Liu Y, Schilling B, Rosenberger G, Bader SL, Chan DW, Gibson BW, Gingras AC, Held JM, Hirayama-Kurogi M, Hou G, Krisp C, Larsen B, Lin L, Liu S, Molloy MP, Moritz RL, Ohtsuki S, Schlapbach R et al (2017) Multi-laboratory assessment of reproducibility, qualitative and quantitative performance of SWATH-mass spectrometry. *Nat Commun* **8**: 291
- Cuomo ASE, Heinen T, Vagiaki D, Horta D, Marioni JC, Stegle O (2022) CellRegMap: a statistical framework for mapping context-specific regulatory variants using scRNA-seq. *Mol Syst Biol* **18**: e10663
- Ellis MJ, Gillette M, Carr SA, Paulovich AG, Smith RD, Rodland KK, Townsend RR, Kinsinger C, Mesri M, Rodriguez H, Liebler DC, Clinical Proteomic Tumor Analysis C (2013) Connecting genomic alterations to cancer biology with proteomics: the NCI Clinical Proteomic Tumor Analysis Consortium. *Cancer Discov* **3**: 1108-1112
- Eraslan B, Wang D, Gusic M, Prokisch H, Hallstrom BM, Uhlen M, Asplund A, Ponten F, Wieland T, Hopf T, Hahne H, Kuster B, Gagneur J (2019) Quantification and discovery of sequence determinants of protein-per-mRNA amount in 29 human tissues. *Mol Syst Biol* **15**: e8513
- Fortelny N, Overall CM, Pavlidis P, Freue GVC (2017) Can we predict protein from mRNA levels? *Nature* **547**: E19-E20
- Gao R, Bai S, Henderson YC, Lin Y, Schalck A, Yan Y, Kumar T, Hu M, Sei E, Davis A, Wang F, Shaitelman SF, Wang JR, Chen K, Moulder S, Lai SY, Navin NE (2021) Delineating copy number and clonal substructure in human tumors from single-cell transcriptomes. *Nat Biotechnol*
- Gene Ontology C, Aleksander SA, Balhoff J, Carbon S, Cherry JM, Drabkin HJ, Ebert D, Feuermann M, Gaudet P, Harris NL, Hill DP, Lee R, Mi H, Moxon S, Mungall CJ, Muruganugan A, Mushayahama T, Sternberg PW, Thomas PD, Van Auken K et al (2023) The Gene Ontology knowledgebase in 2023. *Genetics* **224**
- Gillet LC, Navarro P, Tate S, Rost H, Selevsek N, Reiter L, Bonner R, Aebersold R (2012) Targeted data extraction of the MS/MS spectra generated by data-independent acquisition: a new concept for consistent and accurate proteome analysis. *Mol Cell Proteomics* **11**: O111.016717
- Giurgiu M, Reinhard J, Brauner B, Dunger-Kaltenbach I, Fobo G, Frishman G, Montrone C, Ruepp A (2019) CORUM: the comprehensive resource of mammalian protein complexes-2019. *Nucleic Acids Res* **47**: D559-D563
- Huber W (2019) Reporting p Values. *Cell Syst* **8**: 170-171
- Janes KA (2015) An analysis of critical factors for quantitative immunoblotting. *Sci Signal* **8**: rs2

Kamal A, Arnold C, Claringbould A, Moussa R, Servaas NH, Kholmatov M, Daga N, Nogina D, Mueller-Dott S, Reyes-Palomares A, Palla G, Sigalova O, Bunina D, Pabst C, Zaugg JB (2023) GRaNIE and GRaNPA: inference and evaluation of enhancer-mediated gene regulatory networks. *Mol Syst Biol* **19**: e11627

Kashatus JA, Nascimento A, Myers LJ, Sher A, Byrne FL, Hoehn KL, Counter CM, Kashatus DF (2015) Erk2 phosphorylation of Drp1 promotes mitochondrial fission and MAPK-driven tumor growth. *Mol Cell* **57**: 537-551

La Manno G, Soldatov R, Zeisel A, Braun E, Hochgerner H, Petukhov V, Lidschreiber K, Kastrioti ME, Lonnerberg P, Furlan A, Fan J, Borm LE, Liu Z, van Bruggen D, Guo J, He X, Barker R, Sundstrom E, Castelo-Branco G, Cramer P et al (2018) RNA velocity of single cells. *Nature* **560**: 494-498

Laubenbacher R, Sluka JP, Glazier JA (2021) Using digital twins in viral infection. *Science* **371**: 1105-1106

Lautenbacher L, Samaras P, Muller J, Grafberger A, Shraideh M, Rank J, Fuchs ST, Schmidt TK, The M, Dallago C, Wittges H, Rost B, Krcmar H, Kuster B, Wilhelm M (2022) ProteomicsDB: toward a FAIR open-source resource for life-science research. *Nucleic Acids Res* **50**: D1541-D1552

Lewis JE, Forshaw TE, Boothman DA, Furdai CM, Kemp ML (2021) Personalized Genome-Scale Metabolic Models Identify Targets of Redox Metabolism in Radiation-Resistant Tumors. *Cell Syst* **12**: 68-81 e11

Li J, Cai Z, Bomgarden RD, Pike I, Kuhn K, Rogers JC, Roberts TM, Gygi SP, Paulo JA (2021) TMTpro-18plex: The Expanded and Complete Set of TMTpro Reagents for Sample Multiplexing. *J Proteome Res* **20**: 2964-2972

Liu Y, Mi Y, Mueller T, Kreibich S, Williams EG, Van Drogen A, Borel C, Frank M, Germain PL, Bludau I, Mehnert M, Seifert M, Emmenlauer M, Sorg I, Bezrukov F, Bena FS, Zhou H, Dehio C, Testa G, Saez-Rodriguez J et al (2019) Multi-omic measurements of heterogeneity in HeLa cells across laboratories. *Nat Biotechnol* **37**: 314-322

Lopacinski AB, Sweatt AJ, Smolko CM, Gray-Gaillard E, Borgman CA, Shah M, Janes KA (2021) Modeling the complete kinetics of coxsackievirus B3 reveals human determinants of host-cell feedback. *Cell Syst* **12**: 304-323 e313

Ludwig C, Gillet L, Rosenberger G, Amon S, Collins BC, Aebersold R (2018) Data-independent acquisition-based SWATH-MS for quantitative proteomics: a tutorial. *Mol Syst Biol* **14**: e8126

Martin PCN, Kim H, Lovkvist C, Hong BW, Won KJ (2022) Vesalius: high-resolution in silico anatomization of spatial transcriptomic data using image analysis. *Mol Syst Biol* **18**: e11080

Matsushime H, Ewen ME, Strom DK, Kato JY, Hanks SK, Roussel MF, Sherr CJ (1992) Identification and properties of an atypical catalytic subunit (p34PSK-J3/cdk4) for mammalian D type G1 cyclins. *Cell* **71**: 323-334

Mertins P, Mani DR, Ruggles KV, Gillette MA, Clauser KR, Wang P, Wang X, Qiao JW, Cao S, Petralia F, Kawaler E, Mundt F, Krug K, Tu Z, Lei JT, Gatza ML, Wilkerson M, Perou CM, Yellapantula V, Huang KL et al (2016) Proteogenomics connects somatic mutations to signalling in breast cancer. *Nature* **534**: 55-62

Montagud A, Beal J, Tobalina L, Traynard P, Subramanian V, Szalai B, Alfoldi R, Puskas L, Valencia A, Barillot E, Saez-Rodriguez J, Calzone L (2022) Patient-specific Boolean models of signalling networks guide personalised treatments. *Elife* **11**

Nagdas S, Kashatus JA, Nascimento A, Hussain SS, Trainor RE, Pollock SR, Adair SJ, Michaels AD, Sesaki H, Stelow EB, Bauer TW, Kashatus DF (2019) Drp1 Promotes KRas-Driven Metabolic Changes to Drive Pancreatic Tumor Growth. *Cell Rep* **28**: 1845-1859 e1845

Newman AM, Steen CB, Liu CL, Gentles AJ, Chaudhuri AA, Scherer F, Khodadoust MS, Esfahani MS, Luca BA, Steiner D, Diehn M, Alizadeh AA (2019) Determining cell type abundance and expression from bulk tissues with digital cytometry. *Nat Biotechnol* **37**: 773-782

Nusinow DP, Szpyt J, Ghandi M, Rose CM, McDonald ER, 3rd, Kalocsay M, Jane-Valbuena J, Gelfand E, Schweppe DK, Jedrychowski M, Golji J, Porter DA, Rejtar T, Wang YK, Kryukov GV, Stegmeier F, Erickson BK, Garraway LA, Sellers WR, Gygi SP (2020) Quantitative Proteomics of the Cancer Cell Line Encyclopedia. *Cell* **180**: 387-402 e316

Pappireddi N, Martin L, Wuhr M (2019) A Review on Quantitative Multiplexed Proteomics. *ChemBioChem* **20**: 1210-1224

Pereira EJ, Burns JS, Lee CY, Marohl T, Calderon D, Wang L, Atkins KA, Wang CC, Janes KA (2020) Sporadic activation of an oxidative stress-dependent NRF2-p53 signaling network in breast epithelial spheroids and premalignancies. *Sci Signal* **13**: eaba4200

Reimegard J, Tarbier M, Danielsson M, Schuster J, Baskaran S, Panagiotou S, Dahl N, Friedlander MR, Gallant CJ (2021) A combined approach for single-cell mRNA and intracellular protein expression analysis. *Commun Biol* **4**: 624

Richards AL, Eckhardt M, Krogan NJ (2021) Mass spectrometry-based protein-protein interaction networks for the study of human diseases. *Mol Syst Biol* **17**: e8792

Savitski MM, Mathieson T, Zinn N, Sweetman G, Doce C, Becher I, Pachi F, Kuster B, Bantscheff M (2013) Measuring and managing ratio compression for accurate iTRAQ/TMT quantification. *J Proteome Res* **12**: 3586-3598

Scime A, Li L, Ciavarrà G, Whyte P (2008) Cyclin D1/cdk4 can interact with E2F4/DP1 and disrupts its DNA-binding capacity. *J Cell Physiol* **214**: 568-581

Shah M, Smolko CM, Kinicki S, Chapman ZD, Brautigan DL, Janes KA (2017) Profiling Subcellular Protein Phosphatase Responses to Coxsackievirus B3 Infection of Cardiomyocytes. *Mol Cell Proteomics* **16**: S244-S262

Taggart JC, Zauber H, Selbach M, Li GW, McShane E (2020) Keeping the Proportions of Protein Complex Components in Check. *Cell Syst* **10**: 125-132

Tasaki S, Xu J, Avey DR, Johnson L, Petyuk VA, Dawe RJ, Bennett DA, Wang Y, Gaiteri C (2022) Inferring protein expression changes from mRNA in Alzheimer's dementia using deep neural networks. *Nat Commun* **13**: 655

Thompson A, Schafer J, Kuhn K, Kienle S, Schwarz J, Schmidt G, Neumann T, Johnstone R, Mohammed AK, Hamon C (2003) Tandem mass tags: a novel quantification strategy for comparative analysis of complex protein mixtures by MS/MS. *Anal Chem* **75**: 1895-1904

Wang L, Paudel BB, McKnight RA, Janes KA (2023) Nucleocytoplasmic transport of active HER2 causes fractional escape from the DCIS-like state. *Nat Commun* **14**: 2110

Wang M, Herrmann CJ, Simonovic M, Szklarczyk D, von Mering C (2015) Version 4.0 of PaxDb: Protein abundance data, integrated across model organisms, tissues, and cell-lines. *Proteomics* **15**: 3163-3168

Wilhelm M, Schlegl J, Hahne H, Gholami AM, Lieberenz M, Savitski MM, Ziegler E, Butzmann L, Gessulat S, Marx H, Mathieson T, Lemeer S, Schnatbaum K, Reimer U, Wenschuh H, Mollenhauer M, Slotta-Huspenina J, Boese JH, Bantscheff M, Gerstmair A et al (2014) Mass-spectrometry-based draft of the human proteome. *Nature* **509**: 582-587

Xia Y, Deng Y, Zhou Y, Li D, Sun X, Gu L, Chen Z, Zhao Q (2020) TSPAN31 suppresses cell proliferation in human cervical cancer through down-regulation of its antisense pairing with CDK4. *Cell Biochem Funct* **38**: 660-668

Yang M, Petralia F, Li Z, Li H, Ma W, Song X, Kim S, Lee H, Yu H, Lee B, Bae S, Heo E, Kaczmarczyk J, Stepniak P, Warchol M, Yu T, Calinawan AP, Boutros PC, Payne SH, Reva B et al (2020) Community Assessment of the Predictability of Cancer Protein and Phosphoprotein Levels from Genomics and Transcriptomics. *Cell Syst* **11**: 186-195 e189

Zhang H, Liu T, Zhang Z, Payne SH, Zhang B, McDermott JE, Zhou JY, Petyuk VA, Chen L, Ray D, Sun S, Yang F, Chen L, Wang J, Shah P, Cha SW, Aiyetan P, Woo S, Tian Y, Gritsenko MA et al (2016) Integrated Proteomic Characterization of Human High-Grade Serous Ovarian Cancer. *Cell* **166**: 755-765

25th Jun 2024

RE: Manuscript MSB-2023-11919RR, Proteome-wide copy-number estimation from transcriptomics

Dear Dr Janes,

Thank you for sending us your revised manuscript. We have now heard back from the three reviewers who agreed to evaluate your study. As you will see below, Reviewer #3 still raises significant concerns about issues that were already brought up during the first round of review. Furthermore, Reviewer #2 explicitly indicated that they do not recommend publication in Molecular Systems Biology.

Under these circumstances and considering the overall low level of support provided by the reviewers, I see no other choice than to return the manuscript with the message that we cannot offer to publish it.

I am very sorry that I cannot bring better news on this occasion. I hope that the points raised in the reports will prove helpful to you and that you will not be discouraged from submitting future work to Molecular Systems Biology.

Sincerely,
Jingyi

Jingyi Hou, PhD
Scientific Editor
Molecular Systems Biology

Reviewer #1:

The authors have addressed the points I raised. I think this is an interesting approach and think the paper can now be accepted for publication.

Reviewer #2:

The paper appears to be fundamentally the same and remains flawed. I cannot recommend publication.

Reviewer #3:

I appreciate that the authors have taken great effort to address the concerns raised by all reviewers and they have significantly improved the presentation of their results.

Yet, I am still skeptical regarding my main concern point 1 from the first round of reviews, i.e. the issue that PaxDB and PTR seem to perform significantly worse than random guessing. I understand that the authors want to show that the competing methods make systematic errors in their predictions.

Before explaining my remaining concerns I want to emphasize that I am not involved in any way in PaxDB or PTR. In fact, I have never published a tool for protein level prediction. Thus, there is no conflict of interest.

"Because PaxDb and PTR were calibrated identically to SWATH-MS, it implies that the aggregate distribution of thousands of predicted proteins is skewed or displaced from what it should be."

What exactly does that mean? The authors median-centered the protein levels at 10,000 copies per cell. But there could be remaining variation in how exactly absolute protein abundance is quantified that might lead to systematic deviations.

How is 'absolute protein abundance' defined in the first place? The authors used the top 3 most abundant peptides of each protein. That is a common approach, but certainly not the only possible one. One could also go for the single most abundant peptide (i.e. only using one peptide per protein), one could use more than the top 3 peptides or one could use different selection criteria for the peptides. (E.g. mapDIA is using the correlation between peptides as one selection criterion. doi: 10.1016/j.jprot.2015.09.013) All of these approaches are legitimate, but I don't think there is a trivial relationship between the

actual protein concentration (in mol) and any of those metrics. So, what is the 'true' protein level?

Another possible source of bias is the proteomics measurement itself. Even if all methods were calibrated against SWATH measurements, the data might have been produced on different instruments, using different LC columns, etc. ... all of which might introduce dataset-specific biases.

Thus, in conclusion I don't think there is a fair way of comparing predictions of 'absolute' protein levels across methods, unless all of them were calibrated on the identical input data. (As opposed to: calibrated on identically processed input data.)

This point needs to be addressed.

** As a service to authors, EMBO Press offers the possibility to directly transfer declined manuscripts to another EMBO Press title or to the open access journal Life Science Alliance launched in partnership between EMBO Press, Rockefeller University Press and Cold Spring Harbor Laboratory Press. The full manuscript and if applicable, reviewers' reports, are automatically sent to the receiving journal to allow for fast handling and a prompt decision on your manuscript. For more details of this service, and to transfer your manuscript please click on Link Not Available. **

Reviewer #1:

The authors have addressed the points I raised. I think this is an interesting approach and think the paper can now be accepted for publication.

We thank Reviewer #1 for their positivity and kindly ask that they comment on the points raised by other reviewers in the prior revision and in this minor revision.

Reviewer #2:

The paper appears to be fundamentally the same and remains flawed. I cannot recommend publication.

We are disheartened by this evaluation given our efforts to address Reviewer #2's concerns as completely as possible given the first-round comments provided. In the major revision, we tackled the critique about TMT ratio compression with quantitative immunoblotting for the protein target that was questioned (Fig. 2D–F). Additionally, considerable thought was put into Point #3 of Reviewer #2:

3. The manuscript fails to provide a convincing demonstration of the model's usefulness. In principle, there is value in developing the kind of model the authors describe. However, the model uses many parameters; just slightly beating simple alternatives is not particularly exciting. To make this an interesting study, I would like to see some strong, non-obvious predictions that are rigorously tested.

Using Pinferna, we correctly discriminated between 2800 and 4800 inferred copies of receptor abundance for enterovirus infection (Figs. 4H–K and EV4I–L of the revision) and confirmed predicted cross-gene regulatory effects on protein expression of CDK4 (Fig. 6 and Appendix Figs. S6 and S7 of the revision). As a systems biologist with over 20 years of experience in models—both statistical (*Science* **310**:1646-53 [2005]; *Nature* **448**:604-8 [2007]; *Cell* **135**:343-54 [2008]; *Cell Host Microbe* **13**:67-76 [2013]; *Sci Signal* **9**:a59 [2016]) and mechanistic (*Nat Cell Biol* **16**:345-56 [2014]; *Cell Syst* **2**:112-21 [2016]; *Sci Signal* **13**:eaba4200 [2020]; *Cell Syst* **12**:304-23.e1-e13 [2021]; *Nat Commun* **14**:2110 [2023])—I have a strong sense for model predictions that are non-obvious. The experiments in Figs. 4H–K, 6, EV4I–L, and Appendix Figs. S6 and S7 were extremely risky and would not have been pursued if not for the predictions of Pinferna.

Reviewer #3:

I appreciate that the authors have taken great effort to address the concerns raised by all reviewers and they have significantly improved the presentation of their results.

We thank Reviewer #3 for reading the revised manuscript carefully along with our responses to the other reviewers.

Yet, I am still skeptical regarding my main concern point 1 from the first round of reviews, i.e. the issue that PaxDb and PTR seem to perform significantly worse than random guessing. I understand that the authors want to show that the competing methods make systematic errors in their predictions.

Before explaining my remaining concerns I want to emphasize that I am not involved in any way in PaxDb or PTR. In fact, I have never published a tool for protein level prediction. Thus, there is no conflict of interest.

"Because PaxDb and PTR were calibrated identically to SWATH-MS, it implies that the aggregate distribution of thousands of predicted proteins is skewed or displaced from what it should be."

What exactly does that mean? The authors median-centered the protein levels at 10,000 copies per cell. But there could be remaining variation in how exactly absolute protein abundance is quantified that might lead to systematic deviations.

How is 'absolute protein abundance' defined in the first place? The authors used the top 3 most abundant peptides of each protein. That is a common approach, but certainly not the only possible one. One could also go for the single most abundant peptide (i.e. only using one peptide per protein), one could use more than the top 3 peptides or one could use different selection criteria for the peptides. (E.g. mapDIA is using the correlation between peptides as one selection criterion. doi: 10.1016/j.jprot.2015.09.013) All of these approaches are legitimate, but I don't think there is a trivial relationship between the actual protein concentration (in mol) and any of those metrics. So, what is the 'true' protein level?

Another possible source of bias is the proteomics measurement itself. Even if all methods were calibrated against SWATH measurements, the data might have been produced on different instruments, using different LC columns, etc. ... all of which might introduce dataset-specific biases.

Thus, in conclusion I don't think there is a fair way of comparing predictions of 'absolute' protein levels across methods, unless all of them were calibrated on the identical input data. (As opposed to: calibrated on identically processed input data.)

This point needs to be addressed.

In the major revision, we attempted to clarify the Δ CDF performance metric in Appendix Text S1 and Fig. EV3A,B. However, Reviewer #3's remarks indicate that we need to do a better job explaining the results shown in Fig. 3 and rigorously exclude alternative explanations for the findings. There are no concerns about conflict of interest, our sense is that we may simply be using different terminology than Reviewer #3.

First, we define "absolute protein abundance" as accurate measurements of different proteins within the same sample. For example, in the SWATH-TMT meta-assembly (Dataset EV1) the estimate is that there are ~39,000 copies per cell of EGFR and ~1200 copies per cell of its bypass receptor, AXL, in the lung squamous cell carcinoma line, LUDLU1. The actual numbers themselves are dependent on the choice to center (or calibrate) the overall protein distribution at 10,000 copies per cell. Choosing a different calibration (e.g., a larger or smaller cell) would change the exact numbers but not the estimate that there are 33x as many EGFR protein copies in LUDLU1 cells as AXL protein copies. The thinking here is analogous to RNA-seq profiles, which are commonly reported in transcripts per million (TPM). The precise number of mRNA molecules per cell is less important than the fact that the method counts sequences of any type, yielding 78 TPM for *EGFR* and 1.2 TPM for *AXL* in LUDLU1 cells (Dataset EV2).

Protein data used by PaxDb and PTR lie on arbitrary scales. To use as methods for absolute prediction, we projected the measurements linearly on a biologically plausible scale. This calibration was performed as favorably as possible for PaxDb and PTR without coaxing a best fit—we extracted the proteins that could be predicted by each method from the meta-assembly, calculated the median of the extracted distribution, and then calibrated the median of that method's protein distribution to the extracted median (see "Alternative methods for protein abundance estimation" in Methods). The resulting calibrations were close to 10,000 copies per cell but slightly smaller (8000–9000 copies per cell). After making predictions, we did not alter the output values any further, as we considered them the best-calibrated measurements for a given method.

Please note that Pinferna was not recalibrated at all for independent datasets.

Despite the favorable upstream calibration of PaxDb and PTR, their predictions are both *shifted* (log-log intercept $\neq 0$) and *skewed* (log-log slope $\neq 1$) relative to the absolute protein abundances of the test datasets (Fig. R1A–F). By contrast, the predictions of Pinferna have a much stronger one-to-one correspondence in all cases (Fig. R1G–I). On the surface, the log-log correlations for PaxDb and PTR might not appear too bad, but

Fig. R1. PaxDb and PTR predictions of absolute protein abundance are shifted or skewed compared to SWATH measurements, the median null of random guesses, and Pinferna. Scatterplots for (A–C) PaxDb, (D–F) PTR, (G–I) Pinferna, and (J–L) the median null are shown for representative samples from (A,D,G,J) NCI-60, (B,E,H,K) prostate cancer, and (C,F,I,L) benign prostate as in Fig. 3C–E.

they are significantly worse than random guessing because guessing an abundance based on past measures of that protein does remarkably well under most circumstances (Fig. R1J–L). To illustrate, we return to AXL and its median abundance of ~1200 copies per cell in the SWATH-TMT meta-assembly (IQR: 600–3000 copies per cell). Randomly sampling from this empirical distribution does an adequate job approximating AXL abundances in the test datasets, which have a median abundance of 3400 copies per cell (IQR: 1700–4800 copies per cell). The stringency of this median null is why the apparent gains in Δ CDF appear modest (as described in Appendix Text S1).

Fig. R2. Arbitrary changes in calibration and power-law scaling improve Δ CDF performance of PaxDb and PTR, but they do not surpass Pinferna. Changes in calibration for (A,B) PaxDb and (C,D) PTR relative to the existing scaling (gray dashed; Methods) based on 10,000 copies per cell in the SWATH-TMT meta-assembly (Dataset EV1) for NCI-60 samples with (B,D) or without (A,C) power-law scaling of the assumed linear distribution of protein abundances.

Reviewer #3 is correct that there could be other latent sources of variation in the alternative data or preprocessing steps that pertain to competing methods. We believe this is somewhat less likely for PaxDb, because that repository aggregates data from many different sources and reports them as “parts per million” in a way that is analogous to TPM for RNA-seq. PTR originates from a single study, though, and could be highly dependent on latent effects. We tackled the issue by defining a best-possible performance for PaxDb and PTR that allowed their predictions to undergo arbitrary calibrations (to improve shift) and power-law re-scalings across the proteome (to improve skew). By coaxing a best fit in this way, we found that Δ CDF could be improved for both methods to near the median null or past it (not shown), but neither exceeded the baseline performance of Pinferna (Fig. R2). We conclude that there is no proteome-wide adjustment of PaxDb or PTR predictions that gleans absolute protein abundances as accurately as Pinferna.

Last, we address the comment on dataset-specific biases in the proteomics measurement itself. Our

external view is that the experimental proteomics community has matured to the point that instruments, columns, etc. have relatively minor impacts on absolute protein quantification as defined here. For example, this project would not have continued if SWATH and TMT data had not been as internally consistent as we found (Fig. 1B and EV1B). Changes in SWATH quantitation pipelines had no discernible effects on results (Fig. EV1C,D), further bolstering the reliability of our findings. We see direct analogies to transcriptomics—although it would be foolish to compare *EGFR* mRNA abundance between two different studies without some form of batch corrections, most would expect the *EGFR*-to-*AXL* TPM proportion to be internally consistent in each study.

We appreciate Reviewer #3's patience through our attempts to understand and address the remaining point raised.

30th Jul 2024

Manuscript Number: MSB-2023-11919RRR-Q
Title: Proteome-wide copy-number estimation from transcriptomics
Author: Andrew Sweatt
Cameron Griffiths
Sarah Groves
B. Paudel
Lixin Wang
David Kashatus
Kevin Janes

Dear Kevin,

Thank you for your message asking us to reconsider our decision regarding your manuscript MSB-2023-11919RR and for sending the new analyses addressing Reviewer #3's concerns. We have shared your responses with Reviewer #3 and have requested their feedback. Additionally, since Reviewer #2 did not provide a detailed report during the second round of review, we asked Reviewer #3 to evaluate whether the major points #1 and #2 raised by Reviewer #2 have been adequately addressed.

As you will see from the comments below, Reviewer #3 has indicated that their previous concerns regarding the comparison of Pinferna to competing methods remain unaddressed and they cannot recommend the article for publication until these points are sufficiently addressed.

As you may already know, our general policy allows for only a single round of major revision. Given that the manuscript has already undergone two rounds of revisions, we would typically not consider the study further. However, after consulting with Reviewer #3, we feel that the remaining concerns could potentially be addressed. Therefore, we have decided to offer you the opportunity to address them in another exceptional round of major revision. Please note, however, that the revised manuscript will be reviewed again, and we cannot guarantee its eventual acceptance. If you do choose to proceed, we would ask you to enclose with your submission a point-by-point response to the points raised in the present review.

On a more editorial level, please address the following:

- Please ensure the funding information is complete in the online submission system.
- Remove "Author contributions" section from the manuscript file.
- Remove "data not shown"(page 42).
- Reference format: et al should be used after 10 author names.
- Table 1 and Table S3 are called out but don't exist. Please remove the callouts.
- We noticed that you provided a link to eight Datasets deposited online; each dataset file could have been uploaded to the submission system since the largest one is 101.9 MB (<300MB which is OK); the legends need to be removed from the manuscript file and each should be placed as a separate sheet/tab in each Excel file.
- The "Appendix Supplementary Methods" need to be combined with the "Methods section" in the main manuscript text.
- Please provide a "standfirst text" summarizing the study in one or two sentences (approximately 250 characters, including space), three to four "bullet points" highlighting the main findings.
- Please note that the exact p values are not provided in the legends of figures 3a, c-e, i, k; 4i, k; 5b; 6b, d, f; EV 2c; EV 3f; EV4b.
- Please note that the box plot needs to be defined in terms of minima, maxima, centre, bounds of box and whiskers, and percentile in the legend of figure EV 1f.
- Please note that information related to n is missing in the legend of figure EV 1f.

When you resubmit your manuscript, please download our CHECKLIST (<https://bit.ly/EMBOPressAuthorChecklist>) and include

the completed form in your submission.

Please note that the Author Checklist will be published alongside the paper as part of the transparent process (<https://www.embopress.org/page/journal/17444292/authorguide#transparentprocess>).

I look forward to receiving your revised manuscript soon.

Kind regards,
Jingyi

Jingyi Hou, PhD
Scientific Editor
Molecular Systems Biology

Reviewer #3

I appreciate the authors efforts to further clarify their approach. Unfortunately, I remain skeptical about my previous concerns. Figure R1 seems to suggest that Pinferna indeed performs better than the competitor methods, but it raises concerns. Especially Panels D to F (corresponding to PTR) are puzzling. If the predicted protein levels are calibrated to the median of the proteins available in the meta-assembly and those that could be predicted by the method, how can the medians of observed and predicted values differ so drastically?

In general, the calibration should be performed to remove both skew and shift. If Pinferna continues to outperform the competing methods, I see no reason not to show the (in my opinion) fairer comparison.

Skew and shift both point to either 1) systematic differences in what kind of data (e.g., in dynamic range or methods of combining peptide and fragment amounts to protein levels) was used to train the methods or 2) fundamental flaws in the methods, that are easily fixable (by adjusting skew and shift based on public data). The purpose of the comparison is not just to show that Pinferna achieves a higher Delta-CDF but that it uncovers true patterns in the relationships of RNA and protein levels that are missed by the other methods. The readers have to be convinced that a simple tuning of the competitors is not sufficient to reach the performance of Pinferna.

The authors state that the PTRs were computed as the difference of protein levels and FPKMs (both log₁₀). In the original publication by Eraslan et al, the PTRs were not computed with FPKMs but with normalized values from the DESeq2 workflow. DESeq2 does not correct for gene length. I'm not sure what effect this has on the present analysis. Intuitively one would assume, that using FPKM values would be beneficial, as protein amounts estimated on the most abundant peptides should be influenced by protein length to a small degree only.

Overall, I cannot recommend the article for publication until these points are sufficiently addressed.

Reviewer #3's feedback on your responses to Reviewer #2's comments

Regarding their first major concern, reviewer 2 raises the point that using SWATH data might be suboptimal for anchoring the TMT measurements, stating that SWATH data were reported to show a suboptimal correlation with true protein concentrations. If the goal of a study is to predict true absolute protein concentrations from RNA-Seq data, this concern might be valid. If the goal is instead to predict what abundances would be measured with SWATH, the concern is not justified. Sweatt et al. expressly aim to predict true protein abundances (not impacted by SWATH specific biases for a given peptide). Given this, I would agree that the concern is generally valid.

Reviewer 2 states that SWATH measurements only show weak correlations with absolute protein abundances. Sweatt et al. point to Figure 1B and to a publication by Collins et al. to address this. While Figure 1B demonstrates that the SWATH measurements (adjusted with TMT measurements) used in this study are very similar across cell lines (a lower boundary for reproducibility), it does not speak to the correlation of SWATH measurements to absolute protein abundances. In the study by Collins et al. the question of linearity of SWATH measurements in the presence of greatly differing concentrations of a peptide is addressed. To my understanding it does not address the question of peptide specific biases in SWATH measurements. Given that the mean expression of transcripts and average protein levels determined by SWATH generally correlate well across genes, I do not share the concern of reviewer 2 that these biases are severe enough to fundamentally affect how useful Pinferna can be as a method. These peptide specific biases might affect the performance of the methods not developed with SWATH data in mind. In my opinion, this major concern could be addressed by directly comparing the SWATH measurements to absolute abundance measurements over a wide range of proteins within the same cell line. Because this requires extensive experimental work (<https://doi.org/10.1016/j.cell.2012.09.019>), I recommend to adapt the claim of predicting absolute abundances instead. The authors defined absolute protein quantification as such:

"First, we define "absolute protein abundance" as accurate measurements of different proteins within the same sample." To my understanding this definition would only be valid for relative differences within the same sample. Absolute quantification instead requires a way to translate abundances directly to molecular numbers, i.e. a zero on a scale.

Therefore, the claim that absolute abundances (without any measurement bias) are predicted, should be abandoned, in favor of stating that relative SWATH-abundances are predicted. This does not mean that the approach is less useful.

Reviewer 2 raises a second major concern, that the dynamic range of TMT is not sufficient to learn the true relationship between mRNA and protein abundances. Here Sweatt et al. make several convincing arguments. First, they argue, that because the CCLE data contains controls, the dynamic range is larger than assumed by reviewer 2. Second, the training of models would only be affected for genes, where the range of protein abundances is very large. A difference of 100x for protein levels does occur, but is not very common. For the affected proteins, most data points would still be of high quality, as only the outliers would be affected by the error introduced by the lack of dynamic range.

Therefore, I would argue that the second concern was sufficiently addressed.

EDITORIAL COMMENTS

- *Please ensure the funding information is complete in the online submission system.*

Completed.

- *Remove "Author contributions" section from the manuscript file.*

Removed.

- *Remove "data not shown"(page 42).*

This figure caption did not state "data not shown" but instead referred to examples not included in the main figure. We have revised the caption to read "besides those of".

- *Reference format: et al should be used after 10 author names.*

Fixed, and we also changed the reference formatting to agree with current MSB papers.

- *Table 1 and Table S3 are called out but don't exist. Please remove the callouts.*

Table 1 is the Reagents and Resources file that is required of all Method submissions. We neglected to add a figure caption for Table 1 and have added it to the second revision. Table S3 referred to a supplementary table of Champi et al, 2020; we have removed this callout.

- *We noticed that you provided a link to eight Datasets deposited online; each dataset file could have been uploaded to the submission system since the largest one is 101.9 MB (<300MB which is OK); the legends need to be removed from the manuscript file and each should be placed as a separate sheet/tab in each Excel file.*

Thank you, we were unaware of the flexible file requirements. We have uploaded the eight Datasets separately as xlsx files with the first sheet as the caption. Legends have been removed from the manuscript file.

-*The "Appendix Supplementary Methods" need to be combined with the "Methods section" in the main manuscript text.*

Combined.

- *Please provide a "standfirst text" summarizing the study in one or two sentences (approximately 250 characters, including space), three to four "bullet points" highlighting the main findings.*

Apologies, we must have neglected to upload this synopsis text during the last revision. It has now been included.

- *Please note that the exact p values are not provided in the legends of figures 3a, c-e, i, k; 4i, k; 5b; 6b, d, f; EV 2c; EV 3f; EV4b.*

We have added exact p values throughout, including Appendix Figs. S2 and S7.

- *Please note that the box plot needs to be defined in terms of minima, maxima, centre, bounds of box and whiskers, and percentile in the legend of figure EV 1f.*

We have added details on the boxplots for Fig. EV1F.

- *Please note that information related to n is missing in the legend of figure EV 1f.*

We have added in the caption that $n = 4366$ genes for Fig. EV1F.

REVIEWER #3

I appreciate the authors efforts to further clarify their approach. Unfortunately, I remain skeptical about my previous concerns.

Figure R1 seems to suggest that Pinferna indeed performs better than the competitor methods, but it raises concerns. Especially Panels D to F (corresponding to PTR) are puzzling. If the predicted protein levels are calibrated to the median of the proteins available in the meta-assembly and those that could be predicted by the method, how can the medians of observed and predicted values differ so drastically?

In the earlier appeal, we mapped the predicted proteomes of PaxDb and PTR so that each sample has the same predicted median copy number. We had not explicitly shifted predictions to align optimally with measured results or adjusted the skewness by scaling the predictions. In the second revision, we calibrate shift and scaling before making predictions with PTR or PaxDb. Details of this calibration (and alternatives considered) are explained below.

In general, the calibration should be performed to remove both skew and shift. If Pinferna continues to outperform the competing methods, I see no reason not to show the (in my opinion) fairer comparison.

Skew and shift both point to either 1) systematic differences in what kind of data (e.g., in dynamic range or methods of combining peptide and fragment amounts to protein levels) was used to train the methods or 2) fundamental flaws in the methods, that are easily fixable (by adjusting skew and shift based on public data). The purpose of the comparison is not just to show that Pinferna achieves a higher Delta-CDF but that it uncovers true patterns in the relationships of RNA and protein levels that are missed by the other methods. The readers have to be convinced that a simple tuning of the competitors is not sufficient to reach the performance of Pinferna.

We greatly appreciate this suggestion, as it indicated the best way to put competing methods on the same footing as ours. In the second revision, we began by applying PaxDb and PTR to the CCLE meta-assembly that was used to train Pinferna (Fig. 1). Predictions were then calibrated to match the measured proteins by shifting and scaling the predicted outputs (y): $y_{new} = ay^b$, where a is the shift parameter and b is the scaling parameter estimated by maximizing the Δ CDF of the calibration (Methods; Appendix Fig. S1 of the second revision). Thereafter, shift and scale parameters were kept fixed to predict NCI-60 cell lines and primary prostate samples outside the training data.

Fitting a and b with a Δ CDF-based cost function causes the predicted-vs.-measured plot for PTR to appear suboptimal (Appendix Fig. S1B). If instead a and b are parameterized to predicted-vs.-measured, the Appendix Fig. S1B plot looks better (Fig. R1), but the Δ CDF of the calibration (and all resulting predictions) is considerably worse (not shown). This is because Δ CDF is based on residuals scaled to the overall variation of a protein across the meta-assembly, whereas a predicted-vs.-measured fit is based on minimizing a standard residual (i.e., sum-of-squared error). Given Reviewer #3's prior difficulties with reporting very-low Δ CDF statistics for PaxDb and PTR, we decided to optimize the shift-scaling so that these measures looked as favorable as possible.

Even with pre-calibration to Δ CDF, PaxDb and PTR remained inferior to Pinferna for NCI-60 predictions (Fig. 3C). For primary prostate samples, we found that prediction accuracy by Δ CDF was comparable across the methods after shift-scale calibration (Fig. 3D, inset). Although nominally as accurate, the overall predictions for PaxDb and PTR did not reflect the marginal distribution of measured proteins as well as Pinferna (Fig. 3D). We conclude that Pinferna provides the best mapping of RNA to protein both in vitro and in vivo.

Fig. R1. Calibration of PTR by predicted-measured concordance or Δ CDF accuracy.

The authors state that the PTRs were computed as the difference of protein levels and FPKMs (both log10). In the original publication by Eraslan et al, the PTRs were not computed with FPKMs but with normalized values from the DESeq2 workflow. DESeq2 does not correct for gene length. I'm not sure what effect this has on the

present analysis. Intuitively one would assume, that using FPKM values would be beneficial, as protein amounts estimated on the most abundant peptides should be influenced by protein length to a small degree only.

Overall, I cannot recommend the article for publication until these points are sufficiently addressed.

Reviewer #3 is correct that Table EV2 of Eraslan et al. reports discrete DESeq2 counts. However, in the methods of this paper it states (relevant regions underlined):

To estimate the mature mRNA levels, for each sample (each replicate in each tissue) the number of reads that map to exonic and intronic regions of the transcript (which was decided to be used based on the major protein isoform) was counted separately (Table EV2) and then normalized by the total exonic and intronic region lengths, respectively. Next, the intronic counts normalized by the intronic region length were subtracted from exonic counts normalized by the exonic region length. The resulting normalized exonic counts per sample (i.e., each replicate of each tissue) were corrected by the library size factor obtained with the Bioconductor package DESeq2 and further log-transformed (log10).

Which was how we calculated FPKM. Our interpretation is confirmed by the readme file for Table EV3 that summarizes the PTR values:

Tissue specific mRNA (median across replicates, FPKM-log10), protein (iBAQ-log10) measurements and protein-to-mRNA ratios (log10) of 11,575 transcripts considered in the study. Values that are 0 in natural scale, and FPKM values smaller than 1 in log10 scale are encoded as missing values (NA).

Thus, we do not believe that the extensive shift-scaling needed for PTR is due to misapplication of the method as described in the original publication.

A new complication arises with FPKM when performing the shift-scaling calibration requested by Reviewer #3 (see above), because the RNA-seq data of the meta-assembly are reported in the manuscript as TPM for Pinferna rather than FPKM (Dataset EV2). Since the FPKM-to-TPM is simply a coefficient shift and we were already committed to shift-scale calibration for the revision, we calibrated PTR to TPM inputs of the meta-assembly and then made all predictions with RNA-seq as TPM rather than FPKM in the revision. The Methods have been revised to reflect this change.

Reviewer #3's feedback on your responses to Reviewer #2's comments

Regarding their first major concern, reviewer 2 raises the point that using SWATH data might be suboptimal for anchoring the TMT measurements, stating that SWATH data were reported to show a suboptimal correlation with true protein concentrations. If the goal of a study is to predict true absolute protein concentrations from RNA-Seq data, this concern might be valid. If the goal is instead to predict what abundances would be measured with SWATH, the concern is not justified. Sweatt et al. expressly aim to predict true protein abundances (not impacted by SWATH specific biases for a given peptide). Given this, I would agree that the concern is generally valid.

We have changed the emphasis of the manuscript to absolute ("true") protein abundances to proportional protein abundances as defined by SWATH. See below.

Reviewer 2 states that SWATH measurements only show weak correlations with absolute protein abundances. Sweatt et al. point to Figure 1B and to a publication by Collins et al. to address this. While Figure 1B demonstrates that the SWATH measurements (adjusted with TMT measurements) used in this study are very similar across cell lines (a lower boundary for reproducibility), it does not speak to the correlation of SWATH measurements to absolute protein abundances. In the study by Collins et al. the question of linearity of SWATH measurements in the presence of greatly differing concentrations of a peptide is addressed. To my understanding it does not address the question of peptide specific biases in SWATH measurements. Given that the mean expression of transcripts and average protein levels determined by SWATH generally correlate well across genes, I do not share the concern of reviewer 2 that these biases are severe enough to fundamentally affect how useful Pinferna can be as a method. These peptide specific biases might affect the performance of the methods not developed with SWATH data in mind. In my opinion, this major concern could be addressed by directly comparing the SWATH measurements to absolute abundance measurements over a wide range of proteins within the same cell line. Because this requires extensive experimental work

(<https://doi.org/10.1016/j.cell.2012.09.019>), I recommend to adapt the claim of predicting absolute abundances instead. The authors defined absolute protein quantification as such:

"First, we define "absolute protein abundance" as accurate measurements of different proteins within the same sample." To my understanding this definition would only be valid for relative differences within the same sample. Absolute quantification instead requires a way to translate abundances directly to molecular numbers, i.e. a zero on a scale.

Therefore, the claim that absolute abundances (without any measurement bias) are predicted, should be abandoned, in favor of stating that relative SWATH-abundances are predicted. This does not mean that the approach is less useful.

We thank Reviewer #3 for providing feedback on Reviewer #2's comments and for communicating a clear path to move ahead. We have abandoned the term "absolute protein abundances" and substituted "proportional protein abundances" to communicate quantitative differences between proteins measured by SWATH. We avoided "relative protein abundances" because many use this term when referring to fold-change abundances of a single protein across different samples.

Reviewer 2 raises a second major concern, that the dynamic range of TMT is not sufficient to learn the true relationship between mRNA and protein abundances. Here Sweatt et al. make several convincing arguments. First, they argue, that because the CCLE data contains controls, the dynamic range is larger than assumed by reviewer 2. Second, the training of models would only be affected for genes, where the range of protein abundances is very large. A difference of 100x for protein levels does occur, but is not very common. For the affected proteins, most data points would still be of high quality, as only the outliers would be affected by the error introduced by the lack of dynamic range.

Therefore, I would argue that the second concern was sufficiently addressed.

Thank you.

2nd Sep 2024

Manuscript number: MSB-2023-11919RRRR

Title: Proteome-wide copy-number estimation from transcriptomics

Dear Kevin,

Thank you again for sending us your revised manuscript. As you will see below, the reviewer is now satisfied with the modifications made and I am pleased to inform you that your paper has been accepted for publication.

Your manuscript will then be processed for publication by EMBO Press. It will be copy edited and you will receive page proofs prior to publication. Please note that you will be contacted by Springer Nature Author Services to complete licensing and payment information.

Kind regards,
Jingyi

Jingyi Hou, PhD
Scientific Editor
Molecular Systems Biology

Reviewer #4:

I appreciate the efforts made by the authors to improve the manuscript. My concerns have been sufficiently addressed and I recommend the paper for publication
